

# Correlation between marine aerosol optical properties and wind fields over remote oceans with use of spaceborne lidar observations

Kangwen Sun[1], Guangyao Dai[1], Songhua Wu[1, 2, 3], Oliver Reitebuch[4], Holger Baars[5], Jiqiao Liu[6], Suping Zhang[7]

[1]College of Marine Technology, Faculty of Information Science and Engineering, Ocean University of China, 266100 Qingdao, China
[2]Laoshan Laboratory, 266237 Qingdao, China
[3]Institute for Advanced Ocean Study, Ocean University of China, 266100 Qingdao, China
[4]Institut für Physik der Atmosphäre, Deutsches Zentrum für Luft- und Raumfahrt e.V. (DLR), 82234 Oberpfaffenhofen, Germany
[5]Leibniz Institute for Tropospheric Research (TROPOS), 04318 Leipzig, Germany
[6]Laboratory of Space Laser Engineering, Shanghai Institute of Optics and Fine Mechanics, Chinese Academy of Sciences, 201800 Shanghai, China
[7]Physical Oceanography Laboratory, Ocean University of China, 266100 Qingdao, China

*Correspondence to*: Guangyao Dai (daiguangyao@ouc.edu.cn)

**Abstract.** By utilizing Level 2A products (particle optical properties and numerical weather prediction data) and Level 2C products (numerical weather prediction wind vector assimilated with observed wind component) provided by the Atmospheric Laser Doppler Instrument (ALADIN) onboard the Aeolus mission, and Level 2 vertical feature mask (VFM) products provided by Cloud-Aerosol Lidar with Orthogonal Polarization (CALIOP) onboard Cloud-Aerosol Lidar and Infrared Pathfinder Satellite Observation (CALIPSO) mission, three remote ocean areas are selected and the optical properties at 355 nm of marine aerosol are derived. The combined analysis of marine aerosol optical properties at 355 nm and instantaneous co-located wind speeds above the remote ocean areas are conducted. Eventually their relationships are explored and discussed at two sperate vertical atmospheric layers (0-1 km and 1-2 km, correspond to the heights within and above marine atmospheric boundary layer (MABL)), revealing the marine aerosol related atmospheric background states. Pure marine aerosol optical properties at 355 nm are obtained after quality control, cloud screening and backscatter coefficient correction from the ALADIN observations. The spatial distributions of marine aerosol optical properties and wind speed above the study areas are presented and analysed, respectively, at two vertical layers. The statistical results of the marine aerosol optical properties along with the wind speed grids at two vertical layers together with the corresponding regression curves fitted by power law functions are acquired and analysed, for each remote ocean area. The optical properties present increasing trends with wind speed in all cases, implying that the atmosphere of the two vertical layers will both receive the marine aerosol input produced and transported by the wind and the turbulence. The marine aerosol enhancement caused by the wind speed at the lower layer is more intensive than at the higher layer. As derived data from ALADIN, the averaged marine aerosol optical depth ( $\mathrm{AOD_{mar}}$ ) and the averaged marine aerosol lidar ratio ( $\mathrm{LR_{mar}}$ ) at 355 nm are acquired and discussed along the wind speed range. The



marine aerosol optical properties distributions, wind speed bins, and the marine aerosol variation tendencies along wind speed above the individual study areas are not totally similar, implying that the development and evolution of the marine aerosol above the ocean might not only be dominated by the drive of the wind, but also be impacted by other meteorological and environmental factors, e.g., atmospheric stability, sea and air temperature, or relative humidity. Combined analysis on the aerosol optical properties and wind with additional atmospheric parameters above the ocean might be capable to provide more

detailed information of marine aerosol production, entrainment, transport and removal.

## 1 Introduction

The global ocean is the largest source of natural aerosol. According to the Intergovernmental Panel on Climate Change (IPCC) Fifth Assessment Report, the total emission of marine aerosol produced from ocean is 1400 to 6800 $Tg \cdot yr^{-1}$ (Boucher et al., 2013). Accurate estimation of marine aerosol production, evolution and dissipation, and the knowledge of marine aerosol

spatial and temporal distribution are significate for studying the global energy budget, aerosol-cloud interactions and visibility changes (Latham and Smith, 1990; Murphy et al., 1998; O'Dowd et al., 1999; Haywood et al., 1999; de Leeuw et al., 2000; Kaufman et al., 2002; Smirnov et al., 2012). Radiative forcing caused by marine aerosol is an important component of the global energy budget. It was reported that the average marine aerosol optical depth ( $AOD_{mar}$ ) is approximately 0.15 while the volume concentration of cloud condensation nuclei from marine aerosol is around 60 $cm^{-3}$ (Kaufman et al., 2002; Lewis and

Schwartz, 2004). Therefore, marine aerosol has the direct impact and the indirect impact on radiative forcing, which are scattering and absorbing solar radiation, and converting cloud microphysical properties, respectively (Murphy et al., 1998; Pierce and Adams, 2006). The knowledge of the impact of the magnitude and changes of marine aerosol emissions on the shifts in climate and marine ecosystem processes is limited (IPCC, 2021).

Marine aerosols mainly include primary sea spray particles and secondary aerosols produced by the oxidation of emitted

precursors. Sea spray particles, composed of sea-salt and primary organic aerosols, are produced by wind induced wave breaking as well as the wind driving direct mechanical disruption of waves crests (O'Dowd and de Leeuw, 2007; IPCC, 2021). Moreover, as a dynamical meteorological factor, wind speed also has vital influence on the transport, evolution and dissipation of aerosols. Consequently, the wind speed is a crucial factor which governs the production and life cycle of marine aerosol (Lewis and Schwartz, 2004). Exploring the accurate relationship between aerosol optical depth (AOD) and wind speed is

significant for improving global aerosol transport models (Jaegle et al., 2011; Madry et al., 2011; Fan and Toon, 2011), for enhancing satellite-retrieved AODs (Kahn et al., 2010; Kleidman et al., 2012), for atmospheric correction of ocean color (Zibordi et al., 2011) and for the study of biogeochemical cycles (Meskhidze and Nenes, 2010). Several efforts have been reported to explore the relationship between the AOD or aerosol extinction coefficient over the ocean and wind speed. Utilizing either satellite-retrieved AODs (Glantz et al., 2009; Huang et al., 2010; Lehahn et al., 2010; O'Dowd et al., 2010; Grandey et

al., 2011) or surface (coast, island or ship)-based measurement AODs (Platt and Patterson, 1986; Villevalde et al., 1994;



Smirnov et al., 1995; Wilson and Forgan, 2002; Smirnov et al., 2003; Shinozuka et al., 2004; Mulcahy et al., 2008; Lehahn et al., 2010; Adames et al., 2011; Sayer et al., 2012; Smirnov et al., 2012), major previous researches focused on the AOD measured by passive instruments (mainly sun-photometer). From these studies, various power-law or linear relationships revealing positive correlation between AODs over the ocean and surface wind speed were established. The passive instruments

lack the abilities of distinguishing marine aerosol from other aerosols, acquiring vertical profiles of aerosols, and retrieving aerosol optical properties without sunlight (except for lunar-photometer) and under cloudy conditions (Kiliyanpilakkil and Meskhidze, 2011; Winker and Pelon, 2003). Active optical instruments for aerosol measurements, mainly like lidar, were also used in revealing the relationship between AOD/extinction coefficient of marine aerosol and wind speed. A shipborne depolarization lidar was occupied to acquire aerosol extinction coefficients over the East Sea of Korea near Busan and Pohang,

associated with the measurement of an anemometer mounted on a mast, finding a positive linear relationship ($R^2$ =0.57) between extinction (532 nm) at $300 \pm 50$ m and wind speed at 20 m (Shin et al., 2014). However, this relationship was established with data offshore thus it can not be representative for the global ocean. Cloud-Aerosol Lidar with Orthogonal Polarization (CALIOP) onboard Cloud-Aerosol Lidar and Infrared Pathfinder Satellite Observation (CALIPSO) mission is capable of measuring the global aerosol optical properties vertical distributions and recognizing aerosol types (include "clean

marine"). Kiliyanpilakkil and Meskhidze (2011) selected CALIOP-retrieved pure $AOD_{mar}$ below 2 km over ocean by utilizing the CALIOP aerosol subtype products and combined them with the surface wind speed provided by the Advanced Microwave Scanning Radiometer (AMSR-E) on board the Aqua satellite, acquiring a relatively complex increasing regression function, which will be presented and compared in Section 4.4.2 of this paper. Besides, Prijith et al. (2014) also made use of CALIOP-retrieved AODs below 0.5 km over ocean and the surface wind speed, obtaining nearly positive correlation linear relationships.

Nevertheless, the assumed lidar ratio (20 sr at 532nm) is used in the $AOD_{mar}$ retrieval process of CALIOP (Kiliyanpilakkil and Meskhidze, 2011), but the lidar ratio of marine aerosol can vary from 5 sr to more than 30 sr at 532 nm (Groß et al., 2013; Groß et al., 2015; Bohlmann et al., 2018), which could generate deviations during the retrieval of $AOD_{mar}$. In summary, to explore the accurate relationship between the marine aerosol optical properties and the wind speed, it is essential to conduct global continuous observations and obtain the information of aerosol type identification, while vertical profiles of aerosols can

provide extra spatial information for further analysis. Moreover, previous studies mostly focused on the layer $AOD_{mar}$ and ocean surface wind speed, exploring the probable production of marine aerosol driven by surface wind. The relationship between the vertical marine aerosol optical properties and corresponding spatiotemporally synchronous wind speed is still to be investigated, which represents the marine-atmospheric background state and may reveal the transport and evolution of the marine aerosol vertically.

Atmospheric Laser Doppler Instrument (ALADIN) is a first ever spaceborne direct detection wind lidar, as the single payload installed on the Aeolus mission from the European Space Agency (ESA), which was launched into space in August 2018 (Stoffelen et al., 2005; Reitebuch et al., 2012; Kanitz et al., 2019). As a direct detection high-spectral-resolution lidar, ALADIN has the capability in providing the global aerosol optical properties (e.g., extinction coefficient and backscatter coefficient)



profiles at 355 nm (Level 2A product), the horizontal-line-of-sight (HLOS) wind speed profiles (Level 2B product), and the
wind vector profile from the European Centre for Medium-Range Weather Forecasts (ECMWF) model along the Aeolus track
(Level 2C product) (Rennie et al., 2020). It should be emphasized that the aerosol and wind product are retrieved from the
backscattered signal of the same laser light pulse emitted from ALADIN to the atmosphere, hence the geolocation and time
information of these products is completely consistent for every profile. The detection altitude range of these products is from
the earth surface to around 20 km and the vertical resolutions varies from 0.25 km to 2 km (from bottom to top). Though
regarded as a by-product, the particle optical property products are still demonstrated to provide valuable information about
particles, especially on the detection and characterization of aerosol and cloud layers and on the lidar ratios (Baars et al., 2021;
Flament et al., 2021; Abril-Gago et al., 2022). It should be emphasized that the lowest altitude bins of Aeolus Level 2A and
Level 2B products could be contaminated by reflections from the land or ocean surface, and are thus not representative for the
atmospheric wind speed and the aerosol optical properties (Wu et al., 2022). Dai et al. (2022) conducted the first attempt on
the combined application of the aerosol products (Level 2A products) and the wind vector products (Level 2C products) of
ALADIN, observing an enormous dust transport event occurred in June 2020 from the Sahara to the Americas, describing the
transport quantitatively by calculating dust advection.

As mentioned above, Aeolus can provide global high spatial and temporal resolution aerosol optical properties profiles and
wind speed profiles despite the lack of the lowest bins close to the ground. Additionally, CALIOP can provide global aerosol
types information. Hence, the combination of Aeolus-CALIOP products is capable of analysing the relationship between the
marine aerosol optical properties (e.g., AOD, extinction coefficient) at 355 nm and wind speed globally and vertically. In this
paper, by utilizing Aeolus Level 2A, Level 2C products and CALIOP aerosol subtype products, we firstly 1) select ocean areas
far from land and examine the domination of marine aerosol over these areas with the CALIOP aerosol classification products,
and then 2) try to acquire the pure marine aerosol optical properties (extinction coefficient, backscatter coefficient, AOD, lidar
ratio) at 355 nm and the corresponding wind speeds from Aeolus products, and to analyse the spatial distributions of those
atmospheric state parameters at two separate vertical layers (ocean surface to 1 km, 1 km to 2 km, corresponding to the layers
within and above the marine atmospheric boundary layer (MABL), respectively), and finally 3) explore the relationship
between the marine aerosol optical properties and the wind speeds vertically above ocean. Generally, the highlights of this
work mainly include 1) first ever deriving pure marine aerosol optical properties from Aeolus products 2) acquiring the
spatiotemporally synchronous relationship with the aerosol optical properties and the instantaneous wind speeds, which could
indicate the background atmosphere states within and above the MABL over remote ocean, 3) conducting analysis at two
separate height layers above ocean surface to explore the vertical differences, and 4) selecting low latitude regions and middle
latitude regions in the Southern Hemisphere and Northern Hemisphere as study areas respectively.

The paper is organized as follows: section 2 introduces the spaceborne lidars and their specific products used in this study;
section 3 provides the methodology of study areas selection, data pre-processing and data analyses for relationship exploration
between marine aerosol optical properties and wind speed; section 4 presents the procedure of study areas selection,  then
analyses and discusses the marine aerosol optical properties, the wind speed, and their relationship above three selected areas.



## 2 Spaceborne lidars and products

### 2.1 ALADIN/Aeolus

Since its launch in August 2018, ALADIN, has been globally observing the profiles of the component of the wind vector along the laser's line of sight (LOS), and the profiles of aerosol optical properties, for more than four years. Aeolus flies at a mean altitude of about 320 km in a sun-synchronous orbit with the local equatorial crossing time of about 06:00 and 18:00, a daily quasi-global coverage (about 16 orbits per day) with an orbit repeat cycle of 1 week (111 orbits) (Reitebuch, 2012). Designed as a high-spectral-resolution lidar with a laser wavelength of 354.8 nm, ALADIN has the ability to acquire wind profiles and

particle optical properties simultaneously with its two separate optical frequency discrimination channels named as Rayleigh channel and Mie channel. The detailed descriptions of the instrument design and the measurement concept are introduced in, e.g., Ansmann et al. (2007), Dabas et al. (2008), Flamant et al. (2008), Reitebuch (2012), Lux et al. (2020) and Flament et al. (2021).

Processed in different phases, Aeolus data products are classified at several levels: Level 0 (instrument housekeeping data),

Level 1B (engineering-corrected HLOS winds), Level 2A (aerosol and cloud layer optical properties), Level 2B (meteorologically representative HLOS winds) and Level 2C (Aeolus-assisted wind vectors) (Flamant et al., 2008; Tan et al.,2008; Rennie et al., 2020). It should be emphasized that Level 2C wind vectors are the outputs from the assimilation of the Aeolus Level 2B products in the ECMWF numerical weather prediction (NWP) operational model after 9 January 2022, where Aeolus was used operationally by ECMWF (Rennie et al., 2021)). In addition, the products of Aeolus are available into

different Baselines which correspond to different processor versions used to derive the products. The products were firstly released as Baseline 07 at the beginning and updated to Baseline 14 until this study is conducted. As mentioned above, we use Level 2A and Level 2C products of Aeolus for the study of the relationship between marine aerosol optical properties and wind speeds. As the because Level 2C products can provide both components of the wind vector that we use Level 2C instead of Level 2B products of Aeolus. The time coverage of Aeolus products used in this study is from 20 April 2020 to 4 July 2022.

Thus, in the aspect of the utilized Level 2A products, the data processors are Baseline 11 (20 April 2020 to 26 May 2021), Baseline 12 (26 May 2021 to 6 December 2021), Baseline 13 (6 December 2021 to 29 March 2022) and Baseline 14 (29 March 2022 to 4 July 2022), while as for the Level 2C products, the data processors are Baseline 09 (20 April 2020 to 9 July 2020), Baseline 10 (9 July 2020 to 8 October 2020), Baseline 11 (8 October 2020 to 26 May 2021), Baseline 12 (26 May 2021 to 6 December 2021), Baseline 13 (6 December 2021 to 29 March 2022) and Baseline 14 (29 March 2022 to 4 July 2022),

respectively (https://aeolus-ds.eo.esa.int/oads/access/, last access: 16 February 2023). The Level 2C NWP wind vector products from ECMWF used in this study are obtained after assimilation of the Level 2B observed HLOS wind products.

### 2.2 CALIOP/CALIPSO

CALIOP, one of the payloads installed on CALIPSO, has been measuring global vertical aerosol and cloud optical properties profiles for more than 16 years since 2006. It can provide backscatter coefficient at 532 nm and 1064 nm, extinction coefficient



at 532 nm and 1064 nm, depolarization ratio at 532 nm, vertical feature mask (VFM) products and so on (Winker et al., 2009). The VFM products comprise the vertical information along every profile on the identification of clouds and aerosols, and further, on the subtype classification of clouds and aerosols. For cloud and aerosols identification, the cloud-aerosol discrimination (CAD) algorithm is applied based on layer averages of attenuated backscatter at 532 nm, attenuated total color ratio and the mid-layer altitude (Liu et al., 2019). The aerosol sub-types are distinguished as "marine", "dusty marine", "dust",

"polluted dust", "continental", "polluted continental", "elevated smoke" and "others" via the joint analysis of depolarization ratio, integrated attenuated backscatter coefficient and surface type (Kim et al., 2018). In this study, CALIOP Level (L2) VFM products are applied to confirm the domination of the marine aerosol over the selected ocean areas. Different versions of the CALIOP L2 VFM product are used, respectively, the versions are 4.10 (20 April 2020 to 1 July 2020), 4.20 (1 July 2020 to 19 January 2022) and 3.41 (19 January 2022 to 4 July 2022).




## 3 Methodology

In general, the data processing and analysis procedure of this study can be concluded briefly as three parts including selection of the study areas, data pre-processing and data analyses, respectively, as shown in Fig. 1.

**Step 1: Selection of study areas with CALIOP**

Select Ocean areas far from land
Reduce the influence of terrestrial aerosols

**CALIOP VFM products**
Statistical analysis of aerosol types in the selected area

Marine aerosol dominates? — No → stop

Yes

**Step 2: Data processing of Aeolus products**

Level 2A product
- **Extinction coefficient at 355 nm**
- **Backscatter coefficient at 355 nm**

NWP model parameters from Level 2A product
- Relative humidity (RH)
- Molecular backscatter coefficient

NWP model winds from Level 2C product
- U component of wind vectors
- V component of wind vectors

**Quality control**
- Valid data selection with QC flags
- Outliers elimination with Tukey's test

**Cloud screening**
Retain data with: RH < 94%; backscatter ratio < 2.5

**Backscatter coefficient correction**
Correct with: Depolarization ratio = 2% of marine aerosol

Wind speed

**Step 3: Data analyses**

**Optical properties and wind speeds distribution analyses**
At two sperate vertical layers (ocean surface to 1 km; 1 km to 2 km) over selected ocean areas

**Correlation analysis between optical properties and wind speed**
- Averaging of optical properties along the wind speed grid of 1 m·s⁻¹
- Parametric curve fitting of the optical mean properties vs. wind speed

**Derived aerosol optical properties analysis with wind speed**
- Aerosol optical depth vs. wind speed
- Lidar ratio vs. wind speed

**Figure 1: Flowchart of the study methodology**



Firstly, this work mainly focuses on the marine aerosol, hence the ocean areas for the study are supposed to be far away from land to reduce the influence of terrestrial aerosols. The aerosol classification information from CALIOP VFM products are utilized to statistically analyse the aerosol types of the selected areas. It is found that the marine aerosols are mostly distributed at the altitude range of 0 km to 2 km during the VFM processing. Therefore, the statistical analysis of aerosol types is conducted

at the same altitude range. It is considered that the marine aerosol dominates in the selected area if the percentage of aerosol subtype "marine" is larger than 75% meanwhile the percentage sum of "marine" and "dusty marine" is above 90%, then the study can be continued for this area.

Extinction coefficient at 355 nm and backscatter coefficient at 355 nm retrieved by the standard correction algorithm (SCA) from Aeolus Level 2A product are used in this study, as the SCA processing is capable to produce more stable extinction

coefficient and backscatter coefficient than other algorithms (Flament et al., 2021). Furthermore, the mid-bin product (sca_optical_properties_mid_bins) of the SCA product are chosen as a result from that the product retrieved as the mid-bin algorithm is more robust (Baars et al., 2021; Flament et al., 2021). To ensure a high data quality and hence to acquire the relationship between the optical properties and wind speed, a rigorous quality control has to be applied. In the aspect of quality control, negative extinction coefficients and backscatter coefficients are excluded, and then the quality flags ("bin_1_clear"

and "processing_qc_flag") provided in the Level 2A product are applied to filter out invalid data (Trapon et al., 2022). Additionally, the outliers are eliminated by the method of Tukey's test. By using the lower quartiles $Q_L$ (25% positions of the data) and upper quartiles $Q_U$ (75% positions of the data), this method classifies the data below $Q_L - 3 \cdot (Q_U - Q_L)$ or above $Q_L + 3 \cdot (Q_U - Q_L)$ as outliers (Hoaglin et al., 1986). As the Aeolus products do not distinguish the aerosol layers and cloud layers, the marine aerosol optical properties may be contaminated by the cloud layers. The relative humidity (RH) and

molecular backscatter coefficient of each data bin from the NWP model of ECMWF are provided in the Level 2A product and are utilized to screen the cloud layers. It is considered that a cloud is quite likely to exist if the RH is larger than 94% or the backscatter ratio (BR) (total backscatter coefficient/molecular backscatter coefficient) at 355 nm is larger than 2.5 (Flamant et al., 2020). Therefore, in this study, when the RH is higher than 94% or the BR is larger than 2.5, the corresponding data bin is regarded as cloud contaminated and is eliminated. Due to the instrument design of ALADIN, it can only detect the co-polar

backscatter light, leading to the lack of the depolarized portion of the backscatter coefficient (Flamant et al., 2020). According to Groß et al. (2015), the depolarization ratio at 355 nm of marine aerosol ($\delta_{mar,355}$) is approximately 0.02 when the RH is larger than 50%. Nevertheless, dried marine aerosol layers can significantly depolarize and the depolarization ratio will vary from 0.02 to around 0.1, so the typical $\delta_{mar,355}$ of humid marine aerosol (RH>50%) is not suitable for dried aerosol (Haarig et al. 2017; Bohlmann et al. 2018). Consequently, to correct the marine aerosol backscatter coefficient with the typical $\delta_{mar,355}$

of humid marine aerosol, the data with RH>50% is retained (around 95% data is retained), and thus with the typical $\delta_{mar,355}$ the total marine aerosol backscatter coefficient $\beta_{mar}$ can be calculated by the following Eq. (1):



$$\beta_{mar} = (1 + \delta_{mar,355}) \cdot \beta_{mar,Aeolus}, \tag{1}$$

where $\beta_{mar,Aeolus}$ is the original marine aerosol backscatter coefficient measured by ALADIN. As for the wind vector data, Aeolus Level 2C product provides the $u$ component (zonal components of wind vector) and $v$ component (meridional components of wind vector) from the ECMWF model at the same data bins of the Level 2A optical properties product. Hence the wind speed $ws$ can be calculated with these two components by the following Eq. (2):

$$ws = \sqrt{u^2 + v^2}. \tag{2}$$

With the re-processed marine aerosol optical properties extinction coefficient $\alpha_{mar}$ and $\beta_{mar}$, and the corresponding $ws$, it is possible to explore the relationship between these parameters. At the beginning of data analyses, $\alpha_{mar}$, $\beta_{mar}$ and $ws$ within the altitude range of 0 km to 2 km are selected, where the marine aerosol dominates according to the analysis of CALIOP VFM. Further, the whole study height range is divided into two individual layers. Considering that the MABL height of the remote ocean is around 1 km (Luo et al., 2014; Luo et al., 2016; Alexander et al., 2019), the boundary height of the two vertical layers is set as 1 km in this study to explore the difference between within the MABL and above the MABL. Among them one is called $\text{Layer}_L$ in this paper, which corresponds to the MABL with the altitude range of 0 km to 1 km, and another is called $\text{Layer}_H$, above the MABL with the altitude range of 1 km to 2 km. It should be emphasized that in $\text{Layer}_L$, the lowest Aeolus Level 2A products (particle optical properties) data bins with the altitude of lower than about 0.25 km, are absent to avoid the ground return signals' contamination. This leads to that the actual marine aerosol optical properties altitude range of $\text{Layer}_L$ is around 0.25 km to 1 km. Over the selected ocean areas, the spatial distribution of the $\alpha_{mar}$, $\beta_{mar}$ and $ws$ are acquired with the longitude-latitude grid of $5° \times 5°$ at two separate layers. Then the relationship analyses between the optical properties ($\alpha_{mar}$, $\beta_{mar}$) and $ws$ of these two layers are conducted by the average calculations of the optical properties along $ws$ grids (1 $\text{m} \cdot \text{s}^{-1}$) and the parametric curve fitting. The derived data of $\alpha_{mar}$, $\beta_{mar}$, averaged marine aerosol optical depth ($\text{AOD}_{mar}$) and marine aerosol lidar ratio ($\text{LR}_{mar}$) are obtained and discussed, as well. The $\text{AOD}_{mar}$ is acquired by integrating Aeolus retrieved $\alpha_{mar}$ within 2 km of every single profile. The $\text{AOD}_{mar}$ is calculated within the height of 2 km in order to compare with the previous result of CALIOP, where the integration height is the same as that in this study. The averaged $\text{AOD}_{mar}$ along the $ws$ grid are obtained and then are compared with the $\text{AOD}_{mar}$-$ws$ relationships from a previous study in Section 4.4.1. The $\text{LR}_{mar}$ are derived via dividing $\alpha_{mar}$ by $\beta_{mar}$ for each corresponding data bin. The spatial distribution of the $\text{LR}_{mar}$ are presented in Section 4.2, meanwhile the relationship between the variations of the $\text{LR}_{mar}$ along $ws$ grids and the marine aerosol particle size are discussed in Section 4.4.2.



# 4 Results and discussion

## 4.1 Study areas selection and aerosol types analysis

To focus this study only on marine aerosol, the study areas should be far away from continents and rarely affected by aerosol from land surfaces, e.g. anthropogenic, dust, biomass burning. The selected ocean areas are located in the North Pacific ocean, South Pacific ocean, South Indian ocean, with the latitude and longitude range of $0°$ to $30°$N and $150°$E to $180°$ to $150°$W, $20°$S to $60°$S and $100°$W to $150°$W, $20°$S to $60°$S and $60°$E to $90°$E, respectively, as shown in Fig. 2. Hence, we call these three remote ocean areas "the NP area", "the SP area" and "the SI area" in this paper, respectively.

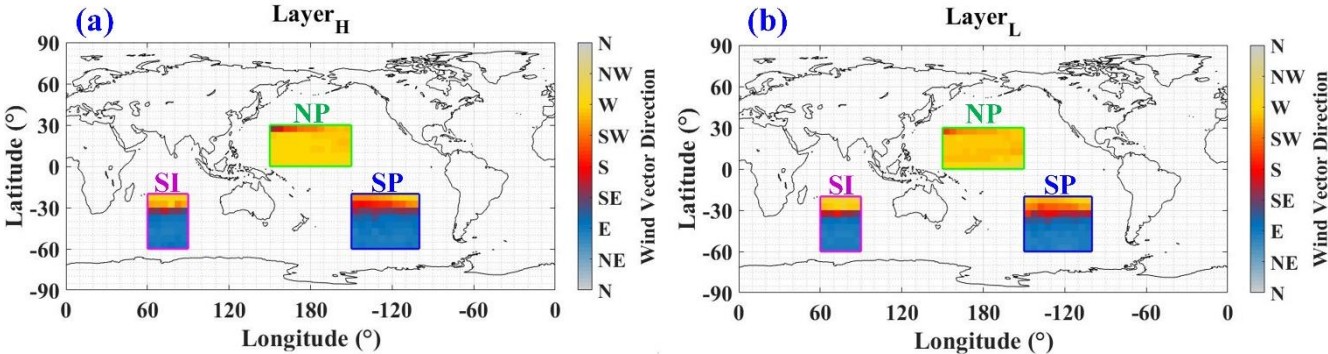

**Figure 2: The selected study areas and their wind vector direction distributions above (a) $Layer_H$ (1-2 km, above the MABL) and (b) $Layer_L$ (0-1 km, correspond to the MABL). All the three study areas are far away from the lands. The colors in the boxes indicate the wind vector directions.**

Wind is the major driver for aerosol transport. Hence, firstly the $5°×5°$ grid averaged wind directions of $Layer_H$ and $Layer_L$ are calculated and presented in Fig. 2. From the distributions, it can be seen that the wind directions of $Layer_H$ and $Layer_L$ are quite similar to each other. The NP area is dominated by westward wind with a small proportion of southward wind in the northwest of this area while the eastern side of this area is far away from land. As a result, it is inferred that the terrestrial aerosol is rarely going to be transported to the NP area. In terms of the SI area and the SP area, of which the wind direction distributions are similar, from south to north, the wind directions vary from eastward to southward and then to westward. Moreover, there is no continent or the continent is far away in the upwind of the SI area and the SP area. Hence, it indicates that these two areas are rarely influenced by other aerosols from lands as well.

To further verify the domination of marine aerosol, as introduced in Section 3 of this paper, the CALIOP VFM aerosol classification products are applied. The proportions of eight aerosol types (marine, dusty marine, dust, polluted dust, continental, polluted continental, smoke and others) are counted at two vertical layers defined in Section 3 over the NP area, the SP area and the SI area, respectively, as shown the histograms in Fig. 3. The proportions of marine aerosol at $Layer_L$ in these three separate areas are 87%, 84% and 84% while the proportions at $Layer_H$ are 84%, 79% and 79% respectively, which are all larger than 75%. Moreover, the percentage sums of marine aerosol and dusty marine aerosol are all above 90%, at both





layers and over all study areas. Consequently, the selected areas NP, SP and SI can be regarded as the marine aerosol
dominating areas.

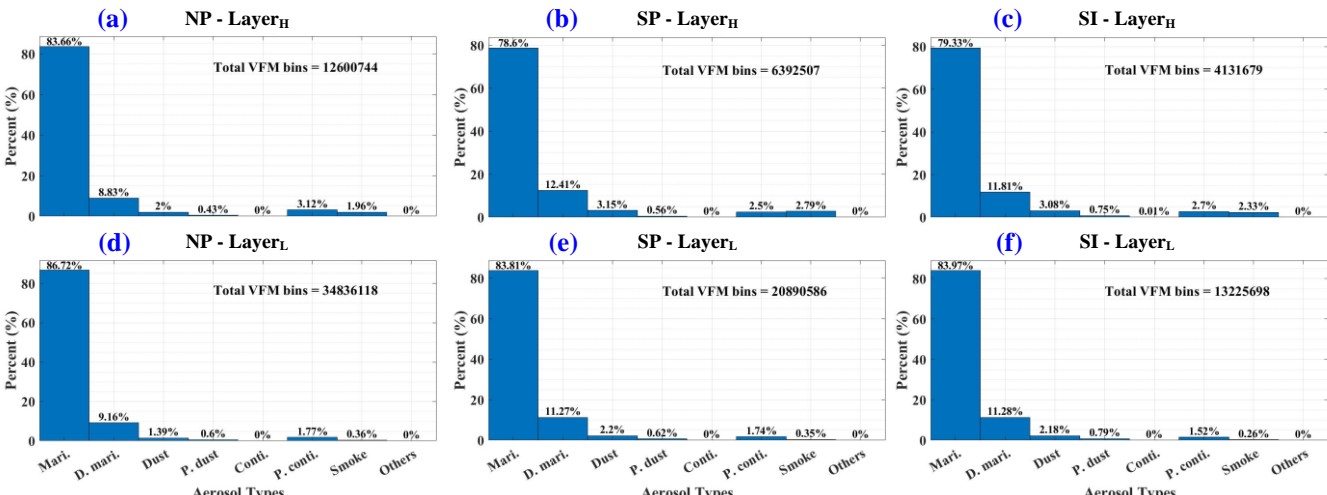

Figure 3: Aerosol types statistical analyses over (a)(d) the NP area, (b)(e) the SP area and (c)(f) the SI area at two sperate layers.

In this section, the study areas in this paper are introduced. With the wind direction distribution analyses and the aerosol types
statistical analyses, the dominations of marine aerosol are confirmed over these three areas. It should be illustrated that among
the areas, the NP area is mainly located in low latitudes or tropics, while the SP area and the SI area are in mid-latitude region.

## 4.2 Spatial distribution of wind speed and aerosol optical properties

With Aeolus L2A products (particle optical properties) and L2C products (ECMWF model winds) from April 2020 to July
2022, calculated for every $5° \times 5°$ grid, the averaged $ws$, $\alpha_{mar}$, $\beta_{mar}$ and $LR_{mar}$ spatial distributions of $Layer_H$ and $Layer_L$
are acquired. The averaged atmospheric parameters spatial distributions of the NP area, the SP area and the SI area are
presented in Fig. 4, Fig. 5 and Fig. 6, respectively. These figures describe the atmospheric background state of optical properties
and wind speed within ($Layer_L$) and above ($Layer_H$) the MABL over the study areas. Additionally, the mean values and the
standard deviations of these atmospheric parameters at $Layer_H$ and $Layer_L$ are calculated for each study area by averaging
the spatial distributions of $5° \times 5°$ grid, and are presented in Fig. 7.



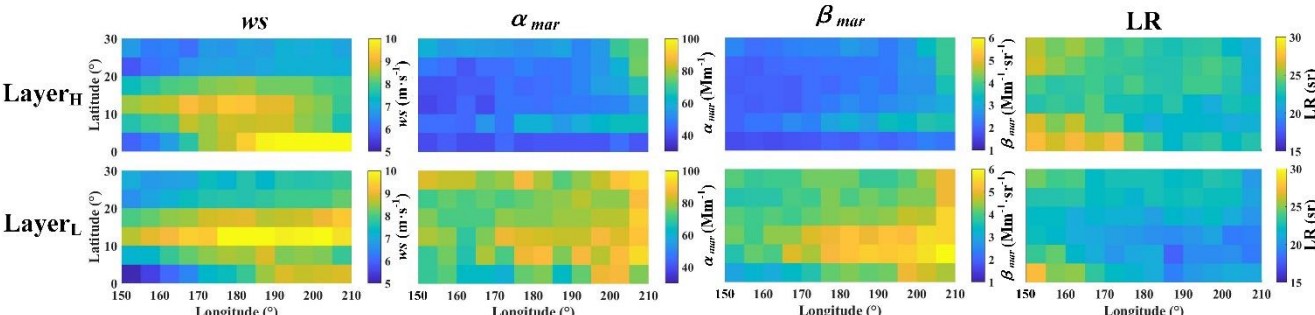

**Figure 4: Wind speed ( *ws* ), marine aerosol extinction coefficient ( $\alpha_{mar}$ ), marine aerosol backscatter coefficient ( $\beta_{mar}$ ), and marine aerosol lidar ratio ( $\mathrm{LR}_{mar}$ ) spatial distributions above the North Pacific (NP) area at** $\mathrm{Layer_H}$ **and** $\mathrm{Layer_L}$ **.**

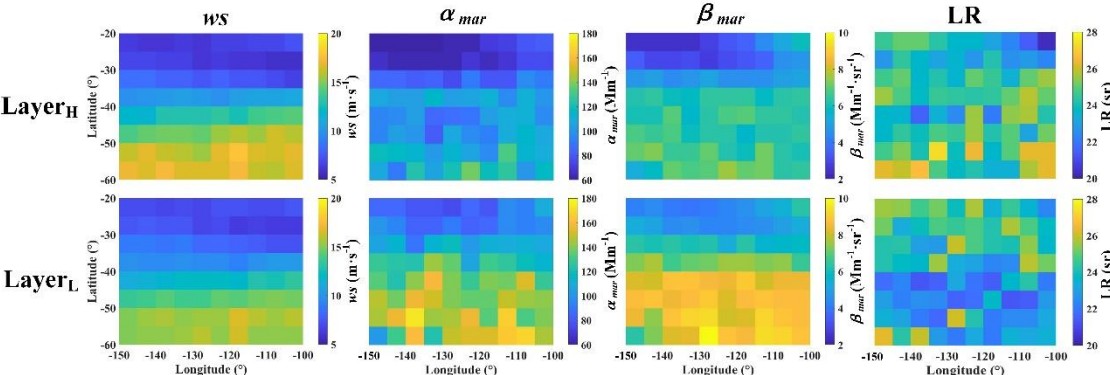

**Figure 5: Wind speed ( *ws* ), marine aerosol extinction coefficient ( $\alpha_{mar}$ ), marine aerosol backscatter coefficient ( $\beta_{mar}$ ), and marine aerosol lidar ratio ( $\mathrm{LR}_{mar}$ ) spatial distributions above the South Pacific (SP) area at** $\mathrm{Layer_H}$ **and** $\mathrm{Layer_L}$ **.**

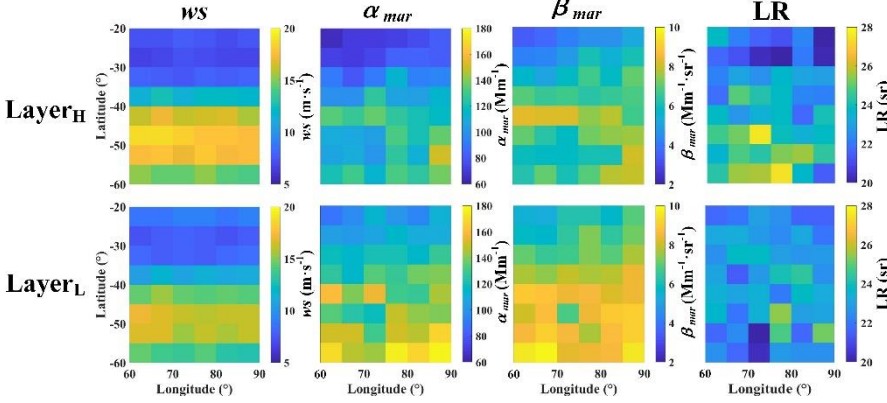

**Figure 6: Wind speed ( *ws* ), marine aerosol extinction coefficient ( $\alpha_{mar}$ ), marine aerosol backscatter coefficient ( $\beta_{mar}$ ), and lidar ratio ( $\mathrm{LR}_{mar}$ ) spatial distributions above the South Indian (SI) area at** $\mathrm{Layer_H}$ **and** $\mathrm{Layer_L}$ **.**



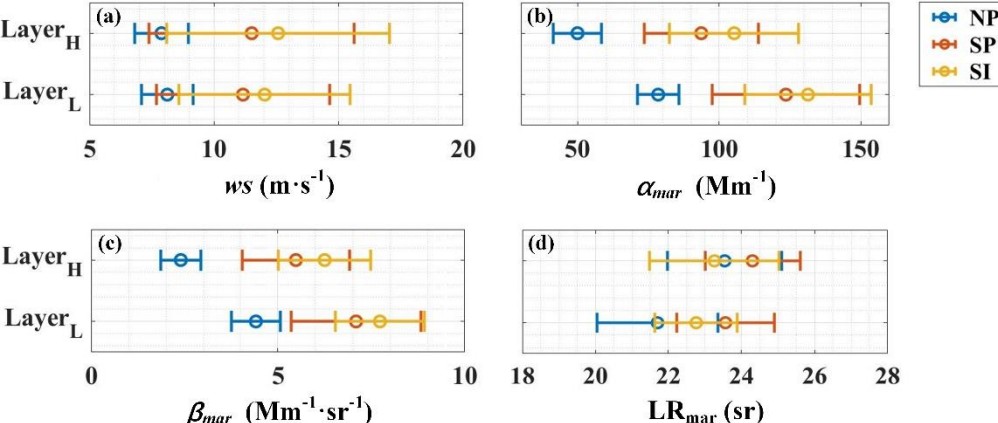

**Figure 7: Mean values at** $\text{Layer}_H$ **and** $\text{Layer}_L$ **of (a) wind speed (** $ws$ **), (b) marine aerosol extinction coefficient (** $\alpha_{mar}$ **), (c) marine aerosol backscatter coefficient (** $\beta_{mar}$ **), and (d) marine aerosol lidar ratio (** $\text{LR}_{mar}$ **) above the North Pacific (NP) area (blue standard deviation bars), the South Pacific (SP) area (red standard deviation bars), and the South Indian (SI) area (yellow standard deviation bars).**

In Fig. 7, the mean values and the deviations of the parameters represent the background atmospheric states within the MABL ( $\text{Layer}_L$ ) and over the MABL ( $\text{Layer}_H$ ) above each study areas. The averaged $ws$ are $8.1\pm1.0\,\text{m}\cdot\text{s}^{-1}$, $11.2\pm3.5\,\text{m}\cdot\text{s}^{-1}$, $12.0\pm3.4\,\text{m}\cdot\text{s}^{-1}$ at $\text{Layer}_L$, while $7.9\pm1.1\,\text{m}\cdot\text{s}^{-1}$, $11.5\pm4.1\,\text{m}\cdot\text{s}^{-1}$, $12.6\pm4.5\,\text{m}\cdot\text{s}^{-1}$ at $\text{Layer}_H$, above the NP area, the SP area, and the SI area, respectively. The averaged $\alpha_{mar}$ are $78\pm7\,\text{Mm}^{-1}$, $124\pm26\,\text{Mm}^{-1}$, $131\pm22\,\text{Mm}^{-1}$ at $\text{Layer}_L$, while $50\pm9\,\text{Mm}^{-1}$, $94\pm20\,\text{Mm}^{-1}$, $105\pm23\,\text{Mm}^{-1}$ at $\text{Layer}_H$, above the NP area, the SP area, and the SI area, respectively. The averaged $\beta_{mar}$ are $4.4\pm0.7\,\text{Mm}^{-1}\cdot\text{sr}^{-1}$, $7.1\pm1.7\,\text{Mm}^{-1}\cdot\text{sr}^{-1}$, $7.7\pm1.2\,\text{Mm}^{-1}\cdot\text{sr}^{-1}$ at $\text{Layer}_L$, while $2.4\pm0.5\,\text{Mm}^{-1}\cdot\text{sr}^{-1}$, $5.5\pm1.4\,\text{Mm}^{-1}\cdot\text{sr}^{-1}$, $6.3\pm1.2\,\text{Mm}^{-1}\cdot\text{sr}^{-1}$ at $\text{Layer}_H$, above the NP area, the SP area, and the SI area, respectively. The averaged $\text{LR}_{mar}$ are $21.7\pm1.7\,\text{sr}$, $23.6\pm1.3\,\text{sr}$, $22.8\pm1.1\,\text{sr}$ at $\text{Layer}_L$, while $23.5\pm1.6\,\text{sr}$, $24.3\pm1.3\,\text{sr}$, $23.3\pm1.8\,\text{sr}$ at $\text{Layer}_H$, above the NP area, the SP area, and the SI area, respectively. It is reported that the typical ranges of $\alpha_{mar}$ and $\beta_{mar}$ at 532 nm over remote ocean areas are around $60\,\text{Mm}^{-1}$ to $80\,\text{Mm}^{-1}$ and around $1\,\text{Mm}^{-1}\cdot\text{sr}^{-1}$ to $5\,\text{Mm}^{-1}\cdot\text{sr}^{-1}$, respectively, observed and retrieved by CALIOP (Prijith et al., 2014; Kiliyanpilakkil and Meskhidze, 2011). Applying the typical $\alpha_{mar}$ Ångström exponent from 532 nm to 355 nm of $0.7\pm1.3$ and the typical $\beta_{mar}$ Ångström exponent from 532 nm to 355 nm of $0.8\pm0.1$ (Floutsi et al., 2022), the converted typical ranges of $\alpha_{mar}$ and $\beta_{mar}$ at 355 nm can be calculated, which are around $47\,\text{Mm}^{-1}$ to $180\,\text{Mm}^{-1}$ and around $1.3\,\text{Mm}^{-1}\cdot\text{sr}^{-1}$ to $7.2\,\text{Mm}^{-1}\cdot\text{sr}^{-1}$. Compared with the CALIOP retrieved and converted marine aerosol optical properties ( $\alpha_{mar}$ and $\beta_{mar}$ ) range, it is considered that the Aeolus retrieved $\alpha_{mar}$ and $\beta_{mar}$ are reasonable. The mean values of $ws$, $\alpha_{mar}$ and $\beta_{mar}$ above the NP area are the lowest among the three areas, both at $\text{Layer}_H$



and $\text{Layer}_\text{L}$, which may be because that this area is located in low latitudes region of the Northern Hemisphere. The highest

mean wind speed of the SI area corresponds to the highest $\alpha_{mar}$ and $\beta_{mar}$. The mean wind speeds of $\text{Layer}_\text{H}$ are both larger than those of $\text{Layer}_\text{L}$ in the SP area and in the SI area, while the phenomenon is on the contrary in the NP area. It is worth noting that in all the study areas, the averaged $\alpha_{mar}$ and $\beta_{mar}$ at $\text{Layer}_\text{L}$ are larger than those at $\text{Layer}_\text{H}$, illustrating that the majority of the aerosol from ocean is trapped in the MABL while a fraction of marine aerosol can be elevated above the MABL. In the aspect of mean $\text{LR}_{mar}$, the values at $\text{Layer}_\text{H}$ are all higher than at $\text{Layer}_\text{L}$, and all the values are in a reasonable range

referring to Bohlmann et al. (2018), Groß et al. (2011), Groß et al. (2015) and Floutsi et al. (2022).

From Fig.4 to Fig. 6, overall, it can be found that the parameters distributions of all three areas have several similar features. Primarily, the spatial variations of $ws$, $\alpha_{mar}$, $\beta_{mar}$ are more apparent along the meridian than zonally, both at $\text{Layer}_\text{H}$ and at $\text{Layer}_\text{L}$. In the aspect of $\text{Layer}_\text{L}$, there are separate distinct high wind speed regions or belts along latitude in the three areas, which are $5°\text{N}$ to $20°\text{N}$ region of the NP area with the wind speed bins of approximately $8 \text{ m·s}^{-1}$ to more than $10 \text{ m·s}^{-1}$, $40$

$°\text{S}$ to $60°\text{S}$ region of the SP area with the wind speed bins of more than $10 \text{ m·s}^{-1}$ to approximately $17 \text{ m·s}^{-1}$, and $35°\text{S}$ to $60°\text{S}$ region of the NI area with the wind speed bins of more than $10 \text{ m·s}^{-1}$ to approximately $17 \text{ m·s}^{-1}$ as well. Inspection of marine aerosol optical properties, $\alpha_{mar}$ and $\beta_{mar}$ in the high wind speed regions are obviously larger than in other regions. Hence, it can be inferred that, in the MABL, the wind speed and the marine aerosol optical properties tend to be positively correlated. Referring to $\text{Layer}_\text{H}$, the spatial variation trends of $ws$, $\alpha_{mar}$, $\beta_{mar}$ in the three areas are alike with those at $\text{Layer}_\text{L}$.

The evident high wind speed regions also exist at $\text{Layer}_\text{H}$ while $\alpha_{mar}$ and $\beta_{mar}$ are slightly enhanced in these regions, which indicates that the wind speed may still have weak positive influence on the marine aerosol optical properties at the higher atmosphere layer above the MABL. Some spatial distribution differences of $ws$, $\alpha_{mar}$, $\beta_{mar}$ between the three areas can be discovered as well. As for the SP area and the SI area, $ws$, $\alpha_{mar}$, $\beta_{mar}$ all mainly present increasing tendencies from north to south. In term of the NP area, besides the obvious enhancements of $ws$, $\alpha_{mar}$, $\beta_{mar}$ in the high wind speed belt, the gradual

enhancements of these atmospheric parameters are presented from west to east in this area.

At both layers of the NP area and at $\text{Layer}_\text{L}$ of the SP area, the $\text{LR}_{mar}$ turn out lower in the relatively high wind speed regions, which illustrates a possible negative correlation between $\text{LR}_{mar}$ and wind speed. The relationship between these two parameters is analysed and discussed in detail in Section 4.4.2 of this paper.

To conclude, this section presents the atmospheric background state of optical properties and wind speed, and analyses the

spatial distributions of $ws$, $\alpha_{mar}$, $\beta_{mar}$ jointly at $\text{Layer}_\text{H}$ and $\text{Layer}_\text{L}$ above the NP area, the SP area and the SI area, respectively. The $\alpha_{mar}$, $\beta_{mar}$ retrieved from Aeolus Level 2A products are in reasonable agreement with CALIOP and the Aeolus-derived $\text{LR}_{mar}$ are also reasonable. It is found that, both at $\text{Layer}_\text{H}$ and at $\text{Layer}_\text{L}$, spatially, the wind speed and $\alpha_{mar}$,



$\beta_{mar}$ show positive correlation though the optical properties at $\text{Layer}_L$ are greater than those at $\text{Layer}_H$, indicating that both layers receive the input of the aerosol produced from ocean by the wind but the majority of the marine aerosol are trapped in

the MABL while only a small fraction can be elevated into the higher layer. In addition, as the three study areas are located in different regions, the spatial distributions of $ws$, $\alpha_{mar}$, $\beta_{mar}$ are different.

## 4.3 Relationship between marine aerosol optical properties and wind speed

The distributions of marine aerosol optical properties versus wind speed are discussed in this section.. The data are counted with $\alpha_{mar}$ - $ws$ grid bins and $\beta_{mar}$ - $ws$ grid bins at two separate layers ($\text{Layer}_H$ and $\text{Layer}_L$) for the study areas. The statistical

results over the NP area, the SP area and the SI area are presented in Fig. 8, Fig. 9 and Fig. 10, respectively. Though the data counts sums of each study area are different due to the difference of the areas' dimensions, the color variation of the data bins can still reveal the normalized distribution. From Fig. 8 to Fig. 10, it can be figured out that the distributions of the optical properties and the wind speeds in the NP area are the most concentrated among three areas while the data in the SI area are quite dispersed. As for the NP area, few data bins appear in the region where wind speeds are greater than 20 $\text{m} \cdot \text{s}^{-1}$, indicating

that wind speed can rarely reach up to 20 $\text{m} \cdot \text{s}^{-1}$ at the altitude range of 0 km to 2 km in this low-latitude area. Nevertheless, in terms of the SP area and the SI area, though the major distribution of the wind speed at $\text{Layer}_H$ and $\text{Layer}_L$ are in the approximate range of 0 $\text{m} \cdot \text{s}^{-1}$ to 20 $\text{m} \cdot \text{s}^{-1}$, the wind speed is reaching up to 30 $\text{m} \cdot \text{s}^{-1}$ as the bins with the wind speed of 20 $\text{m} \cdot \text{s}^{-1}$ to 30 $\text{m} \cdot \text{s}^{-1}$ are almost all filled by blue. Between the two separate layers, these figures show no pronounced difference in the aspect of wind speed bins, whereas the optical properties at the lower layers are distinctly larger than those at the higher

layers. The data distributions of $\alpha_{mar}$ and $\beta_{mar}$ versus $ws$ at $\text{Layer}_L$ of three study areas ((c) and (d) panels of Fig. 8, Fig. 9 and Fig. 10) present the increasing tendency of the marine aerosol optical properties with wind speed while the tendency does not appear distinctly at $\text{Layer}_H$.





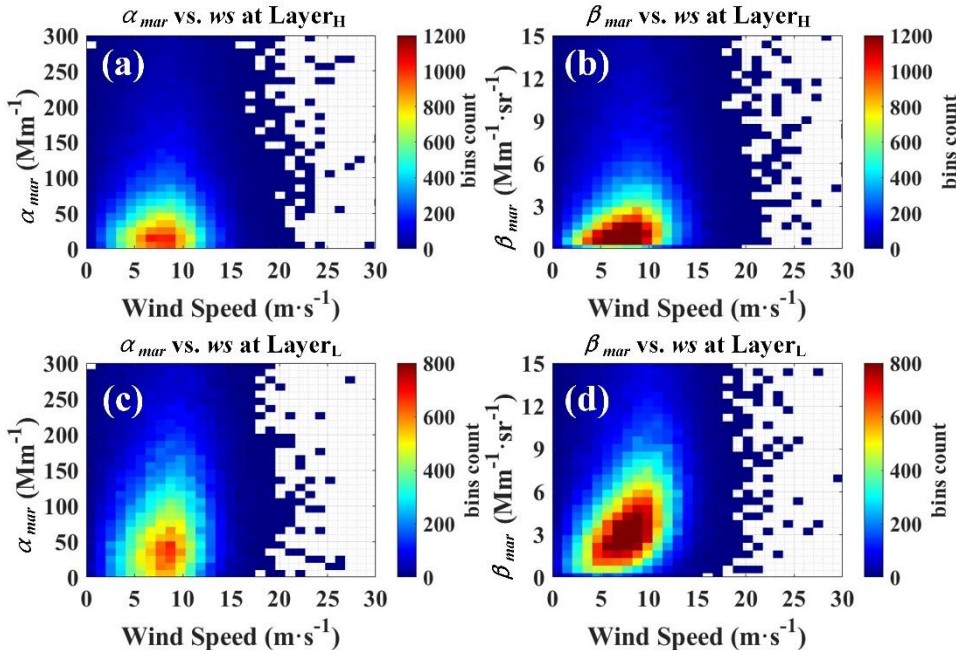

**Figure 8: Data counts of $\alpha_{mar}$ - $ws$ grid bins, $\beta_{mar}$ - $ws$ grid bins at separate layers above the NP area.**

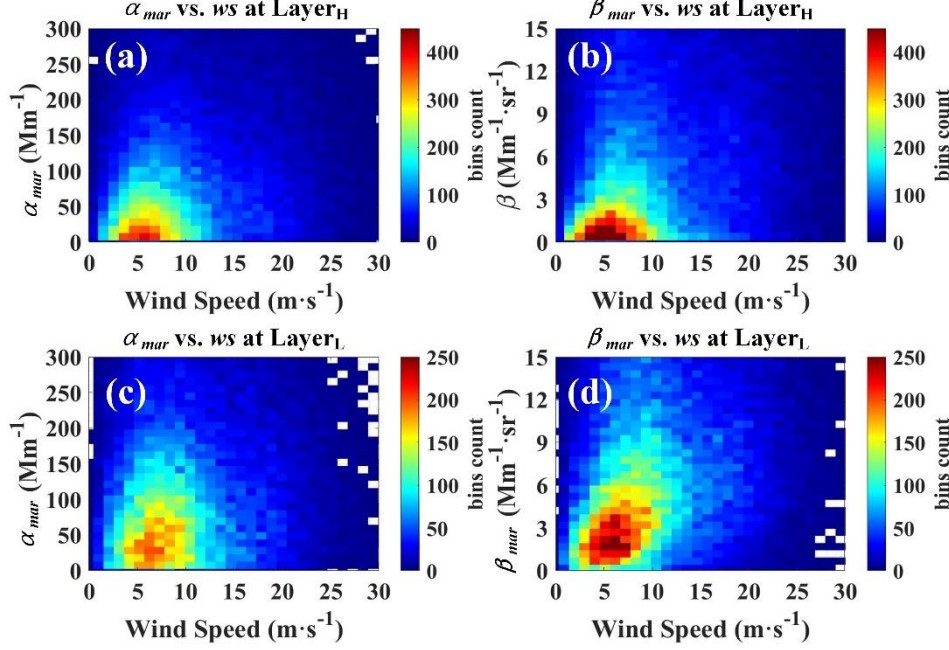


**Figure 9: Data counts of $\alpha_{mar}$ - $ws$ grid bins, $\beta_{mar}$ - $ws$ grid bins at separate layers above the SP area.**





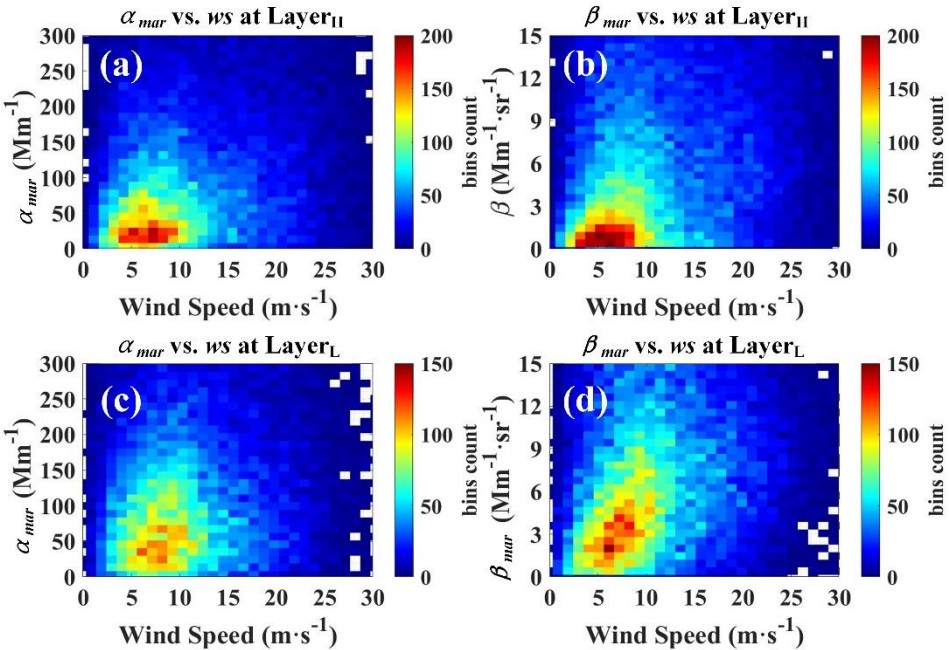

**Figure 10: Data counts of** $\alpha_{mar}$ **-** $ws$ **grid bins,** $\beta_{mar}$ **-** $ws$ **grid bins at separate layers above the SI area.**

In order to determine the explicit relationship between marine aerosol optical properties and corresponding wind speed, a grid

with resolution of 1 m·s⁻¹ from 0 m·s⁻¹ to 30 m·s⁻¹ is defined and the mean values and the standard deviations are calculated along the grid at both layers above the study areas, respectively. It should be emphasized primarily that before the calculation of mean of each wind speed grid, the outliers larger/less than the mean plus/minus one standard deviation are eliminated. About 70% to 80% $\alpha_{mar}$ and $\beta_{mar}$ are retained after the elimination. The quite strict outlier removal is conducted here to reject the data unrepresentative for marine aerosol (may be contaminated by cloud thus become higher than the typical

range). Hence, it can guarantee the data quality and validity of the pure marine aerosol optical properties in the statistical analysis process. Moreover, the wind speed grid of which the data counts are less than 100 is considered unrepresentative and the statistical result of this grid is abandoned. The mean values and standard deviations (after the outlier removal) of $\alpha_{mar}$ and $\beta_{mar}$ along with wind speed grid at two layers above the NP area, the SP area and the SI area are shown in Fig. 11, Fig. 12, and Fig. 13, respectively. The regression curves of the optical properties are presented in those figures as well. A power law

function is utilized for the curve fitting to describe the trend of marine aerosol optical properties with wind speed. Besides, the data counts in every wind speed grid are shown as the histograms in Fig. 11, Fig. 12, and Fig. 13. Table 1 summarizes the regression functions together with the corresponding $R^2$, and the proportions of different wind speed bins together with the count sums, grouped by areas, layers and optical properties.



From the statistical results with wind speed grids and wind speed bins, it can be found that most of the wind speeds are below

15 $m \cdot s^{-1}$ above the NP area, both at $Layer_H$ and $Layer_L$, meanwhile the proportion of low wind speed ($0 < ws \leqslant 8$) is slightly

higher at $Layer_H$ than at $Layer_L$. As for the SP area and the SI area, the high wind speed ($ws > 15$) accounts for around one

fifth and a quarter respectively, and the low wind speed proportion over the SP area is higher than that over the SI area. The

wind speed distribution is more concentrated at $Layer_L$ than at $Layer_H$ above these two areas, in view of the lower proportion

of low/high wind speed and the higher proportion of middle wind speed ($8 < ws \leqslant 15$) at $Layer_L$.

**Table 1: Regression functions of the averaged optical properties and the wind speed grids, together with the corresponding wind speed distributions, grouped by areas and layers.**

| Area | Layer | Optical property | Regression function | $R^2$ | Proportion of wind speed bins ($m \cdot s^{-1}$) | | | Number of counts |
|------|-------|------------------|---------------------|-------|------------------|------------------|--------|-------|
| | | | | | $0 < ws \leqslant 8$ | $8 < ws \leqslant 15$ | $ws > 15$ | |
| NP | H | $\alpha_{mar}$ | $\alpha_{mar} = 0.12 \cdot ws^{1.8} + 42$ | 0.96 | 0.52 | 0.46 | 0.02 | 57545 |
| | | $\beta_{mar}$ | $\beta_{mar} = 0.034 \cdot ws^{1.4} + 1.6$ | 0.96 | 0.53 | 0.45 | 0.01 | 73870 |
| | L | $\alpha_{mar}$ | $\alpha_{mar} = 0.27 \cdot ws^{1.9} + 64$ | 0.98 | 0.48 | 0.51 | 0.01 | 46854 |
| | | $\beta_{mar}$ | $\beta_{mar} = 0.081 \cdot ws^{1.5} + 2.3$ | 1.0 | 0.51 | 0.48 | 0.01 | 63005 |
| SP | H | $\alpha_{mar}$ | $\alpha_{mar} = 5.0 \cdot ws^{0.84} + 54$ | 0.95 | 0.46 | 0.34 | 0.20 | 39999 |
| | | $\beta_{mar}$ | $\beta_{mar} = 1.3 \cdot ws^{0.48} + 1.1$ | 0.96 | 0.48 | 0.31 | 0.21 | 39965 |
| | L | $\alpha_{mar}$ | $\alpha_{mar} = 6.1 \cdot ws^{0.83} + 73$ | 0.98 | 0.43 | 0.38 | 0.19 | 30526 |
| | | $\beta_{mar}$ | $\beta_{mar} = 1.8 \cdot ws^{0.47} + 1.1$ | 0.97 | 0.45 | 0.36 | 0.19 | 32375 |
| SI | H | $\alpha_{mar}$ | $\alpha_{mar} = 1.2 \cdot ws^{1.3} + 67$ | 0.95 | 0.40 | 0.35 | 0.25 | 24012 |
| | | $\beta_{mar}$ | $\beta_{mar} = 1.3 \cdot ws^{0.49} + 1.8$ | 0.96 | 0.40 | 0.33 | 0.27 | 22446 |
| | L | $\alpha_{mar}$ | $\alpha_{mar} = 4.3 \cdot ws^{0.85} + 88$ | 0.91 | 0.36 | 0.41 | 0.23 | 19489 |
| | | $\beta_{mar}$ | $\beta_{mar} = 0.89 \cdot ws^{0.61} + 3.4$ | 0.93 | 0.36 | 0.40 | 0.24 | 19473 |

Generally, in all cases shown in Fig. 11, Fig. 12 and Fig. 13, the optical properties at $Layer_L$ are all larger than those at $Layer_H$

in the same wind speed grid, while the variations of marine aerosol optical properties along with wind speed grid can be clearly

observed that the tendency is increasing with the wind speed. Moreover, the regression curves are fitted pretty well as the $R^2$

are all above 0.90.



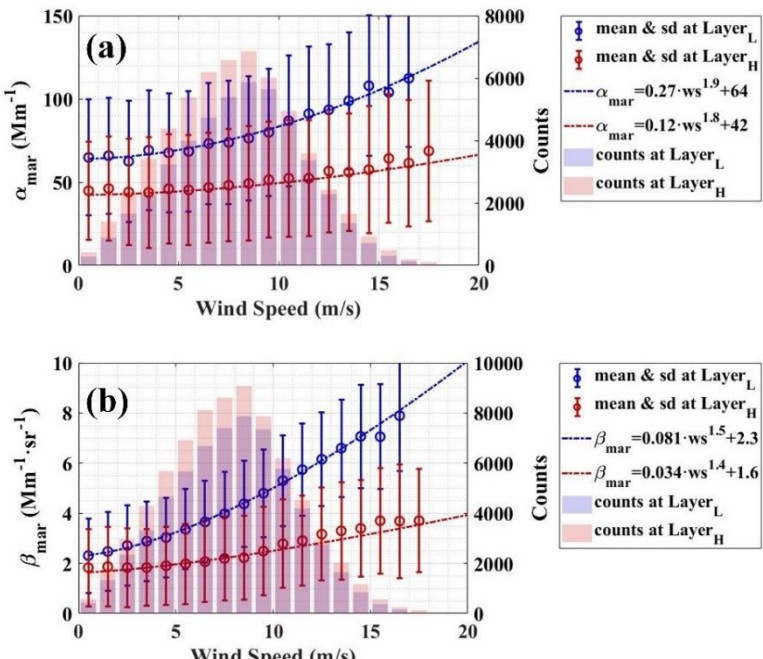

**Figure 11: Relationship between marine aerosol optical properties ((a) for $\alpha_{mar}$, (b) for $\beta_{mar}$) and wind speed above the NP area. The blue circles and error bars represent the means and standard deviations of the optical properties along wind speed grids at $\mathrm{Layer_L}$, while the reds represent the same items at $\mathrm{Layer_H}$. The blue and red dotted-dashed lines are the optical property averages**

**regression curves fitted along the wind speed grid at $\mathrm{Layer_L}$ and $\mathrm{Layer_H}$, respectively. The blue and red histograms indicate the data counts of every wind speed grid at $\mathrm{Layer_L}$ and $\mathrm{Layer_H}$, respectively.**

In the NP area, the exponents of the regression functions are all larger than 1, indicating the growth rates of the optical properties along the wind grid increases. Within the same wind speed grid, the gradient at $\mathrm{Layer_L}$ is larger than that at $\mathrm{Layer_H}$, i.e., the optical properties at $\mathrm{Layer_L}$ will increase more rapidly with wind speed. It is worth to notice that for the case that the

wind speed is below 15 $\mathrm{m \cdot s^{-1}}$, the $\alpha_{mar}$ and $\beta_{mar}$ seem to be fitted better by power law functions, whereas under the condition when the wind speed is higher than 15 $\mathrm{m \cdot s^{-1}}$, the values of the optical properties show higher fluctuations ($\alpha_{mar}/\beta_{mar}$ at $\mathrm{Layer_L}$) or are invariant ($\beta_{mar}$ at $\mathrm{Layer_H}$), deviating from the power laws. It may result from that the limited number of counts when the wind speed is larger than 15 $\mathrm{m \cdot s^{-1}}$, so that the statistical significance turns to be weak. However, the hypothesis could be discussed that there might be two distinct variation trends of the marine aerosol optical properties above or below the

wind speed range of 15 $\mathrm{m \cdot s^{-1}}$.





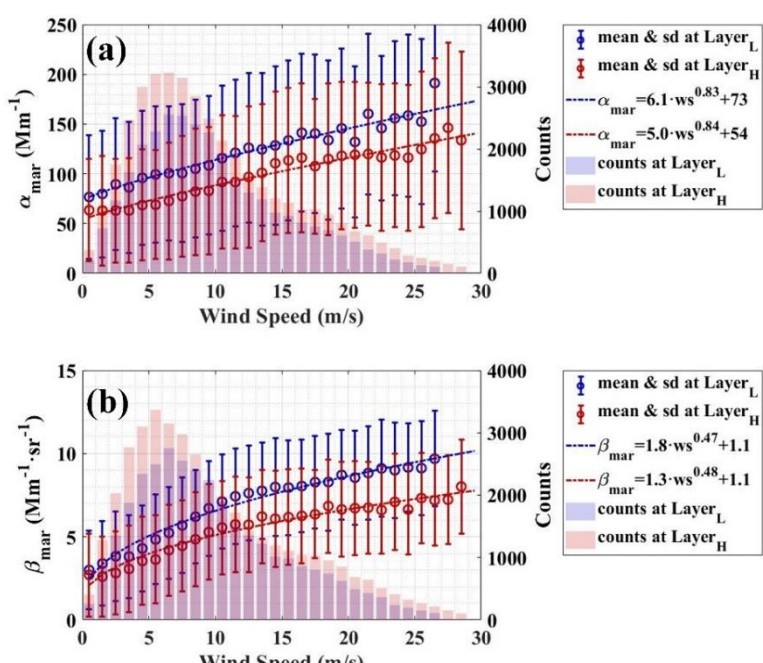

**Figure 12: Relationship between marine aerosol optical properties ((a) for $\alpha_{mar}$, (b) for $\beta_{mar}$) and wind speed above the SP area. The items represent the same as those of Fig. 11.**

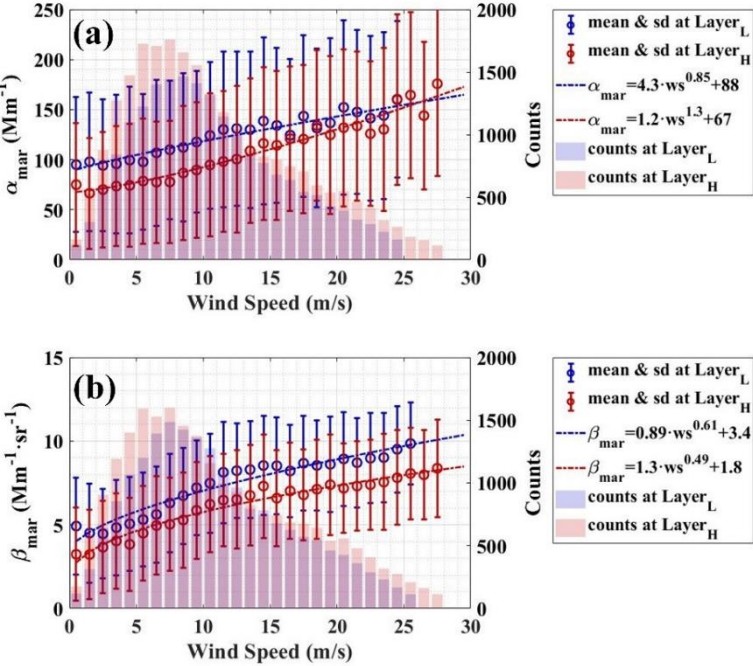

**Figure 13: Relationship between marine aerosol optical properties ((a) for $\alpha_{mar}$, (b) for $\beta_{mar}$) and wind speed above the SI area. The items represent the same as those of Fig. 11.**





For the SP area and the SI area, as mentioned above, the wind speed distribution ranges are larger, of which the maximum is up to 29 $m \cdot s^{-1}$, while the variations of the optical properties along with wind speed are more complicated. As for the $\alpha_{mar}$ over the SP area, they show approximate linear growth tendencies with similar gradients both at $Layer_L$ and at $Layer_H$ when the wind speed is below 17 $m \cdot s^{-1}$. When the wind speeds exceed 17 $m \cdot s^{-1}$, the growth rates of $\alpha_{mar}$ become smaller with several fluctuating values. The change point of the gradient can also be discovered in the variations of $\beta_{mar}$, which is approximately 12 $m \cdot s^{-1}$ for the both layers. Regardless of below the gradient change point or above it, the growth rate of $\beta_{mar}$ at $Layer_L$ is higher than that at $Layer_H$. In the SI area, the variations of the optical properties along with the wind speed grid are similar to those in the SP area. The gradients of $\alpha_{mar}$ are quite similar to each other at both layers while the growth rate of $\beta_{mar}$ at the lower layer is larger than that at the higher layer. The change points of the gradient also exist at around 15 $m \cdot s^{-1}$ and at around 12 $m \cdot s^{-1}$, for $Layer_L$ and $Layer_H$ respectively. However, it is noticed that some optical property values at the wind speed of 1 $m \cdot s^{-1}$ to 5 $m \cdot s^{-1}$ in the SI area are larger than those in the SP area, further the invariability or even the decrease occur at this wind speed bin. It is worth to note that under quite high wind speed condition (24 $m \cdot s^{-1}$ to 28 $m \cdot s^{-1}$), the $\alpha_{mar}$ values at $Layer_H$ in the SI area enhance sharply, resulting in an exponent above 1. It indicates that after the slow growth phase of the $\alpha_{mar}$ in the wind speed bins of 15 $m \cdot s^{-1}$ to 24 $m \cdot s^{-1}$, $\alpha_{mar}$ will still increase significantly. The same phenomenon can be observed at $Layer_H$ in the SP area. Nevertheless, this rapid enhancement of $\alpha_{mar}$ is not possible to be observed at $Layer_L$, which is caused by an insufficient low number of counts for high wind speeds. n. This phenomenon is also not shown in the $\beta_{mar}$ variations at high wind speed in both areas, as the variation trends of $\beta_{mar}$ at high wind speed present similar gradients with those at the slow growth phase.

Consequently, for all of the measurement cases, and identical wind speed bins, the marine aerosol optical properties at $Layer_L$ are larger than those at $Layer_H$, indicating that the MABL possibly receive more marine aerosol produced and transport from the air-sea interface, while the higher layer above the MABL with the upper boundary of 2 km can also be influenced by the marine aerosol but less. In the NP area, the variations of the marine aerosol optical properties with the relatively concentrated wind speed distribution are all fitted quite well by a power law functions of with the exponents above 1. The gradients at $Layer_L$ are obviously larger than at $Layer_H$ for the same wind speed, which implies that the marine aerosol enhancement caused by the background wind are much more intensive at the MABL. In the SP area and the SI area, the enhancement rates of the marine aerosol optical properties at both layers are similar with the large wind speed bins up to 29 $m \cdot s^{-1}$, which indicates that the marine aerosol evolution of both atmospheric layers over the ocean will be affected by the background wind. It should be emphasized that the gradient change points appear during the growth of the optical properties along with wind speed in these two areas. The gradient change points of $\alpha_{mar}$ are greater than those of $\beta_{mar}$, and above them the enhancement



rate becomes lower. It might illustrate that after the gradient change points, the marine aerosol enhancement is likely to diminish with the wind speed increasing. The possible sharp increase of $\alpha_{mar}$ at $\text{Layer}_H$ under high wind speed condition (24 $\text{m} \cdot \text{s}^{-1}$ to 28 $\text{m} \cdot \text{s}^{-1}$) is worth to notice as well.

In this section, histograms for the marine aerosol optical properties at 355 nm versus the wind speed with $\alpha_{mar}$ - $ws$ grid and

$\beta_{mar}$ - $ws$ grid at two separate atmospheric layers are discussed. Then the statistical results of the marine aerosol optical properties at 355 nm along with the wind speed grid at two layers together with the corresponding regression curves are determined and analysed. It is found that both the MABL and the higher layer above the MABL can receive the marine aerosol produced and transported from air-sea interface. Moreover, the marine aerosol load at the lower layer (MABL) is stronger than at the higher layer. The marine aerosol shows enhancement when the wind speed is increasing, indicating that the wind

performs as a significant factor for the marine aerosol at these layers. The wind speed bins and the marine aerosol variation tendencies with wind speed at these vertical layers above the individual study areas (the NP area, located in the Pacific Ocean, the low latitudes of the Northern Hemisphere; the SP area, located in the Pacific Ocean, the middle latitudes of the Southern Hemisphere; the SI area, located in the Indian Ocean, the middle latitudes of the Southern Hemisphere) are different from each other. It implies that the development and evolution of the marine aerosol above the ocean might not only be dominated by the

driving of the wind, but also be impacted by other factors. That is to say, as concluded in Lewis and Schwartz (2004), apart from wind, there are some other meteorological and environmental factors, e.g. atmospheric stability, sea and air temperature, RH and so on, that are capable of affecting marine aerosol production, entrainment, transport and removal.

## 4.4 Dependency of aerosol optical depth and lidar ratio with wind speed

### 4.4.1 Marine aerosol optical depth vs. wind speed

As introduced in Section 1 of this paper, almost all the previous researches on the relationship between marine aerosol's optical properties and wind speed focused on the AOD of marine aerosol. In this study, as well, the effort on the averaged 0-2 km $\text{AOD}_{mar}$ of individual wind speed grid calculation has been conducted to compare the $\text{AOD}_{mar}$ - $ws$ relationship from previous study. The $\text{AOD}_{mar}$ of every single profile is acquired by integrating Aeolus retrieved $\alpha_{mar}$ within 2 km. The wind speed profiles are also averaged for 2 km to correspond to the $\text{AOD}_{mar}$ data. Then the relationship between the $\text{AOD}_{mar}$ and the

wind speeds is obtained by averaging the $\text{AOD}_{mar}$ in each wind speed grid (0 $\text{m} \cdot \text{s}^{-1}$ - 30 $\text{m} \cdot \text{s}^{-1}$, stepped by 1 $\text{m} \cdot \text{s}^{-1}$). The $\text{AOD}_{mar}$ - $ws$ relationship is also explored utilizing the products from the A-Train satellites (Kiliyanpilakkil and Meskhidze, 2011). "clean marine" aerosol AOD at 532 nm above ocean surface (up to 2 km) provided by CALIOP and 10 m daily wind speed provided by AMSR-E were used. It should be noticed that the wind speed used in Kiliyanpilakkil and Meskhidze (2011) is daily ocean surface wind speed, different from that in this study, which is instantaneous layer-averaged wind speed.

Collecting the data for the time period from 2006 to 2011 over 15 remote ocean regions globally, the regression curve of is



acquired with the averaged $AOD_{mar}$ at 532 nm for each wind speed grid and the surface wind speed which is up to 29 $m \cdot s^{-1}$, and the regression function is shown as the following Eq. 3:

$$AOD_{mar,532} = \frac{0.15}{1+6.7 \cdot e^{-0.17 \cdot U_{10}}} ,$$ (3)

where the $U_{10}$ represents daily 10 m ocean surface wind speed.

As described above, the $AOD_{mar}$ data source (from spaceborne lidar observation), the study areas (remote ocean regions globally), and the wind speed range (0 $m \cdot s^{-1}$ - 29 $m \cdot s^{-1}$) of the $AOD_{mar}$ - $ws$ relationship exploration in Kiliyanpilakkil and Meskhidze (2011) are all quite similar with those of this study. Hence, we select the $AOD_{mar}$ - $ws$ relationship established by Kiliyanpilakkil and Meskhidze (2011) for the comparison. Additionally, due to the different wavelengths of $AOD_{mar}$ used in this study (355 nm) and in Kiliyanpilakkil and Meskhidze (2011) (532 nm), the effort on conversion of the $AOD_{mar}$ at 532 nm

to the $AOD_{mar}$ at 355 nm is performed by applying the typical Ångström exponent of marine aerosol. It is reported that the marine aerosol Ångström exponent is surface wind speed related, and a linear relationship was established as the following Eq. 4 (Sayer et al., 2012):

$$A(ws)=0.69-0.030 \cdot ws ,$$ (4)

where $A$ represents the Ångström exponent and $ws$ represents the wind speed. Then the $AOD_{mar}$ at 532 nm can be converted

to the $AOD_{mar}$ at 355 nm by the following Eq. 5:

$$AOD_{mar,355nm}(ws)=\exp[A(ws) \cdot \ln \frac{532}{355}] \cdot AOD_{mar,532nm}(ws) .$$ (5)

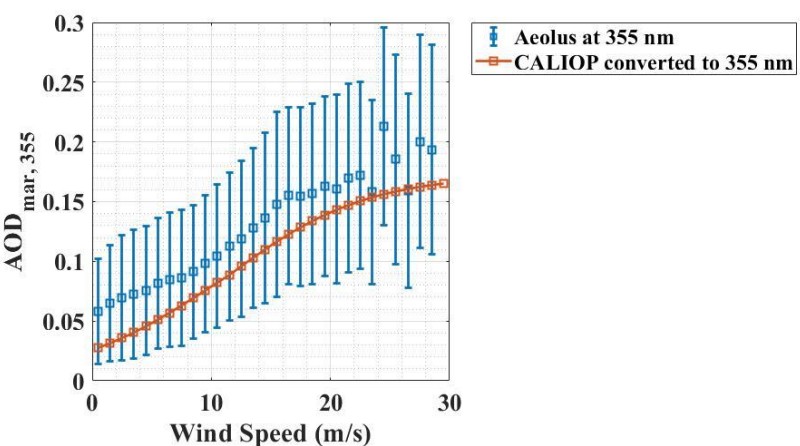

**Figure 14:** $AOD_{mar}$ **at 355 nm versus wind speed. The blue squares and the corresponding error bars represent the** $AOD_{mar}$ **means and standard deviations along the** $ws$ **grid of all the three study areas in this study; the red squares and line represent the** $AOD_{mar}$

**at 355 nm along the** $ws$ **grid converted from the regressive relationship between the** $AOD_{mar}$ **at 532 nm and the ocean surface wind speed reported by Kiliyanpilakkil and Meskhidze (2011).**



In Fig. 14, the averaged $AOD_{mar}$ and the corresponding standard deviations at 355 nm of all three study areas along the wind speed grid are represented as the blue squares and the error bars, while the regression curve of $AOD_{mar}$ at 355 nm versus wind speed converted from Eq. 3 are represented as the red squares and line. Although instantaneous layer-averaged wind speed

from ECMWF model and daily ocean surface wind speed are used in this study and in Kiliyanpilakkil and Meskhidze (2011) individually, quite similar tendency of $AOD_{mar}$ at 355 nm versus wind speed are obtained. $AOD_{mar}$ is increasing with wind speed, and the gradient of $AOD_{mar}$ turns out becoming higher along the wind speed grids when the wind speed is below 15 $m \cdot s^{-1}$ while the variation of $AOD_{mar}$ becomes lower above 15 $m \cdot s^{-1}$. The converted $AOD_{mar}$ are all slightly lower than the Aeolus retrieved $AOD_{mar}$, nevertheless the formers are all in the standard deviations range of the latter, thus it is considered

that the Aeolus retrieved $AOD_{mar}$ and their variation along the wind speed are reasonable. The lower $AOD_{mar}$ from CALIOP after wavelength conversion may arise from using a fixed lidar ratio of 20 sr at 532 nm used for CALIOP $AOD_{mar}$ retrievals while the lidar ratio of marine aerosol can vary with the particle size. Possible underestimation of the CALIOP retrieved $AOD_{mar}$ at 532 nm is discussed in detail in Kiliyanpilakkil and Meskhidze (2011). Besides, as discussed in Section 4.4.2 of this paper, the particle size and the lidar ratio of the marine aerosol will vary with wind speed, so using the $AOD_{mar}$ retrieved

with the fixed lidar ratio may generate additional error in the exploration of the relationship between the $AOD_{mar}$ and the wind speed. Therefore, using Aeolus retrieved $AOD_{mar}$, which is integrated by independently retrieved extinction coefficient without the assumption of lidar ratio, could make the $AOD_{mar}$ - $ws$ relationship more reliable. The slightly high Aeolus retrieved $AOD_{mar}$ may result from the possible cloud contaminations of the marine aerosol data bins.

### 4.4.2 Marine aerosol lidar ratio vs. wind speed

Derived from $\alpha_{mar}$ and $\beta_{mar}$, $LR_{mar}$ is defined as the ratio of the former to the latter. As one of the intensive optical properties, it is independent of the aerosol concentration. It is reported that the $LR_{mar}$ depends on the particle size, and specifically, with the reduction of the coarse mode, the total lidar ratio turns out to increase (Masonis et al., 2003). The possible reason for this phenomenon is that as the particles become smaller, the extinction is enhanced by the increasing sideward scattering and the backscatter gets weaker due to the decrease of the scattering cross section (Haarig et al., 2017). Aeolus L2A product provide

particle extinction-to-backscatter ratio calculated with the raw backscatter coefficient, which lacks the depolarized portion, as introduced in Section 3 of this paper. In this work, the corrected marine aerosol lidar ratio is acquired by dividing the marine aerosol extinction to the marine aerosol depolarization-corrected backscatter. The calculation of the averaged $LR_{mar}$ along wind speed grid has been conducted by averaging the marine aerosol lidar ratios of each 1 $m \cdot s^{-1}$ wind speed bin, meanwhile the standard deviations are acquired as well. It should be noted that before the statistical calculation, the outliers are eliminated

by the method of Tukey's test, which is introduced in Section 3 of this paper.



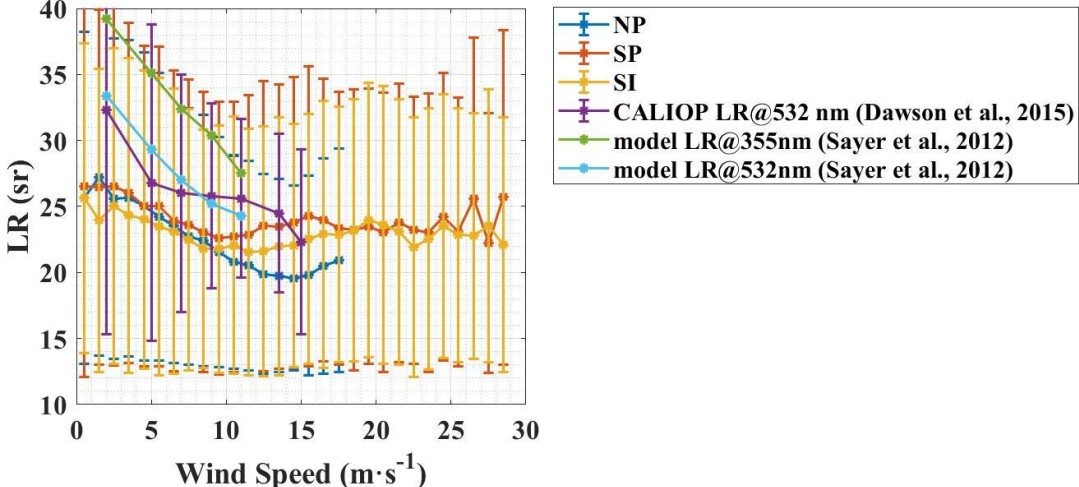

**Figure 15:** $LR_{mar}$ **versus the wind speed. The dark blue curve, red curve, yellow curve and the corresponding error bars represent the averaged** $LR_{mar}$ **and their standard deviations above the NP area, the SP area and the SI area, respectively. The purple curve and the corresponding error bars represent the CALIOP-retrieved** $LR_{mar}$ **at 532 nm (Dawson et al., 2015). The green curve and the**

**light blue curve represent the modelled** $LR_{mar}$ **at 355 nm and at 532 nm, respectively (Sayer et al., 2012).**

In Fig. 15, 0-2 km averaged $LR_{mar}$ variations along with the wind speed above the NP area, the SP area and the SI area are represented as the blue curve, the red curve and the yellow curve, respectively. Generally, the distinct downward trend of the $LR_{mar}$ at relatively low wind speeds(0-15 $m \cdot s^{-1}$ of the NP area, 0-10 $m \cdot s^{-1}$ of the SP area and the SI area) can be observed in all cases. Similar results have been investigated with the measured $LR_{mar}$ or modelled $LR_{mar}$ and the ocean surface wind

speed in the previous studies. Combining the corrected CALIOP-retrieved $LR_{mar}$ at 532 nm and 10 m ocean surface wind speed from the Advanced Microwave Scanning Radiometer (AMSR-E), the negative correlation between the $LR_{mar}$ and wind speed is acquired with the wind speed bins of 0 $m \cdot s^{-1}$ up to $>15$ $m \cdot s^{-1}$ , shown as the purple curve in Fig. 15 (Dawson et al., 2015). The modelled $LR_{mar}$ at 355 nm and at 532 nm also presents decreasing trends with the wind speed increases, presented as the green curve and the light blue curve in Fig. 15 (Sayer et al., 2012). These results seem to imply that as the

wind speed increases for a low wind speed range, the particle size of marine aerosol get larger. The phenomenon is explained by the shift in marine aerosol volume size distribution with wind speed as wind speed increases, the fine mode volume size distribution of marine aerosol turns out decline while the coarse mode distribution becomes larger (Dawson et al., 2015; Smirnov et al., 2003; Sayer et al., 2012). The CALIOP $LR_{mar}$ and the modelled $LR_{mar}$ are all larger than the $LR_{mar}$ of this study but are all in the standard deviation ranges. According to Bohlmann et al. (2018), Groß et al. (2011), Groß et al. (2015),

and Floutsi et al. (2022), the pure $LR_{mar}$ at 355 nm can vary from 10 sr to 40 sr, with the average of around 20 sr, thus it is considered that the averaged $LR_{mar}$ in this study are reasonable. At the middle wind speed range (15 $m \cdot s^{-1}$ -18 $m \cdot s^{-1}$ of the



NP area, 10 m·s$^{-1}$ -16 m·s$^{-1}$ of the SP area, 10 m·s$^{-1}$ -20 m·s$^{-1}$ of the SI area), the LR$_{mar}$ show upward tendencies, implying that the marine aerosol particles might be broken up into smaller ones with a wind speed increase At the very high wind speed (>16 m·s$^{-1}$) above the SP area and the SI area, the marine aerosol particle size variations turn out to a stable state

along with the wind speed.

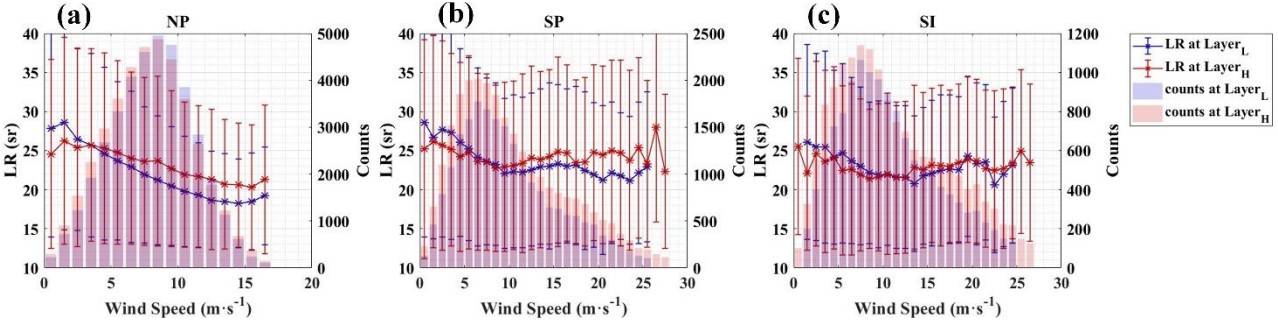

**Figure 16: Averaged** LR$_{mar}$ **versus wind speed at** Layer$_L$ **and** Layer$_H$ **, in (a) the NP area, (b) the SP area and (c) the SI area, respectively.**

The LR$_{mar}$ of two sperate vertical layers are also calculated and discussed. Figure 16 shows the LR$_{mar}$ variations at Layer$_L$

and Layer$_H$ along with the wind speed grid in three study areas. Some divergences of the LR$_{mar}$ variations between the layers can be discovered. In all the three study areas, the particle size of marine aerosol at Layer$_L$ is likely to vary from smaller than at Layer$_H$ to bigger than at Layer$_H$ when the wind speed increases. During the LR$_{mar}$ decreasing phase along with the wind speed, it is indicated that the variations of the particle size at Layer$_L$ are larger than those at Layer$_H$ .

Generally, the LR$_{mar}$ dependency along with the wind speed shows a downward trend at relatively low wind speed, then

upward trend at the middle wind speed, finally stable at the very high wind speed (if exist), which implies that the marine aerosol particle size is increasing along with the wind speed at first, then might be broken up into smaller one by the enhanced wind speed and ultimately turns out to a stable size. Several differences of the LR$_{mar}$ variations along with the wind speed appear between the three study areas and the two vertical layers, which may be due to the difference in meteorological and environmental conditions of the areas and the layers.

**5 Summary and conclusion**

By utilizing Level 2A products (particle optical properties and numerical weather prediction data) and Level 2C products (numerical weather prediction wind vector assimilated with observed wind component) provided by ALADIN, and L2 vertical feature mask (VFM) products provided by CALIOP, the optical properties at 355 nm of pure marine aerosol are derived. Then the combined analysis of marine aerosol optical properties at 355 nm and instantaneous co-located wind speed from ECMWF

model above remote ocean areas is conducted. Finally, their relationships are explored and discussed at two separate vertical



atmospheric layers ( $\text{Layer}_L$ with the height of 0-1 km and $\text{Layer}_H$ with the height of 1-2 km, correspond to the heights within and above marine atmospheric boundary layer (MABL)).

Pure marine aerosol is identified by distinguishing marine aerosol from other aerosols and from clouds. The areas selected procedure in this work includes selecting the ocean areas far away from the lands, checking the potential of terrestrial aerosol input by analysing the wind direction of the selected areas, and then examining the domination of marine aerosol with the aerosol classification data provided by CALIOP VFM products. After quality control, cloud screening and backscatter correction is applied to the Aeolus L2A products, this allow us to obtain reliable, cloud-free marine aerosol optical properties and the corresponding wind speed.

The spatial distributions of marine aerosol optical properties and wind speed above the North Pacific (NP) area, the South Pacific (SP) area and the South Indian (SI) area are respectively presented and analysed within and above the marine atmospheric boundary layer. The results show, since the three study areas are located in different hemispheres, different latitudes and different oceans, that the spatial distributions of wind speed ( $ws$ ), marine aerosol extinction coefficient ( $\alpha_{mar}$ ), marine aerosol backscatter coefficient ( $\beta_{mar}$ ) and marine aerosol lidar ratio ( $\text{LR}_{mar}$ ) are not totally similar. Nevertheless, the marine aerosol optical properties and the wind speed show positive correlation both over $\text{Layer}_H$ and $\text{Layer}_L$ , whereas the optical properties at $\text{Layer}_L$ are larger than those at $\text{Layer}_H$ , indicating that both layers are affected by marine aerosol produced and transported from air-sea interface but the majority of the marine aerosol is trapped in the MABL while a small fraction can be elevated into the higher layer above the MABL.

The statistical results of the marine aerosol optical properties at 355 nm along with the wind speed grid at two vertical layers together with the corresponding regression curves fitted by power law functions are acquired and analysed, for three study areas respectively. The optical properties present increasing trends with wind speed in all cases. As for the NP area, the gradients of the optical properties at $\text{Layer}_L$ are higher than at $\text{Layer}_H$ in the identical wind speed grid, illustrating that the marine aerosol enhancement caused by the background wind are much more intensive in the MABL. The exponents of all the regression functions are above 1, which indicates that the growth rates of the optical properties with wind speed become larger. In aspect of the SP area and the SI area, the growth rates of the marine aerosol optical properties at both layers are similar, which reveals that the marine aerosol evolution within or above the MABL over ocean will be affected by the background wind. The gradient change points appear during the growth of the optical properties along wind speed, above which the growth rate turns out lower, implying that after the gradient change points, the marine aerosol enhancement is likely to diminish with the wind speed increasing. Compared with the regression function between wavelength-converted CALIOP-retrieved $\text{AOD}_{mar}$ and 10 m surface wind speed, the $\text{AOD}_{mar}$ at 355 nm versus the wind speed in this work shows quite consistent tendency with CALIOP's though the wind speeds used are different. The $\text{LR}_{mar}$ depends on the particle size. The relationship between the $\text{LR}_{mar}$ and the wind speed are also explored and discussed, indicating that as the wind speed is increasing, the particle size of

marine aerosol obviously becomes larger at relative low wind speed range, then could be broken up by wind at higher wind speed, and ultimately turns out a stable size state at very high wind speed.

To conclude, this study derived pure marine aerosol optical properties and the corresponding wind speed, explored the relationship between these two elements, revealing the marine aerosol related atmospheric background states at two sperate layers within and above the MABL over the remote oceans. The atmosphere of the two vertical layers over the ocean areas will both receive the marine aerosol input produced and transported by the wind and the turbulence. The marine aerosol enhancement caused by the wind speed in the MABL is more intensive than at the higher layer. The marine aerosol optical properties distributions, wind speed bins, and the marine aerosol variation tendencies along wind speed grid above the individual study areas are not totally similar. It can be inferred that the development and evolution of the marine aerosol above the ocean might not only be dominated by the wind, but also be impacted by other meteorological and environmental factors, e.g. atmospheric stability, sea and air temperature, RH and so on. If future study is capable to obtain more other meteorological parameters above ocean, jointly analysing the aerosol optical properties and the wind together with them, more detailed information of marine aerosol production, entrainment, transport and removal will be acquired.

**Data availability**

The Aeolus data are downloaded via the website https://aeolus-ds.eo.esa.int/oads/access/collection (last access: 9 March 2023). Part of the Aeolus L2A and L2C data we used in this paper are not available publicly at the time the article was submitted. We are allowed to access the data through our participation as a Calibration and Validation team. The CALIOP data can be downloaded from https://eosweb.larc.nasa.gov/project/CALIPSO (last access: 9 March 2023).

**Author contributions**

G. Dai conceived of the idea for correlation between marine aerosol optical properties and wind fields over remote oceans with spaceborne lidars ALADIN, CALIOP; K. Sun wrote the manuscript; K. Sun, G. Dai, S. Wu, O. Reitebuch and H. Baars contributed to the data analyses; J. Liu and S. Zhang contributed to the scientific discussion. All the co-authors reviewed and edited the manuscript.

**Competing interests**

The authors declare that they have no conflict of interest.

**Special issue statement**

This article is part of the special issue "Aeolus data and their application". It is not associated with a conference.



**Acknowledgments**

625 This study has been jointly supported by the Laoshan Laboratory Science and Technology Innovation Projects under grant LSKJ202201406, the National Natural Science Foundation of China (NSFC) under grant 61975191, 41905022 and U2106210. This work was also supported by Dragon 5 program which conducted by European Space Agency (ESA) and the National Remote Sensing Center of China (NRSCC) under grant 59089.

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
