# Peer review of "Effect of wind speed on marine aerosol optical properties over remote oceans with use of spaceborne lidar observations"

_EGUsphere, 2023_

## Author Comment (AC1)

The paper makes an important contribution to the literature and can provide input to the modeling community regarding the sea-salt emissions.

Some minor comments before publication:

Using ECMWF model constraints for RH would not necessarily remove clouds from L2A measurements (it is well-known that clouds are not well-represented in models)

AR: In this study, it is considered that clouds should be adequately removed as far as possible to retain aerosol extinction/backscatter coefficients, and to avoid the clouds' possible impact on the marine aerosol optical properties-wind relationship exploration.

According to the recommendation in Flamant et al. (2020), which is the Aeolus Level-2A Algorithm Theoretical Basis Document, there is a high probability that a cloud be present if RH > 94%. Because of this probability of cloud existence under the atmospheric condition of high RH, the Aeolus L2A aerosol optical properties data bins with RH > 94% were eliminated to avoid the clouds' impact.

Actually, the RH data from ECMWF model is considered as the auxiliary criterion in the cloud screening procedure, while the backscatter ratio is the main criterion as it is from Aeolus measurement. In the revised manuscript, we put the "backscatter ratio" before the "RH" both in Fig. 2 (Fig. 1 in the old version) and in the relevant description to clarify the order of importance, which are shown as below:

[Figure]

**Figure 2: Flowchart of the study methodology**

"It is considered that a cloud is quite likely to exist if the backscatter ratio (BR) (total backscatter coefficient/molecular backscatter coefficient) at 355 nm is larger than 2.5 or the RH is larger than 94% (Flamant et al., 2020). Therefore, in this study, when the BR is larger than 2.5 or the RH is higher than 94%, the corresponding data bin is regarded as cloud contaminated and is eliminated."

Reference: *Flamant, P. H., Lever, V., Martinet, P., Flament, T., Cuesta, J., Dabas, A., Olivier, M., Huber, D., Trapon, D., and Lacour, A.: Aeolus Level-2A Algorithm Theoretical Basis Document, version 5.7, https://earth.esa.int/eogateway/documents/20142/37627/Aeolus-L2A-Algorithm-Theoretical-Baseline-Document (last access: 9 November 2022), 2020.*

It is better to use marine particle depolarization at 355nm from the Delian model (Floutsi et al., 2023). Gross' paper reports depol values at 532nm (even though the difference is not large, 1.3 vs 2%).

AR: Thanks for the suggestion. We have re-processed all the results with the marine particle depolarization at 355nm of 1.3 % from Floutsi et al. (2023). The figures and the relevant descriptions have been replaced and rephrased in the revised manuscript.

CALIPSO cannot verify the presence of a specific aerosol type, since the aerosol type is inferred based on assumptions on the surface type. Even though the regions selected are dominated by marine particles, it is better to rephrase as it concerns CALIPSO and further validate through a global model that there are no other types present (e.g. from ship emissions).

AR: According to Kim et al. (2018), the aerosol types discrimination of CALIPSO is based not only on surface type, but also on the particulate depolarization ratio, integrated attenuated backscatter coefficient at 532 nm, layer top altitude and layer base altitude. As shown below, Fig. 1 is Fig. 1 in Kim et al. (2018), which presents the Flowchart of the CALIPSO aerosol subtype selection scheme for tropospheric aerosols and the method was well studied and discussed in this paper. Therefore, the aerosol types discrimination of CALIPSO is not totally based on assumptions, but mainly infers from the lidar measurement and combines with the assumptions on the surface type. It is considered that the aerosol subtype data provided from CALIPSO is reasonable.

[Figure]

**Figure 1: Flowchart of the CALIPSO aerosol subtype selection scheme for tropospheric aerosols (Fig. 1 from Kim et al. (2018)).**

As for the shipping emission aerosol, the Intergovernmental Panel on Climate Change (IPCC) Fifth

Assessment Report provided the average value of international shipping emission aerosols, which is 5.5 $Tg \cdot yr^{-1}$ with the minimum of 3.6 $Tg \cdot yr^{-1}$ and the maximum of 8.7 $Tg \cdot yr^{-1}$. Comparing to the estimated sea spray aerosol emission of 1400-6800 $Tg \cdot yr^{-1}$, it is considered that the shipping emission aerosol is negligible.

*Reference: Kim, M.-H., Omar, A. H., Tackett, J. L., Vaughan, M. A., Winker, D. M., Trepte, C. R., Hu, Y., Liu, Z., Poole, L. R., Pitts, M. C., Kar, J., and Magill, B. E.: The CALIPSO version 4 automated aerosol classification and lidar ratio selection algorithm, Atmos. Meas. Tech., 11, 6107–6135, https://doi.org/10.5194/amt-11-6107-2018, 2018.*

In 4.2 the comparison of Aeolus with CALIPSO on extensive properties (a, b), should be restricted for backscatter only (CALIPSO cannot deliver extinction). Extinction could be evaluated against passive sensors such as MODIS AODs over the region.

AR: The purposes of the section of "Marine aerosol optical depth vs. wind speed" is to compare the AOD-wind speed relationship acquired from Aeolus with the result in a peer reviewed, published work. We quoted the AOD-wind speed relationship in Kiliyanpilakkil and Meskhidze (2011), which is acquired from the combination of AOD at 532 nm from CALIOP and 10 m wind speed from AMSR-E. The reason of choosing the result from Kiliyanpilakkil and Meskhidze (2011) for the comparison is that the AOD data source (from spaceborne lidar observation), the study areas (remote ocean regions globally), and the wind speed range (0 m/s- 29 m/s) of the AOD-wind speed relationship exploration in Kiliyanpilakkil and Meskhidze (2011) are all quite similar with those of this study. This comparison is considered capable to verify the AOD-wind speed relationship from Aeolus, and exactly, further to highlight the advantage of Aeolus on the AOD-wind speed relationship exploration, as the CALIPSO cannot deliver extinction precisely.

Fig. 11 from the revised manuscript (as shown below) shows the comparison between these two relationships. Indeed, AODs provided by CALIPSO-CALIOP are retrieved with the combination of the measurements of total attenuated backscatter coefficients and the assumptions of aerosol lidar ratios. Though the CALIOP AODs are based partly on assumptions, the AOD values from Aeolus are quite close (though slightly higher at low wind speed) to those from CALIOP and the tendencies of the two relationships are similar. The difference between AODs from CALIOP and from Aeolus are discussed in the manuscript as:

"The lower $AOD_{mar}$ from CALIOP after wavelength conversion at low wind speed may arise from using a fixed $LR_{mar}$ of 20 sr at 532 nm used for CALIOP $AOD_{mar}$ retrievals while the $LR_{mar}$ can vary with the particle size. Possible underestimation of the CALIOP retrieved $AOD_{mar}$ at 532 nm is discussed in detail in Kiliyanpilakkil and Meskhidze (2011). Besides, as discussed in Section 4.4.2 of this paper, the particle size and the LR of the marine aerosol will vary with wind speed, so using the CALIOP $AOD_{mar}$ retrieved with the fixed $LR_{mar}$ may generate additional error in the exploration of the relationship between the $AOD_{mar}$ and the wind speed. Therefore, using Aeolus retrieved $AOD_{mar}$, which is integrated by independently retrieved extinction coefficient without the assumption of $LR_{mar}$, could make the $AOD_{mar}$ - $ws$ relationship more reliable." From the comparison and the description, the marine AOD-wind speed relationship from Aeolus is verified while the advantage of Aeolus that it can retrieve AOD without any assumptions than CALIOP is illustrated.

[Figure]

**Figure 11:** $AOD_{mar}$ **at 355 nm versus wind speed. The blue squares and the corresponding error bars represent the** $AOD_{mar}$ **means and standard deviations along the** $ws$ **grid of all the three study areas in this study; the red squares and line represent the** $AOD_{mar}$ **at 355 nm along the** $ws$ **grid converted from the regressive relationship between the** $AOD_{mar}$ **at 532 nm and the ocean surface wind speed reported by Kiliyanpilakkil and Meskhidze (2011).**

*Reference: Kiliyanpilakkil, V. P. and Meskhidze, N.: Deriving the effect of wind speed on clean marine aerosol optical properties using the A-Train satellites, Atmos. Chem. Phys., 11, 11401–11413, https://doi.org/10.5194/acp-11-11401-2011, 2011.*

---

## Author Comment (AC2)

1. **Original Submission**

**1.1. Recommendation**

Major Revision

1. **Comments to Author:**

**Correlation between marine aerosol optical properties and wind fields over remote oceans with use of spaceborne lidar observations**

**Overall opinion:** The paper elucidates the relationship between marine aerosol optical depth (AOD) and near surface wind speeds using Aeolus. In short, you demonstrated that Aeolus can unveil positive relationship between aerosol optical properties and wind speed over oceans efficiently. Aeolus can resolve gradient of the aerosol optical variations depending on the wind speed as well. The differences between marine boundary layer and the layer above can be unveiled nearly exclusively by relying on Aeolus, which is a promising finding considering low signal to noise ratio of Aeolus. However, your current effort should be carefully revised because of poorly justified methodological choices, ambiguities (and, even more critically, inconsistency) in the research aim, introduction gaps, and text that is unfriendly to general readers. Most notably, some results are not persuasive; the relationship between marine AOD and wind speed has not been quantified in some cases, and only displayed in figures and discussed in the text in others without trends being quantified or correlations/causal connections being reported. Conclusions are just short and plain version of results (no summary, no overview, no holistic opinion presented in conclusions). Further details are outlined below.

AR: Thank for your careful review. According to your suggestions, we have revised the manuscript carefully. We have explained and modified the methodology, clarified the research scope as much as possible. Besides, the gradients of marine aerosol optical properties with wind speed (shown in the panel (c) and (d) of Fig. 8, Fig. 9 and Fig. 10) are supplemented in the revised manuscript to quantify the variation tendency. The abstract, result section and the conclusion section were rephrased to quantify the results and to clarify the research scope and the conclusion. The point-to-point responses are shown below.

**2.1. Comments:**

1. **Abstract:** Although the paper is submitted to the special issue dedicated to Aeolus, I think you should foremost think about general readers. Please emphasize your research aim boldly, ensure this research aim agrees with what you state in the end of the introduction and in the end of the paper. Logically, there is little sense to start your abstract by introducing satellite-specific terms such as Level 2A product. At least, please mention Aeolus first. Ideally, state in the first sentence of the introduction why you think marine aerosol – wind speed relationship is important and Aeolus is a good choice to address it.

AR: Thanks. According to your advice, we have stated in the first sentence why marine aerosol – wind speed relationship is considered important and Aeolus is a good choice to address it, and removed the introduction of satellites' products. The revised abstract is shown as below:

"**Abstract.** Marine aerosol is mainly produced by wind, which is also a vital element impacting the transport, evolution and dissipation of marine aerosol. The understanding of the accurate relationships between marine aerosol optical properties and wind speed will improve the global aerosol transport models, the satellite-retrieved AODs, the atmospheric correction of ocean color and the study of biogeochemical cycles. Aeolus, the worldwide first ever wind detection lidar satellite, had the ability to measure wind information and particulate optical properties simultaneously, which provide the opportunity to explore the absolutely synchronous relationships between marine aerosol optical properties and wind speeds.  Furthermore, thanks to the Aeolus measurement of vertical profiles, the relationships can be discussed in different vertical layers. In this paper, utilizing Aeolus data, the relationships between the optical properties at 355 nm of marine aerosol and the corresponding instantaneous co-located wind speeds of three remote ocean areas are explored and discussed at two sperate vertical atmospheric layers (0-1 km and 1-2 km, correspond to the heights within and above marine atmospheric boundary layer (MABL)), revealing the marine aerosol related atmospheric background states. The marine aerosol extinction/backscatter coefficients and the background wind speeds show positive relationships and they were fitted by power law functions, of which the corresponding $R^2$ are all higher than 0.9. Both the MABL and the higher layer above the MABL will receive the marine aerosol produced and transported by the wind from the air-sea interface. The marine aerosol load at the lower layer (MABL) is stronger than at the higher layer. The marine aerosol enhancements caused by the background wind are more intensive at the MABL. The gradient change points of marine aerosol extinction/backscatter coefficients appear during the growth of them with wind speed, above which the growth rate becomes lower. It might illustrate that the enhancement of marine aerosol driven by wind includes two phases, among which one is rapid growth phase with high dependency of wind, and another is slower growth phase after the gradient change points. As derived data from Aeolus, the averaged marine aerosol optical depth along with wind speed is acquired and utilized to verify the results by the comparison with CALIPSO retrieved results reported in previous work, and besides, the averaged marine aerosol lidar ratio at 355 nm along with wind speed is discussed for the relationship between marine aerosol particle size and wind speed."

2. **Introduction:** Three problems here.

o First, there is an extreme ambiguity on which aspect of AOD_mar and wind speed relationship you want to address. You include so-called highlights, which actually erode the clarity of your research aim. State directly: do you want to examine whether there is a relationship between AOD_mar and wind speed using Aeolus? If yes, say it boldly and underpin all highlights (i.e., objectives) to this research aim please.

AR: The basic research aim is to explore the relationship between marine aerosol optical properties (extinction coefficient, backscatter coefficient, lidar ratio) and wind speed within two vertical layers using Aeolus. As introduced in the introduction, almost all the previous studies mainly focused on the layer integrated optical properties (AOD) and ocean surface wind speed.

This study is going to explore the relationship between the vertical marine aerosol optical properties and the corresponding spatiotemporally synchronous wind speed using vertical profiles of Aeolus measurements, which could represent the marine-atmospheric aerosol background state and may reveal the transport and evolution of the marine aerosol vertically. To clarify the research aim, the highlights has been revised as:

"Generally, the highlights of this work mainly include 1) acquiring the spatiotemporally synchronous relationship between the aerosol optical properties (extinction coefficient, backscatter coefficient, lidar ratio) and the instantaneous wind speeds, which could indicate the background atmosphere states within and above the MABL over remote ocean, 2) conducting analysis at two separate height layers above ocean surface to explore the vertical differences in aspect of the wind-drive marine aerosol evolution."

o  Second, it looks like you placed everything what is related to AOD_mar – wind speed relationship, Aeolus and CALIPSO aerosol observations of aerosol together, but in barely comprehensible logic. To be more specific, your introduction is neither centered over your research aim, nor logically guides a reader to this research aim paragraph by paragraph though showing important milestones and gaps made in this research field. In other words, it is related to the study topic, but chaotically structured.

AR: Thanks for your comments. Actually, it is considered that the introduction section was organized by guiding readers from research significance to research aim paragraph by paragraph.

The first paragraph states the necessity of studying marine aerosol.

The second paragraph introduces the significance of marine aerosol-wind relationship, then by analyzing some previous studies in this field, the shortcomings of those, mainly in the measurement of marine aerosol optical properties, are summarized in this paragraph as: for the passive measurements, "The passive instruments lack the abilities of distinguishing marine aerosol from other aerosols, acquiring vertical profiles of aerosols, and retrieving aerosol optical properties without sunlight (except for lunar-photometer) and under cloudy conditions (Kiliyanpilakkil and Meskhidze, 2011; Winker and Pelon, 2003)."; for the active measurements (lidar), the shipborne lidar's measurements can not be representative for the global ocean results while CALIOP measurements are restricted by the assuming lidar ratio. So, the key notes and what can be improved in the marine aerosol-wind speed relationship exploration were summarized as: "In summary, to explore the accurate relationship between the marine aerosol optical properties and the wind speed, it is essential to conduct global continuous observations and obtain the information of aerosol type identification, while vertical profiles of aerosols can provide extra spatial information for further analysis. Moreover, previous studies mostly focused on the layer $AOD_{mar}$ and ocean surface wind speed, exploring the probable production of marine aerosol driven by surface wind. The relationship between the vertical marine aerosol optical properties and the corresponding spatiotemporally synchronous wind speed is still to be investigated, which represents the marine-atmospheric background state and may reveal the transport and evolution of the marine aerosol vertically."

To overcome the shortcomings and make some progress in the exploration of the relationship between marine aerosol optical properties and wind speed, the third paragraph introduces a spaceborne instrument, ALADIN, firstly used in this field, which had the ability to measure global aerosol and wind field vertical profile simultaneously. The fourth paragraph describes the data and the method used in this paper briefly and states the highlights of this work. It is considered that the third and the fourth paragraph raised a new approach which will solve some previous shortcomings in this study field.

To conclude, the framework of the introduction section was organized as: introduce the study subject (marine aerosol, the relationship between marine aerosol optical properties and wind speed), summarize the shortcomings of the previous studies, raise a new approach trying to overcome the shortcomings and make some progress.

*Reference:*

*Kiliyanpilakkil, V. P. and Meskhidze, N.: Deriving the effect of wind speed on clean marine aerosol optical properties using the A-Train satellites, Atmos. Chem. Phys., 11, 11401–11413, https://doi.org/10.5194/acp-11-11401-2011, 2011.*

*Winker, D. M. and Pelon, J.: The CALIPSO mission, Geoscience and Remote Sensing Symposium, IGARSS '03, Proceedings, IEEE International, 2, 1329-1331, https://10.1109/IGARSS.2003.1294098, 2003.*

o  Third, you discuss marine aerosol optical properties – near wind relationship from lidar perspective, but omitted a large corpus of works dedicated to this issue. In particular, Josset et al. (2008) demonstrated that there is inverse relationship between wind speed and surface attenuated backscattering; while both these parameters have direct link to aerosol optical depth and this paper shows the formula how they are linked. Further works of Josset et al. (2008, 2010, 2018); Hu et al. (2008), Venkata and Reagan (2016) have elucidated this relationship in detail for CALIPSO and there were pre-launch Aeolus works on this topic such as Li et al. (2010). Moreover, some yet unfinalized studies, but ongoing efforts of Labzovskii et al. (2022) and Dionisi et al. (2022) addressing ocean surface-aerosol optical properties-wind interplay can be found as conference proceedings. By omitting all these CALIPSO and Aeolus-focused efforts, you hint that you are not aware about a hidden fundamental link between AOD and wind speed over oceans if your AOD is calculated using lidars. This can be a pitfall and a single point of failure for your methodology if this link does exist. Please check all the aforementioned works (I provided references in the end of this document) and incorporate their experience in your introduction, where the interpretation is up to you indeed.

AR: Thanks for the recommendation of these works. The research objective of this study is to explore the relationship between marine aerosol optical properties and wind speed using Aeolus, based on the physical principle that marine aerosol is produced and developed by the drive of wind. We have reviewed all of them and it is found that all indeed focus on spaceborne lidar

(CALIPSO, Aeolus) and are about ocean, but none of them are related to the relationship between marine aerosol optical properties and wind speed.

The three works of Josset et al. (2008, 2010, 2018) mainly talked about the aerosol optical depth retrieval method using ocean surface or land surface echoes. Among them, in Josset et al. (2008) the inverse relationship is between wind speed and surface attenuated backscattering. The surface attenuated backscattering is a parameter representing the lidar echo intensity of ocean surface and is totally distinct from atmospheric aerosol optical properties (extinction coefficient, backscatter coefficient, AOD, etc.). Hu et al. (2008), Venkata and Reagan (2016) applied this relationship, conducting the sea surface wind speed estimation from CALIPSO's ocean surface backscatter (Hu et al., 2008) and retrieving aerosol from CALIPSO's ocean surface returns (Venkata and Reagan, 2016), respectively. Though Li et al. (2010) focused on Aeolus and also used this relationship, it mainly cared about the sea surface reflectance for different incidence angles. For the recent works about Aeolus, Labzovskii et al. (2022) analyzed the sensitivity of Aeolus Lidar Surface Returns (LSR) to the types of surface, of which the research objective was exploring the retrieval of AOD using LSR over ocean, while Dionisi et al. (2022) was going to evaluate and document the feasibility of deriving an in-water prototype product from the analysis of the signal acquired Aeolus.

Thanks again for your recommendation. These works definitely broaden our horizons. As these works have little link with the subject of this manuscript (the relationship between marine aerosol optical properties and the corresponding spatiotemporally synchronous wind speed), it is decided that they are not quoted in the manuscript.

3. **Methodology:** This part of the manuscript is a potential pitfall as well. In particular, you have introduced your own framework of marine aerosol domination, cloud screening and wind speed-aerosol optical property analysis. However, nearly every stage of the framework you showed on Figure 1 should be justified because you made numerous debatable assumptions. The assumptions about effective ability to classify marine aerosol using CALIPSO to be applied for Aeolus, assumption about efficiency of cloud screening based on Rayleigh channel information of Aeolus. I will raise the following issues

o   According to 2.1, it looks like you used the wind speed from official Aeolus observational product. However, Figure 2 implies that you used AUX_MET winds from NWP/ECMWF simulations and L2C data. Clarify this aspect in every section please.

AR: We used the wind speed from official Aeolus Level 2C wind vector products, which are the outputs from the assimilation of the Aeolus Level 2B products (observational wind) in the ECMWF numerical weather prediction (NWP) operational model after 9 January 2020. Actually, the "NWP model winds from Level 2C product" means we used model winds from ECMWF provided in the L2C data, which are assimilated with observational wind.

In Section 2.1, we describe the Level 2C product of Aeolus as "It should be emphasized that Level 2C wind vectors are the outputs from the assimilation of the Aeolus Level 2B products in the ECMWF numerical weather prediction (NWP) operational model after 9 January 2020

(Rennie et al., 2021)." And for what data we used in this study, it was stated that "As mentioned above, we use Level 2A and Level 2C products of Aeolus for the study of the relationship between marine aerosol optical properties and wind speeds."

In Section 3, to clarify this aspect, the relevant description has been revised as "As for the wind vector data, Aeolus Level 2C product provides the $u$ component (zonal components of wind vector) and $v$ component (meridional components of wind vector) from the ECMWF model after assimilation of Level 2B observational wind product, at the same data bins of the Level 2A optical properties product."

*Reference:*

*Rennie, M. P., Isaksen, L., Weiler, F., de Kloe, J., Kanitz, T., and Reitebuch, O.: The impact of Aeolus wind retrievals on ECMWF global weather forecasts, Q. J. Roy. Meteor. Soc., 147, 3555–3586, https://doi.org/10.1002/qj.4142, 2021.*

o   First, you may describe the methodology and only then, introduce Figure 1. In most cases, the description of a figure comes before the figure itself. Moreover, think about a reader, without reading a text, he/she might be easily confused by reading terms and concepts that you not introduced, nor described yet.

AR: Thanks for the advice, revised.

o   You do not need to repeatedly use the term 'aerosol optical depth' as the acronym AOD is quite common. For instance, you keep using the term 'aerosol optical depth' without acronym even in Figure 1.

AR: Thanks for the advice. Besides the term "aerosol optical depth" in Figure 1, we have checked the whole manuscript and replaced the unnecessary "aerosol optical depth" with "AOD".

o   Once you state that SCA product is more robust than other algorithms, either mention these algorithms directly or just state that SCA product is robust.

AR: Thanks for the advice. The sentence has been revised as "Extinction coefficient at 355 nm and backscatter coefficient at 355 nm retrieved by the standard correction algorithm (SCA) from Aeolus Level 2A product are used in this study, as the SCA processing is capable to produce more stable extinction coefficient and backscatter coefficient than the Mie channel algorithm (Flament et al., 2021)."

*Reference:*

*Flament, T., Trapon, D., Lacour, A., Dabas, A., Ehlers, F., and Huber, D.: Aeolus L2A aerosol optical properties product: standard correct algorithm and Mie correct algorithm, Atmos. Meas. Tech., 14, 7851–7871, https://doi.org/10.5194/amt-14-7851-2021, 2021.*

○ Cloud screening. The presence of undetected clouds can severely plague your clear sky assumptions. How did you ensure that this screening strategy worked well? I did not notice any evaluation or statistical analysis of "cloud-free" and "cloud-containing" layers in your paper. Thus, the efficacy of this approach is questioned. Moreover, why one needs to use molecular optical depth and not Mie backscattering or extinction product (fundamentally more sensitive to thin clouds or liquid water clouds than Rayleigh product) directly to detect clouds at various heights? Let alone, you said that the SCA product is stable and you can confidently rely on these products.

AR: Sorry for that the vague sentence "The relative humidity (RH) and molecular backscatter coefficient of each data bin from the NWP model of ECMWF are provided in the Level 2A product and are utilized to screen the cloud layers." misled you. This sentence has been revised as "Aeolus measured particulate $\beta$, combined with relative humidity (RH) and molecular $\beta$ from the ECMWF NWP model provided in the Level 2A product are utilized to screen the cloud layers." in the revised manuscript.

Actually, we used backscatter ratio (BR), which is the ratio of total $\beta$ and molecular $\beta$, as the criteria to conduct cloud screening. Specifically, when the BR is larger than 2.5, the data bin is considered cloud contaminated and is eliminated. This approach was proposed in the Aeolus Level -2A Algorithm Theoretical Basis Document, i.e., Flament et al. (2021). It is indeed principally based on the Mie backscattering (particulate $\beta$), as Mie backscattering is fundamentally more sensitive to thin clouds or liquid water clouds while molecular $\beta$ depending on temperature and pressure at various heights is relatively stable. Therefore, the approach is regarded available for cloud screening.

Moreover, as you advised, we supplemented the statistical analysis of "cloud-free" and "cloud-containing" layers in the revised manuscript as "With this cloud screening approach, in this study, 9%, 35%, 40% data in the altitude range of 0-2 km was eliminated for the NP area, the SP area and the SI area, respectively." to support the feasibility of this approach.

*Reference:*

*Flament, T., Trapon, D., Lacour, A., Dabas, A., Ehlers, F., and Huber, D.: Aeolus L2A aerosol optical properties product: standard correct algorithm and Mie correct algorithm, Atmos. Meas. Tech., 14, 7851–7871, https://doi.org/10.5194/amt-14-7851-2021, 2021.*

○ Eliminating outliers. Okay, you referenced the paper from 1986 to justify statistical filtering of outliers. However, how did you ensure that in the particular case of Aeolus data, you have not filtered useful data using this particular statistical filter? I mean statistical filtering is helpful for sure, but only when you can understand when to apply this statistical filtering from physical point of view. Thus, please justify physical aspect of this choice here. The same is applied to outlier removal step introduced in Lines 368 – 375.

AR: This approach of eliminating outliers referred from Hoaglin et al. (1986) is the same as one used for boxplot. It is a widely used approach in data analysis, especially in the statistical analysis.

It can identify outliers from the dataset which are meaningless and will affect the statistical analysis results. For this study, though the valid aerosol extinction coefficients and backscatter coefficients from Aeolus were selected with the quality control flags, there will still be outliers that are unreasonable values. Before the elimination, the outliers of extinction coefficients and backscatter coefficients can catch up to 1000 $\mathrm{Mm^{-1}}$ and 30 $\mathrm{Mm^{-1} \cdot sr^{-1}}$ while generally the particulate extinction coefficients and backscatter coefficients are within 300 $\mathrm{Mm^{-1}}$ and 10 $\mathrm{Mm^{-1} \cdot sr^{-1}}$. Therefore, it is considered essential to eliminate these unreasonable values by the Tukey's test method.

*Reference:*

*Hoaglin, D. C., Iglewicz, B., and Tukey, J. W.: Performance of some resistant rules for outlier labelling, Journal of the American Statistical Association, 81(396), 991-999, https://doi.org/10.1080/01621459.1986.10478363, 1986.*

o   Why the title of 2.1 is "ALADIN/Aeolus" with right slash? Give more comprehensible name to the section please. Same applies for 2.2.

AR: The titles of Section 2.1 and Section 2.2 have been revised as "ALADIN" and "CALIOP".

o   (1) You introduce a depolarization correction of backscattering based on the assumption that CALIPSO can ideally detect marine aerosol, but this is not the case. What if you just introduced positive bias in many cases of your analysis by assuming their marine nature, while their depolarization was not typical for marine aerosol cases?

AR: Actually, for the defined "marine aerosol dominates areas" in this manuscript, there are a few terrestrial aerosols like dust, polluted dust, polluted continental and smoke, with the total proportion of no more than 10%, while among them the depolarization ratios at 355 nm of dust and polluted dust are 0.22-0.24 and 0.16 respectively, much larger than $\delta_{mar,355nm}$. Consequently, regarding all the aerosols as marine aerosol and correcting $\beta_{mar}$ by formula (1) leads to the obvious underestimation of the $\beta$ for dust and polluted dust. Nevertheless, in view of the small proportions of dust (no more than 3.15%) and polluted dust (no more than 0.79%) above the study areas and thanks to the statistical analyses of data for a long term, the assumption that regarding all the aerosols as marine aerosol will have little impact on the statistical analyses between $\beta_{mar}$ and wind speed and is considered acceptable. The comments of this issue have been supplemented in Section 3 of the revised manuscript, as presented below: "It should be illustrated that all the aerosol $\beta$ s from Aeolus identified as $\beta_{mar,Aeolus-co}$ s and then utilized to calculate $\beta_{mar}$ s by formula (1) is under the ideal assumption that marine aerosol is the only aerosol type in the study areas. Though the study areas are all located in the remote ocean far away from land and are evaluated as "marine aerosol dominate" by CALIOP, there are a few terrestrial aerosols like dust, polluted dust, polluted continental and smoke, with the total proportion of no more than 10% (see Section 4.1 for the detail). For the part of terrestrial aerosols, the depolarization ratios at 355 nm of them are 0.22-0.24 for dust, 0.16 for polluted dust, 0.01 for polluted continental and 0.03 for

smoke, among which the dust's and the polluted dust's are much larger than $\delta_{mar,355nm}$ (Floutsi et al., 2023). Consequently, regarding all the aerosols as marine aerosol and correcting $\beta_{mar}$ by formula (1) leads to the obvious underestimation of the $\beta$ for dust and polluted dust. Nevertheless, in view of the small proportions of dust (no more than 3.15%) and polluted dust (no more than 0.79%) above the study areas and thanks to the statistical analyses of data for a long term, the assumption that regarding all the aerosols as marine aerosol is considered not to critically impact the $\beta_{mar}$ - wind speed relationship, while it should be noticed that the actual $\beta$ is a little bit larger than the $\beta_{mar}$ ."

*Reference:*

*Floutsi, A. A., Baars, H., Engelmann, R., Althausen, D., Ansmann, A., Bohlmann, S., Heese, B., Hofer, J., Kanitz, T., Haarig, M., Ohneiser, K., Radenz, M., Seifert, P., Skupin, A., Yin, Z., Abdullaev, S. F., Komppula, M., Filioglou, M., Giannakaki, E., Stachlewska, I. S., Janicka, L., Bortoli, D., Marinou, E., Amiridis, V., Gialitaki, A., Mamouri, R.-E., Barja, B., and Wandinger, U.: DeLiAn – a growing collection of depolarization ratio, lidar ratio and Ångström exponent for different aerosol types and mixtures from ground-based lidar observations, Atmos. Meas. Tech., 16, 2353–2379, https://doi.org/10.5194/amt-16-2353-2023, 2023.*

o   Potential inaccuracies due to assumptions (not calculations or objective information) about MABL of ~1 km are not discussed.

AR: Thank you for your comment. In the revised manuscript, besides the MABL height of around 1 km summarized from several references, the mean MABL height values of $787.47 \pm 231.77$ m at the NP area, $939.39 \pm 360.20$ m at the SP area and $1005.29 \pm 366.60$ m at the SI area are calculated from ECMWF provided boundary layer height as extra argument to support the MABL height. The MABL heights are variable and thus set as 1 km will lead to the potential inaccuracies. However, restricted by the relatively low height resolution of Aeolus (0.25 km below 0.5 km, 0.5 km in the range of 0.5 km to 2 km), utilizing more precise height boundaries won't make more sense. Therefore, it is considered that the statistical results of the 0-1 km layers and the 1-2 km layers are capable to generally represent the atmospheric conditions within the MABL and above the MABL.

The discussion about MABL heights are revised in the manuscript as:

"Referring the results of Luo et al. (2014), Luo et al. (2016) and Alexander et al. (2019), the MABL height of the remote ocean is summarized as around 1 km. Moreover, calculated with ECMWF provided boundary layer heights at the three study areas for the time period of 20 April 2020 to 26 May 2021, the mean values and the standard deviations are $787.47 \pm 231.77$ m at the NP area, $939.39 \pm 360.20$ m at the SP area and $1005.29 \pm 366.60$ m at the SI area. Hence, the boundary height of the two vertical layers is set as 1 km, approximately corresponding to the mean MABL height of remote ocean. Though the MABL heights are variable and thus set as 1 km will lead to the potential inaccuracies, restricted by the relatively low height resolution of Aeolus (0.25 km below 0.5 km, 0.5 km in the range of 0.5 km to 2 km), utilizing more precise

height boundaries won't make more sense. It is considered that the statistical results of the 0-1 km layers and the 1-2 km layers are capable to generally represent the atmospheric conditions within the MABL and above the MABL."

*Reference:*

*Alexander, S. P. and Protat, A.: Vertical profiling of aerosols with a combined Raman-elastic backscatter lidar in the remote Southern Ocean marine boundary layer (43–66°S, 132–150°E), J. Geophys. Res.-Atmos., 124, 12107–12125, https://doi.org/10.1029/2019JD030628, 2019.*

*Luo, T., Yuan, R., and Wang, Z.: Lidar-based remote sensing of atmospheric boundary layer height over land and ocean, Atmos. Meas. Tech., 7, 173–182, https://doi.org/10.5194/amt-7-173-2014, 2014.*

*Luo, T., Wang, Z., Zhang, D., and Chen, B.: Marine boundary layer structure as observed by A-train satellites, Atmos. Chem. Phys., 16, 5891–5903, https://doi.org/10.5194/acp-16-5891-2016, 2016.*

o LayerL and LayerH are counterintuitive terms. At least, LayerL could be referred as Layer$_{MABL}$ for clarity throughout the text. Explain if I am wrong here and missing some intuitive links with L and H letters.

AR: The letter L of Layer$_L$ represents "lower", indicating it is the lower layer, of which the altitude range is 0-1 km. Likewise, the letter H of Layer$_H$ represents "higher", indicating it is the higher layer, of which the altitude range is 1-2 km. To clarify this issue, the relevant description in the manuscript has been revised as "The lower layer with the altitude range of 0 km to 1 km is called Layer$_L$ in this paper and the higher layer with the altitude range of 1 km to 2 km is called Layer$_H$."

4. **Results:**

o Some information you placed into the Results obviously fits the methodological description in more logical way (see the lines from the start of Results 4 to Line 260). Lines 368 – 370 as well, where you talk about elimination of statistical outliers; this should be explained in the methodology not in the middle of the results section.

AR: Thanks for the suggestion. The study areas selection part has been moved to the second paragraph of the methodology section. And the wind direction analysis part was removed according to the third comment of results. The detailed explanation can be found in the response there. Likewise, the average calculation part including the outlier elimination part was moved to the last paragraph of the methodology section.

o Some terminological problems are visible. For instance, you say "it is considered that the Aeolus retrieved extinction and backscattering area reasonable". First, considered by whom? Second, what is "reasonable" from physical point of view? (Line 305 and above). This aspect is better clarified in Line 315.

AR: We compare Aeolus retrieved $\alpha_{mar}$ and $\beta_{mar}$ with converted typical $\alpha_{mar}$ and $\beta_{mar}$ ranges at 355 nm, calculated from the typical marine aerosol optical properties' ranges at 532 nm reported by Prijith et al. (2014), Kiliyanpilakkil and Meskhidze (2011) and the typical conversion coefficients, i.e., Ångström exponents reported by Floutsi et al. (2023). The data sources for the comparison are published and recognized, thus compared to the typical ranges, it is considered that the Aeolus retrieved extinction and backscattering area reasonable. The sentence has been revised, and we think it is clear if one combines this sentence with several previous sentences, which are shown as below:

"It is reported that the typical ranges of $\alpha_{mar}$ and $\beta_{mar}$ at 532 nm over remote ocean areas are around 60 $Mm^{-1}$ to 80 $Mm^{-1}$ and around 1 $Mm^{-1} \cdot sr^{-1}$ to 5 $Mm^{-1} \cdot sr^{-1}$, respectively, observed and retrieved by CALIOP (Prijith et al., 2014; Kiliyanpilakkil and Meskhidze, 2011). Applying the typical $\alpha_{mar}$ Ångström exponent from 532 nm to 355 nm of $0.7 \pm 1.3$ and the typical $\beta_{mar}$ Ångström exponent from 532 nm to 355 nm of $0.8 \pm 0.1$ (Floutsi et al., 2023), the converted typical ranges of $\alpha_{mar}$ and $\beta_{mar}$ at 355 nm can be calculated, which are around 47 $Mm^{-1}$ to 180 $Mm^{-1}$ and around 1.3 $Mm^{-1} \cdot sr^{-1}$ to 7.2 $Mm^{-1} \cdot sr^{-1}$. Compared with the typical ranges of $\alpha_{mar}$ and $\beta_{mar}$ at 355 nm, calculated from CALIOP retrieved typical ranges of marine aerosol optical properties and the typical conversion coefficients, it is considered that the Aeolus retrieved $\alpha_{mar}$ and $\beta_{mar}$ are reasonable."

*Reference:*

*Floutsi, A. A., Baars, H., Engelmann, R., Althausen, D., Ansmann, A., Bohlmann, S., Heese, B., Hofer, J., Kanitz, T., Haarig, M., Ohneiser, K., Radenz, M., Seifert, P., Skupin, A., Yin, Z., Abdullaev, S. F., Komppula, M., Filioglou, M., Giannakaki, E., Stachlewska, I. S., Janicka, L., Bortoli, D., Marinou, E., Amiridis, V., Gialitaki, A., Mamouri, R.-E., Barja, B., and Wandinger, U.: DeLiAn – a growing collection of depolarization ratio, lidar ratio and Ångström exponent for different aerosol types and mixtures from ground-based lidar observations, Atmos. Meas. Tech., 16, 2353–2379, https://doi.org/10.5194/amt-16-2353-2023, 2023.*

*Kiliyanpilakkil, V. P. and Meskhidze, N.: Deriving the effect of wind speed on clean marine aerosol optical properties using the A-Train satellites, Atmos. Chem. Phys., 11, 11401–11413, https://doi.org/10.5194/acp-11-11401-2011, 2011.*

*Prijith, S. S., Aloysius, M., and Mohan, M.: Relationship between wind speed and sea salt aerosol production: A new approach, Journal of Atmospheric and Solar-Terrestrial Physics, 108, 34-40, https://10.1016/j.jastp.2013.12.009, 2014.*

o   You dedicate considerable efforts to prove that your aerosol in ocean zones has marine/ocean origin. For instance, you speculate about winds in MABL and above. Why CALIPSO classification is not enough as methodological choice to determine once and for good that you have marine aerosol? You are eroding your research scope by devoting too much efforts to prove this point in the results, not in the methodology.

AR: Thanks for the advice. CALIPSO classification was considered enough as methodological choice to prove the "marine aerosol dominates areas". The original thought was that the wind directions were regarded as the assist evidences for the "marine aerosol dominates areas". But after discussion, we thought this part was an indirect and weak clue to the objective, and as you mentioned, this part eroded the research scope. Therefore, we decided to remove the wind direction part to maintain the concentration of the research scope.

o The style of reporting lacks references to certain figures, which hampers review process. You added three figures together in a row (4 – 6) and each consists of multiple panels. In this case, it is not helpful to refer to figures like "From Figures 4 – 6 you can see…" (line in the case of Line 316).

AR: Thanks for the suggestion. This sentence has been revised as "Figure 4, Fig. 5, and Fig. 6 presents the parameters distributions at two layers above the NP area, the SP area and the SI area".

o Line 316. I do not see this similarity qualitatively. Please try to use quantitative terms or more explicit qualitative description of similarity.

AR: The description that "have several similar features" is considered unspecific for the three figures including 24 panels totally. It is thought meaningless and was removed in the revised manuscript.

o Line 325 Numerical reference to what is "evident high wind speed region" is missing. Use numbers or direct references to figures here and elsewhere.

AR: Thanks for the suggestion. To clarify the description of "evident high wind speed region", the sentences have been revised as "Referring to $Layer_H$, shown in the upper four panels of Fig. 4, Fig. 5 and Fig. 6, it can be found that the spatial variation trends of $ws$, $\alpha_{mar}$, $\beta_{mar}$ in the three areas are alike with those at $Layer_L$. The evident high wind speed regions, where the wind speeds are up to around 8-10 $m \cdot s^{-1}$ in 5°N to 20°N of the NP area, 15-18 $m \cdot s^{-1}$ in 40°S to 60°S of the SP area and 13-19 $m \cdot s^{-1}$ in 35°S to 60° of the SI area, also exist at $Layer_H$ while $\alpha_{mar}$ and $\beta_{mar}$ are slightly enhanced in these regions, which indicates that the wind speed may still have weak positive influence on the marine aerosol optical properties at the higher atmosphere layer above the MABL."

o Section 4.3 Use quantitative metrics while talking about such phenomena as increasing tendency. When I am looking at Figures 8 and 9, what we need here are rather: statistical agreement metrics (correlation or anything similar), metrics of statistical significance if this agreement exists, trend metrics. From first glance, I do not see any correlation for panels a, b, c. Perhaps, some correlation at panel d, but you did not articulate it.

AR: Actually, the previous thoughts of Fig. 8, 9 and 10 were to only present the data value distributions of marine aerosol optical properties and wind speed with 2-D histogram. That's why we only discussed about the data value ranges in the first paragraph of Section 4.3. It was

considered that the 2-D histograms were not intuitive enough for the trend analyses. Consequently, there were no statistical agreement metrics, metrics of statistical significance and trend metrics in this part. The statistical agreement metrics and trend metrics are analyzed adequately in Fig. 11, 12 and 13, which also provide the wind speed data count distributions by the histogram. To conclude, in the aspect of trend metric, Fig. 8, 9 and 10 were not that intuitive as Fig. 11, 12 and 13; in the aspect of data value distribution, Fig. 11, 12 and 13 can also provide sufficient information. Therefore, it is considered that the information carried by Fig. 8, 9 and 10 is limited, while retaining these figures are redundant and repetitive. In the revised manuscript, we decided to remove Fig. 8, 9, 10 and the relevant description, then the previous Fig. 11-Fig. 16 were updated to Fig. 8-Fig. 13 consequently.

o   Lines 403 – 405. You present an unsupported hypothesis here where it is not possible to establish whether you are facing the lack of wind data of > 15 m/s, some other physical phenomena such as response of ocean surface backscatter to stronger winds, which affects AOD in the end or even something else. Such unsupported surmises are not advisable for journals, focused on atmospheric physical phenomena.

AR: Thanks for your reminding. We did the gradient analyses of marine aerosol extinction and backscatter with wind speed. The results were added in the revised Fig. 11 (new Fig. 8) shown as below. We think with the gradients as prove, the surmise that there might be two distinct variation trends of $\alpha_{mar}$ and $\beta_{mar}$ above or below the wind speed of 10 $\mathrm{m \cdot s^{-1}}$ can be guessed. The argument and the statement have been revised as "Referring to the panel (c) and (d) of Fig. 11, within the same wind speed interval, the gradient at $\mathrm{Layer_L}$ is larger than that at $\mathrm{Layer_H}$, i.e., the optical properties at $\mathrm{Layer_L}$ will increase more rapidly with wind speed. It is worth to notice that for the case that the wind speed is above 10 $\mathrm{m \cdot s^{-1}}$, the gradients of $\alpha_{mar}$ and $\beta_{mar}$ seem to show decreasing tendencies, whereas under the condition when the wind speed is lower than 10 $\mathrm{m \cdot s^{-1}}$, the values of the optical properties' gradients present increasing tendencies, indicating the better fitting by power law functions at lower wind speed. This phenomenon may imply that there might be two distinct variation trends of $\alpha_{mar}$ and $\beta_{mar}$ above or below the wind speed of 10 $\mathrm{m \cdot s^{-1}}$."

[Figure]

**Figure 11: Relationship between marine aerosol optical properties ((a) for $\alpha_{mar}$, (b) for $\beta_{mar}$) and wind speed above the NP area. The blue circles and error bars represent the means and standard deviations of the optical properties along wind speed grids at $\text{Layer}_\text{L}$, while the reds represent the same items at $\text{Layer}_\text{H}$. The blue and red dotted-dashed lines are the optical property averages regression curves fitted along the wind speed grid at $\text{Layer}_\text{L}$ and $\text{Layer}_\text{H}$, respectively. The blue and red histograms indicate the data counts of every wind speed grid at $\text{Layer}_\text{L}$ and $\text{Layer}_\text{H}$, respectively. (c) and (d) represent the gradients of $\alpha_{mar}$ and $\beta_{mar}$ with wind speed.**

- o Lines 411 – 429 In this paragraph, you quantify only wind speed values, not optical properties of aerosols. I do not see any value in these speculations of optical properties of aerosols becoming "larger" or "smaller" at certain wind speed intervals without (1) strong quantitative arguments about relationship of these optical properties to wind speeds, (2) consistent, centered narration around the pattern you identified. Moreover, it is unclear why you analyzed both extinction and backscattering, it does not seem that you make any difference between these parameters. Neither in the way treating them, nor in making conclusions, you just report that they change in some way with wind. Without quantitative arguments of relationship with wind, without explicit references to figures, where you noticed these patterns, why would it matter after all?

AR: Thanks for your comments. We have rephrased this section with more strong quantitative arguments and modified with consistent, centered narration in the revised manuscript. We have supplemented the gradients of extinction and backscattering in the panel (c) and (d) of Fig. 8, Fig. 9 and Fig. 10 to describe the variation tendencies quantitatively. Extinction and backscattering are two most typical optical properties measured by lidar, which represents the attenuation property on light and the scattering property on light of aerosol, respectively. The objective of analyzing both of them is to try to establish extinction-wind speed and backscattering-wind speed relationship as inputs of the radiation transfer model.

o Line 468. Once again, Josset et al. [2008] have examined relationship between WS (AMSR-E from A-Train) and aerosol optical properties from collocated CALIPSO observations. Only over ocean. Please consider this point while writing your elaborations on WS-AOD_mar interplay here.

AR: The interplay of wind speed and AOD_mar intended to explore in this study is mainly based on the physical principle that the marine aerosol is produced and developed by the drive of wind. The specific relationships are between marine aerosol optical properties (extinction, backscatter, AOD and lidar ratio) and wind speed. The marine aerosol optical properties are calculated based on the atmospheric backscatter lights of lidar beams, independent of wind speed. Nevertheless, the relationship mentioned in Josset et al. (2008) was a method to calculate AOD using wind speed as an input parameter. Specifically, the relationship was established between the normalized surface reflectance and wind stress (Cox and Munk, 1954), and then, as an input, the normalized surface reflectance can be conducted into the calculation of AOD. This AOD is retrieved using wind speed as an input parameter. Consequently, the research objective in Josset et al. (2008) was distinct with ours. We decided not to discuss their work.

o Line 508 Unsupported surmise about the presence of clouds. Arguments are needed here.

AR: Thanks. Regarding to the AOD comparison between Aeolus retrieved and CALIOP retrieved presented in Fig. 14 (Fig. 11 in the revised manuscript), the lower CALIOP AOD at low wind speed has been discussed for the reason that using fixed lidar ratio leads to the possible underestimation. Besides, the strict cloud screening strategy was conducted to avoid cloud contamination as introduced in Section 3. Therefore, the surmise that high Aeolus AOD results from the possible cloud contaminations are considered unsupported and unnecessary. We decided to remove the sentence "The slightly high Aeolus retrieved $AOD_{mar}$ may result from the possible cloud contaminations of the marine aerosol data bins." in the revised manuscript.

o Line 510 Quite late to introduce lidar ratio as you already spoke about it before. Please address the consistency of terminology and your acronyms here and elsewhere

AR: Thanks for the suggestion. The marine aerosol lidar ratio has been introduced in Section 3 so we decided to remove the first sentence of Section 4.4.2. Besides, the terminology "marine aerosol lidar ratio" was checked and replaced by "$LR_{mar}$" throughout the manuscript besides the first appearance.

o Figure 15. The deviations of LR from Aeolus are around 50%? For many types of data, such deviations make the quantification nearly meaningless. Your lidar ratio can jump from the values of 10 to >40.

AR:

According to the lidar observations, the pure marine aerosol LR at 355 nm can vary from 10 sr to 40 sr, while the simulated results showed that the pure marine aerosol LR can vary from 10 sr to 90 sr (Groß et al., 2013; Groß et al., 2015; Bohlmann et al., 2018; Floutsi et al., 2023; Masonis

et al., 2003). Though the deviations of LR from Aeolus are large and the LR can jump from the values of 10 to >40, they are considered in the reasonable range physically. The large LR standard deviations at each wind speeds may result from the fluctuations of marine aerosol LR. From Fig. 12 (Fig. 15 of the old version, shown as below), it can be found that the not only ALADIN, but the CALIOP retrieved LRs reported by Dawson et al. (2015) are also have large deviations. However, focusing the mean values of LRs, the tendencies are comparable and similar.

Moreover, some errorbars and even data points are omitted (see intersection of 40 LR and wind speed of 1 m/s).

Figure 12 (Fig. 15 of the old version) has been modified as below:

[Figure]

**Figure 12:** $LR_{mar}$ **versus the wind speed. The dark blue curve, red curve, yellow curve and the corresponding error bars represent the averaged** $LR_{mar}$ **and their standard deviations above the NP area, the SP area and the SI area, respectively. The purple curve and the corresponding error bars represent the CALIOP-retrieved** $LR_{mar}$ **at 532 nm (Dawson et al., 2015). The green curve and the light blue curve represent the modelled** $LR_{mar}$ **at 355 nm and at 532 nm, respectively (Sayer et al., 2012).**

*Reference:*

*Bohlmann, S., Baars, H., Radenz, M., Engelmann, R., and Macke, A.: Ship-borne aerosol profiling with lidar over the Atlantic Ocean: from pure marine conditions to complex dust–smoke mixtures, Atmos. Chem. Phys., 18, 9661–9679, https://doi.org/10.5194/acp-18-9661-2018, 2018.*
*Dawson, K. W., Meskhidze, N., Josset, D., and Gassó, S.: Spaceborne observations of the lidar ratio of marine aerosols, Atmos. Chem. Phys., 15, 3241–3255, https://doi.org/10.5194/acp-15-3241-2015, 2015.*
*Floutsi, A. A., Baars, H., Engelmann, R., Althausen, D., Ansmann, A., Bohlmann, S., Heese, B., Hofer, J., Kanitz, T., Haarig, M., Ohneiser, K., Radenz, M., Seifert, P., Skupin, A., Yin, Z., Abdullaev, S. F., Komppula, M., Filioglou, M., Giannakaki, E., Stachlewska, I. S., Janicka, L., Bortoli, D., Marinou, E., Amiridis, V., Gialitaki, A., Mamouri, R.-E., Barja, B., and Wandinger, U.: DeLiAn – a growing collection of depolarization ratio, lidar ratio and Ångström exponent for different aerosol types and mixtures from ground-based lidar observations, Atmos. Meas. Tech., 16, 2353–2379, https://doi.org/10.5194/amt-16-2353-2023, 2023.*

*Groß, S., Esselborn, M., Weinzierl, B., Wirth, M., Fix, A., and Petzold, A.: Aerosol classification by airborne high spectral resolution lidar observations, Atmos. Chem. Phys., 13, 2487–2505, https://doi.org/10.5194/acp-13-2487-2013, 2013.*

*Groß, S., Freudenthaler, V., Wirth, M., and Weinzierl, B.: Towards an aerosol classification scheme for future EarthCARE lidar observations and implications for research needs, Atmos. Sci. Lett., 16: 77-82, https://doi.org/10.1002/asl2.524, 2015.*

*Masonis, S. J., Anderson, T. L., Covert, D. S., Kapustin, V., Clarke, A. D., Howell, S., and Moore, K.: A study of the extinction-to-backscatter ratio of marine aerosol during the Shoreline Environment Aerosol Study, J. Atmos. Ocean. Tech., 20, 1388–1402, https://10.1175/1520-0426(2003)020<1388:ASOTER>2.0.CO;2, 2003.*

*Sayer, A. M., Smirnov, A., Hsu, N. C., and Holben, B. N.: A pure marine aerosol model, for use in remote sensing applications, J. Geophys. Res., https://10.1029/2011JD016689, 2012.*

o Line 529. You cannot say that similar results have been shown in previous studies. Rather, your study being chronologically newer, reports similar results, not the other way round. Moreover, specify the study you meant here directly.

AR: Thanks for the suggestion. The sentence has been revised as "The results reported in this paper are similar to those in the previous studies, of which Dawson et al. (2015) and Sayer et al. (2012) investigated the relationship between $LR_{mar}$ and wind speed utilizing measured $LR_{mar}$ and modelled $LR_{mar}$ respectively."

*Reference:*

*Dawson, K. W., Meskhidze, N., Josset, D., and Gassó, S.: Spaceborne observations of the lidar ratio of marine aerosols, Atmos. Chem. Phys., 15, 3241–3255, https://doi.org/10.5194/acp-15-3241-2015, 2015.*

*Sayer, A. M., Smirnov, A., Hsu, N. C., and Holben, B. N.: A pure marine aerosol model, for use in remote sensing applications, J. Geophys. Res., https://10.1029/2011JD016689, 2012.*

o Line 531 You already used term AMSR-E, but explain it here once again.

AR: Thanks, revised.

o Minor comments on this section you do not need to respond to, just consider this criticism while rewriting your results

AR: Thanks for the comments. We considered all of them while rewriting the results.

1. Avoid ambiguous terms like "explicit relationship"; stick to statistical, mathematical or physical terminology in such cases.

AR: Thanks. We have deleted the word "explicit".

2. Line 375 As mentioned in the comment about introduction, some previous studies have reported negative relationship between wind speed and aerosol optical properties given wind speed-AOD-surface backscattering fundamental relationship once we are dealing with water surface. The discussion on this aspect with the link to previous studies is missing in both introduction and results sections.

AR: As explained in the response to the comment about introduction, the marine aerosol optical properties – wind speed relationships are based on the physical principle that the production and the development of marine aerosol are driven by wind. The marine aerosol optical properties were retrieved from the atmospheric backscatter lights of lidar beams. The surface bins were eliminated to avoid ocean surface return signals. Nevertheless, the negative relationship between wind speed and aerosol optical properties in some previous studies was exactly using surface return signal to calculate AOD, while the ocean surface return signal is negative with wind speed. Therefore, the research objective of our study is distinct with theirs.

- Line 418 Growth rates become smaller is a dubious formulation. Did you mean growth rates slowed down? The same applies to "change points". What are "change points" in line 420? Same for "wind speed distribution ranges are larger" (Line 413).

AR: These formulations have been removed as we almost rewrote this section.

The first sentence of this paragraph has been revised as "For the SP area and the SI area, the maximal wind speed can reach up to 28 $\mathrm{m \cdot s^{-1}}$, while the variations of the optical properties along with wind speed are more complicated.".

1. Line 423 24 – 28 m/s is not quite strong wind, it's basically storm

AR: This formulation has been removed as we almost rewrote this section.

2. Line 424 You basically said that extinction will sharply increase under the condition of increased extinction, right? Re-read the sentence please or explain what I understood wrongly here.

AR: This sentence has been removed as we almost rewrote this section.

3. Line 425 Statistical significance should be supported by arguments here

AR: This statement has been removed.

- Line 440 Gradient change points? Where we can see that, which figure? Numbers are not mentioned here also to judge. Please update the entire paragraph with exact references to figures or numbers.

AR: Thanks. The sentence has been revised as "The gradient change point of $\alpha_{mar}$ (15 $\mathrm{m\cdot s^{-1}}$) is greater than that of $\beta_{mar}$ (10 $\mathrm{m\cdot s^{-1}}$), and above them the enhancement rate becomes lower." This paragraph has been entirely revised.

- Line 444 There is no section starting here. Please update this paragraph using the same guidelines as I gave above (more quantitative analysis).

AR: Thanks for the suggestion. This paragraph has been removed and the statements in this paragraph has been integrated to other parts.

1. Line 459. The language of this section has been visibly deteriorated compared to previous sections. Please revise it as well to make it more readable: the effort (line 461)?, grid? (465, maybe grid cell?)

AR: Thanks. "The effort" has been revised as "the attempt". The "wind speed grid" has been revised as "wind speed interval".

2. Line 478 "Quite similar" is unscientific. Quantitative arguments please

AR: "Quite similar" here refers to the AOD data source, the study areas and the wind speed range of this study is similar to Kiliyanpilakkil and Meskhidze (2011). The specific arguments were presented in the front of this sentence as "As described above, the $\mathrm{AOD_{mar}}$ data source (from spaceborne lidar observation), the study areas (remote ocean regions globally), and the wind speed range (0 $\mathrm{m\cdot s^{-1}}$ - 29 $\mathrm{m\cdot s^{-1}}$) of the $\mathrm{AOD_{mar}}$ - $ws$ relationship exploration in Kiliyanpilakkil and Meskhidze (2011) match well with those of this study."

3. Line 505 So how did you avoid enormously high errors due to wrongly fixed lidar ratios or at least quantified it?

AR: These errors belong to CALIOP retrieved AOD. Using CALIOP retrieved AOD can not avoid these errors while using Aeolus retrieved AOD can deal with this problem because Aeolus AOD retrieval is without the assumption of marine aerosol lidar ratios. This part is to illustrate the strength of Aeolus in the study of marine aerosol optical properties – wind speed relationship. To clarify this issue, the sentence has been revised as "Besides, as discussed in Section 4.4.2 of this paper, the particle size and the LR of the marine aerosol will vary with wind speed, so using the CALIOP $\mathrm{AOD_{mar}}$ retrieved with the fixed $\mathrm{LR_{mar}}$ may generate additional error in the exploration of the relationship between the $\mathrm{AOD_{mar}}$ and the wind speed."

1. **Conclusions:** Conclusions should be meticulously revisited after the revision. Most importantly, you missed the opportunity to give a holistic summary on what your analysis revealed in terms of physical behavior of aerosol. Did we learn something new about marine aerosols from physical standpoint? According to your conclusions – unlikely. Alternatively, you could fill up

this summary by speculating on the value of your findings with regards to Aeolus capabilities (low signal-to-noise ratio, low resolution, presence of clouds, complex relationship between ocean properties, AOD and wind speed). In the current form, you just plainly repeated the results and methodological choices in briefer form. Other, more concrete problems are here:

o   You hinted that you identified marine aerosol in the results, but it is rather implied that you introduced some methodological tool to assume marine aerosol. Once again, it is unclear why CALIPSO classification was not enough for this purpose (Line 568)

AR: We identified marine aerosol with CALIPSO classification and cloud screening method and did not use any other methodological tool to assume marine aerosol. It is considered CALIPSO classification is enough for this purpose. To reduce ambiguity, this paragraph has been rephrased as "Three study areas located in remote ocean were selected, which were named the North Pacific (NP) area, the South Pacific (SP) area and the South Indian (SI) area, respectively. Then we examined the domination of marine aerosol with the aerosol classification data provided by CALIOP VFM products. The proportions of marine aerosol in these three areas are all larger than 79% respectively while the percentage sums of marine aerosol and dusty marine aerosol are all above 90%. After quality control, cloud screening was conducted with the criteria (relative humidity and backscatter ratio), and 9%, 35%, 40% data was identified cloud contaminated in the altitude range of 0-2 km then was eliminated for the NP area, the SP area and the SI area, respectively. Finally, backscatter correction is applied to the Aeolus L2A products. These procedures allow us to obtain reliable, cloud-free marine aerosol optical properties and the corresponding wind speed."

o   Sensitivity analysis on distinguishing clouds from aerosols has not been shown, so I have doubts that you actually separated them quantitatively (Line 568). This statement definitely does not report your actual findings and therefore does not fit the conclusive tone it takes.

AR: The statistical analysis of cloud contaminated data has been conducted and the result was added in Section 3 and this part. The description in this part is "After quality control, cloud screening was conducted with the criteria (relative humidity and backscatter ratio), and 9%, 35%, 40% data was identified cloud contaminated in the altitude range of 0-2 km then was eliminated for the NP area, the SP area and the SI area, respectively." We think it can be the argument to the statement that "These procedures allow us to obtain reliable, cloud-free marine aerosol optical properties and the corresponding wind speed." in this paragraph.

o   Vague methodological descriptions are redundant and uninformative for readers in the conclusions (Lines 568 – 570 about defining the areas of the study for example)

AR: Thanks. The methodological description part has been shorten and revised as "Three study areas located in remote ocean were selected, which were named the North Pacific (NP) area, the South Pacific (SP) area and the South Indian (SI) area, respectively. Then we examined the domination of marine aerosol with the aerosol classification data provided by CALIOP VFM products. The proportions of marine aerosol in these three areas are all larger than 79% respectively while the percentage sums of marine aerosol and dusty marine aerosol are all above

90%. After quality control, cloud screening was conducted with the criteria (relative humidity and backscatter ratio), and 9%, 35%, 40% data was identified cloud contaminated in the altitude range of 0-2 km then was eliminated for the NP area, the SP area and the SI area, respectively. Finally, backscatter correction is applied to the Aeolus L2A products. These procedures allow us to obtain reliable, cloud-free marine aerosol optical properties and the corresponding wind speed."

o Repetitive formulations, partly reflecting your research aim/objectives are spotted (Line 574)

AR: Thank, we have revised the repetitive formulations.

o The information preceding the line 583 is redundant for conclusions from my point of view and should be either shortened or be more concrete in terms of reporting.

AR: Thanks. This paragraph has been removed. The statements were integrated into the next paragraph.

o Line 600 – repetitive, you said it three times in the conclusions. Report conclusion directly without repeating research aim/question.

AR: Thanks. The sentence has been revised as "The $\alpha_{mar} - ws$, $\beta_{mar} - ws$ models within and above MABL at remote ocean areas were established with Aeolus provided data."

o As mentioned, you do not shed the light on the difference between backscattering coefficient and extinction coefficient; they were treated identically and discussed as statistical parameters, not physical properties of aerosols.

AR: Extinction and backscattering are distinct, the two most typical optical properties measured by lidar, which represents the attenuation property on light and the scattering property on light of aerosol, respectively. They are defined as extensive optical properties, dependent on the aerosol concentration. The objective of analyzing both of them is to try to establish extinction-wind speed and backscattering-wind speed relationship as inputs of the radiation transfer model.

o Lines 595 – 599 You cannot judge about the size of the particle without either deriving microphysical properties of aerosols or at least by using Angstrom exponent. Lidar ratio does not fit such purpose, it just shows the ratio between light being extinct and light being scattered.

AR: Thanks for your comments. As you mentioned, it is better to use Ångström exponent to indicate the particle size of aerosol. However, with the only work wavelength of 355 nm, Aeolus can not provide Ångström exponent. Generally, aerosol lidar ratio shows the ratio between light being extinct and light being scattered. However, it was reported that the marine aerosol lidar ratio and its particle size have negative relationship (Masonis et al., 2003). The relationship between lidar ratio and marine aerosol particle size was introduced in Section 4.4.2 as:

"It is reported that the $LR_{mar}$ depends on the particle size, and specifically, with the reduction of the coarse mode, the total LR turns out to increase (Masonis et al., 2003). The possible reason for this phenomenon is that as the particles become smaller, the extinction is enhanced by the increasing sideward scattering and the backscatter gets weaker due to the decrease of the scattering cross section (Haarig et al., 2017)."

Therefore, it is considered that we can use marine aerosol lidar ratio implying its particle size. The statement in the conclusion section has been revised as "The $LR_{mar}$ and marine aerosol particle size have negative relationship (Masonis et al., 2003). From the relationship between the $LR_{mar}$ and the wind speed, it indicates that as the wind speed is increasing, the particle size of marine aerosol obviously becomes larger at relative low wind speed range, then could be broken up into smaller by wind at higher wind speed, and ultimately turns out a larger state again at very high wind speed."

o   Line 603 You have not analyzed turbulence in your study.

AR: Thanks. We have deleted this sentence.

o   Line 604 Wind speed bin is not scientific term known by a general reader.

AR: Thanks. We have deleted this sentence.

o   Line 605 The statement about 'not total similarity' of aerosol variation tendencies does not bring any new knowledge and is, therefore, not useful for conclusions.

AR: The statement has been revised as "Nevertheless, the regression curves of $\alpha_{mar}$ - $ws$ and

$\beta_{mar}$ - $ws$ above three study areas (the NP area, located in the Pacific Ocean, the low latitudes of the Northern Hemisphere; the SP area, located in the Pacific Ocean, the middle latitudes of the Southern Hemisphere; the SI area, located in the Indian Ocean, the middle latitudes of the Southern Hemisphere) are not totally consistent, while the meteorological and environmental conditions apart from wind are also distinct at different regions."

o   Lines 605 – 607 On the development of aerosols over ocean due to complex factors – you have not shown this in your study. As a suggestion, it is trivial because the complex relationship between aerosol evolution over ocean and processes on the ocean-atmosphere interface, as well as in the atmosphere; this relationship is obvious. Bring numbers and facts if you'd like to add some new value to this common knowledge.

AR: Thanks for the comments. We have tried to make this paragraph logic. The argument and the corresponding statement were revised as:

"Nevertheless, the regression curves of $\alpha_{mar}$ - $ws$ and $\beta_{mar}$ - $ws$ above three study areas (the NP area, located in the Pacific Ocean, the low latitudes of the Northern Hemisphere; the SP area, located in the Pacific Ocean, the middle latitudes of the Southern Hemisphere; the SI area, located in the Indian Ocean, the middle latitudes of the Southern Hemisphere) are not totally consistent, while the meteorological and environmental conditions apart from wind are also distinct at different regions. It implies that in order to obtain more precise $\alpha_{mar}$ and $\beta_{mar}$ models, besides wind speed, other meteorological and environmental factors, e.g., atmospheric stability, sea and air temperature, RH, etc. should participate in the establishment of the models, because the production, entrainment, transport and removal of the marine aerosol above the ocean are not only dominated by the wind, but also be impacted by these factors (Lewis and Schwartz, 2004). If future study is capable to obtain more other meteorological parameters above ocean, jointly analysing the aerosol optical properties and the wind together with them, more detailed information of marine aerosol production, entrainment, transport and removal will be acquired."

2. **Language and Format:** Language should be revised either by seeking assistance of colleagues, more familiar with the standard of academic English or automated software for language check at minimum. Multiple stylistic (PDF-oriented font format is spotted at line 48 as example), and more critically, grammar errors are found ('significate' at line 45 for instance). Formatting caveats also include redundant spacing (line 131), double bracketing, inconsistent introduction of acronyms, etc. Please eliminate all these drawbacks, which currently emphasize the raw condition of your draft.

AR: Thanks for your comments. We have carefully reviewed and revised the manuscript throughout to avoid grammar errors and formatting caveats.

**Mentioned references:**

1. Dionisi et al., (https://meetingorganizer.copernicus.org/EGU23/EGU23-16196.html)

2. Hu, Y., Stamnes, K., Vaughan, M., Pelon, J., Weimer, C., Wu, D., Cisewski, M., Sun, W., Yang, P., Lin, B., Omar, A., Flittner, D., Hostetler, C., Trepte, C., Winker, D., Gibson, G., Santa-Maria, M., 2008. Sea surface wind speed estimation from space-based lidar measurements. Atmos. Chem. Phys. 10.

3. Josset, D., Pelon, J., Hu, Y., 2010 Multi-Instrument Calibration Method Based on a Multiwavelength Ocean Surface Model. IEEE Geosci. Remote Sensing Lett. 7, 195–199. https://doi.org/10.1109/LGRS.2009.2030906

4. Josset, D., Pelon, J., Pascal, N., Hu, Y., Hou, W., 2018. On the Use of CALIPSO Land Surface Returns to Retrieve Aerosol and Cloud Optical Depths. IEEE Trans. Geosci. Remote Sensing 56, 3256–3264. https://doi.org/10.1109/TGRS.2018.2796850

5.  Josset, D., Pelon, J., Protat, A., Flamant, C., 2008. New approach to determine aerosol optical depth from combined CALIPSO and CloudSat ocean surface echoes: New approach to determine AOD. Geophys. Res. Lett. 35. https://doi.org/10.1029/2008GL033442

6.  Labzovskii L., van Zadelhoff G-J, Donovan D, De Kloe J., Josset D., 2022. How sensitive are Aeolus Lidar Surface Returns (LSR) to the types of surface? Insights for LSR-based retrieval of AOD over ocean by using Aeolus; https://doi.org/10.5194/egusphere-egu22-12079

7.  Li, Z., Lemmerz, C., Paffrath, U., Reitebuch, O., Witschas, B., 2010. Airborne Doppler Lidar Investigation of Sea Surface Reflectance at a 355-nm Ultraviolet Wavelength. Journal of Atmospheric and Oceanic Technology 27, 693–704. https://doi.org/10.1175/2009JTECHA1302.1

8.  Venkata, S., Reagan, J., 2016. Aerosol Retrievals from CALIPSO Lidar Ocean Surface Returns. MDPI Remote Sensing 8, 1006. https://doi.org/10.3390/rs8121006

---

## Author Comment (AC3)

This manuscript combines Aeolus and CALIPSO data along with ECMWF wind products that assimilate Aeolus to understand the relationship between wind speed and marine aerosol optical properties in 3 remote oceanic basins. The manuscript is well written and results are clearly presented. I recommend publication with minor revision:

1) The authors have a hard cut-off at 1 km to mark the MABL top throughough the manuscript. I would suspect this would vary somewhat with region. Can the authors comment on any ramifications of this? Would using ECMWF MABL help?

AR:

a. That the MABL height of the remote ocean areas is around 1 km was summarized from several references (Luo et al., 2014; Luo et al., 2016; Alexander et al., 2019).

b. Thank you for your advice, and we have calculated the mean values of ECMWF provided boundary layer heights at the three study areas for the time period of 20 April 2020 to 26 May 2021, which are $787.47 \pm 231.77$ m at the NP area, $939.39 \pm 360.20$ m at the SP area and $1005.29 \pm 366.60$ m at the SI area.

c. Moreover, the height resolution (0.25 km below 0.5 km, 0.5 km in the range of 0.5 km to 2 km) of Aeolus is relatively low. Using 1 km as the boundary to split Aeolus data vertically is feasible while more precise MABL height won't make more sense. Therefore, it is considered that the consequent lower layer (0-1 km) is capable to generally represent the MABL.

As you advised, the discussion about MABL heights are revised in the manuscript as:

"Referring the results of Luo et al. (2014), Luo et al. (2016) and Alexander et al. (2019), the MABL height of the remote ocean is summarized as around 1 km. Moreover, calculated with ECMWF provided boundary layer heights at the three study areas for the time period of 20 April 2020 to 26 May 2021, the mean values and the standard deviations are $787.47 \pm 231.77$ m at the NP area, $939.39 \pm 360.20$ m at the SP area and $1005.29 \pm 366.60$ m at the SI area. Hence, the boundary height of the two vertical layers is set as 1 km, approximately corresponding to the mean MABL height of remote ocean. Though the

MABL heights are variable and thus set as 1 km will lead to the potential inaccuracies, restricted by the relatively low height resolution of Aeolus (0.25 km below 0.5 km, 0.5 km in the range of 0.5 km to 2 km), utilizing more precise height boundaries won't make more sense. It is considered that the statistical results of the 0-1 km layers and the 1-2 km layers are capable to generally represent the atmospheric conditions within the MABL and above the MABL."

*Reference:*

*Alexander, S. P. and Protat, A.: Vertical profiling of aerosols with a combined Raman-elastic backscatter lidar in the remote Southern Ocean marine boundary layer (43–66°S, 132–150°E), J. Geophys. Res.-Atmos., 124, 12107–12125, https://doi.org/10.1029/2019JD030628, 2019.*

*Luo, T., Yuan, R., and Wang, Z.: Lidar-based remote sensing of atmospheric boundary layer height over land and ocean, Atmos. Meas. Tech., 7, 173–182, https://doi.org/10.5194/amt-7-173-2014, 2014.*

*Luo, T., Wang, Z., Zhang, D., and Chen, B.: Marine boundary layer structure as observed by A-train satellites, Atmos. Chem. Phys., 16, 5891–5903, https://doi.org/10.5194/acp-16-5891-2016, 2016.*

2) Line 41. Please specify - by area, by mass.

AR: The first sentence of Section 1 has been revised as "According to the Intergovernmental Panel on Climate Change (IPCC) Fifth Assessment Report, the total emission of marine aerosol (including marine primary organic aerosol) produced from ocean is 1400 to 6800 $\mathrm{Tg \cdot yr^{-1}}$, which is considered the largest natural aerosol input to the atmosphere globally (Boucher et al., 2013)." to make it more logical.

*Reference:*

*Boucher, O., D. Randall, P. Artaxo, C. Bretherton, G. Feingold, P. Forster, V.-M. Kerminen, Y. Kondo, H. Liao, U. Lohmann, P. Rasch, S.K. Satheesh, S. Sherwood, B. Stevens and X.Y. Zhang, 2013: Clouds and Aerosols. In: Climate Change 2013: The Physical Science Basis. Contribution of Working Group I to the Fifth Assessment Report of the Intergovernmental Panel on Climate Change [Stocker, T.F., D. Qin,*

*G.-K. Plattner, M. Tignor, S.K. Allen, J. Boschung, A. Nauels, Y. Xia, V. Bex and P.M. Midgley (eds.)].*

*Cambridge University Press, Cambridge, United Kingdom and New York, NY, USA.*

3) Line 95. Change from "is" to "was" as ALADIN stopped science operations at the end of April 2023.

AR: Thanks, revised.

4) Line 174. Why was version 3.41 used for January 2022- July 2022? Was version 4 data not available? This has implications for continuity of the VFM algorithm.

AR: Only version 3.41 of CALIOP VFM product is available for January 2022- July 2022, while version 4 data was not available in this period.

Figure 1 shows the Fig. 1 from Kim et al. (2018), which is the flowchart of the CALIPSO aerosol subtype selection scheme for tropospheric aerosols. In this flowchart, the blue-shaded region and blue-dotted arrows are used in V3 but removed in V4 while the red-shaded region and solid red arrows are newly added in V4. It can be learnt from the flowchart that the main framework of the aerosol subtype discrimination algorithm is the same between V3 and V4. Nevertheless, considering that the new-added procedure of version 4 in the red-shaded region identifies part of "polluted dust" aerosol subtype as "dusty marine", partly using version 3 VFM data will lead to the underestimate of "dusty marine" aerosol and the "marine" subtype discrimination will not be influenced. As describe above, using the version 3.41 of the CALIOP VFM data for January 2022- July 2022 in this study led to the underestimate of total marine aerosol portion, which means the real total marine aerosol percentage is larger than the statistic. To conclude, though there are several upgrades between version 4 and version 3, the statistical results of aerosol subtype are acceptable and the "marine aerosol dominating areas" conclusion is not affected.

[Figure]

Figure 1: Flowchart of the CALIPSO aerosol subtype selection scheme for tropospheric aerosols (Fig. 1 from Kim et al. (2018)).

To briefly clarify this fact, the relevant description has been added to the Section 4.1 as "It should be illustrated that "dusty marine" was a new aerosol subtype raised for the first time in the version 4.10 of the CALIOP VFM product and was absent in the version 3.41, which was identified from part of version 3.41's "polluted dust" with the criteria of "surface type" and "layer base altitude". Using the version 3.41 of the CALIOP VFM data for the period of 19 January 2022 to 4 July 2022 led to the underestimate of "dusty marine" portion and the total marine aerosol portion. Even though under the condition of underestimate, the percentage of total marine aerosol are larger than 90%, which means the real proportion of total marine aerosol is higher, and hence the conclusion that the marine aerosol dominates in the altitude range of 0-2 km above these three areas is still valid."

*Reference:*

*Kim, M.-H., Omar, A. H., Tackett, J. L., Vaughan, M. A., Winker, D. M., Trepte, C. R., Hu, Y., Liu, Z., Poole, L. R., Pitts, M. C., Kar, J., and Magill, B. E.: The CALIPSO version 4 automated aerosol classification and lidar ratio selection algorithm, Atmos. Meas. Tech., 11, 6107–6135, https://doi.org/10.5194/amt-11-6107-2018, 2018.*

5) The authors have not discussed the implication for the time offset of CALIPSO and Aeolus. While I don't think this is a major impact, the aerosol typing by CALIPSO will be after Aeolus observed winds and would be seeing the aftermath. Please consider discussing this point in the manuscript.

AR:

a. Generally, as for the closest orbits of Aeolus and CALIPSO, CALIPSO scanning tracks are about 4 h ahead of Aeolus (Dai et al., 2022). The lifetime of marine aerosol is usually with in the range of 1 day to 1 week, depending on its size (Boucher et al., 2013). Compared to the lifetime of marine aerosol, the 4 h time offset between CALIPSO and Aeolus observations is much shorter, and hence the same marine aerosol layer will be observed by two spaceborne lidars with very high probability.

b. The aerosol type analyses is a statistical result using more than two years CALIPSO aerosol typing data, which represent the background aerosol type conditions in the 0-2 km height atmospheric layers of the three study areas, and is considered to be independent of wind speed.

To conclude, the implication for the time offset of CALIPSO and Aeolus is intended to be ignored resulting from the much shorter time offset compared to the marine aerosol lifetime and the long-term statistical analyses of CALIPSO aerosol typing data.

*Reference:*

*Boucher, O., D. Randall, P. Artaxo, C. Bretherton, G. Feingold, P. Forster, V.-M. Kerminen, Y. Kondo, H. Liao, U. Lohmann, P. Rasch, S.K. Satheesh, S. Sherwood, B. Stevens and X.Y. Zhang, 2013: Clouds and Aerosols. In: Climate Change 2013: The Physical Science Basis. Contribution of Working Group I to the Fifth Assessment Report of the Intergovernmental Panel on Climate Change [Stocker, T.F., D. Qin, G.-K. Plattner, M. Tignor, S.K. Allen, J. Boschung, A. Nauels, Y. Xia, V. Bex and P.M. Midgley (eds.)]. Cambridge University Press, Cambridge, United Kingdom and New York, NY, USA*

*Dai, G., Sun, K., Wang, X., Wu, S., E, X., Liu, Q., and Liu, B.: Dust transport and advection measurement with spaceborne lidars ALADIN and CALIOP and model reanalysis data, Atmos. Chem. Phys., 22, 7975–7993, https://doi.org/10.5194/acp-22-7975-2022, 2022.*

6) Equation 1. Consider labeling Bmar, Aeolus as Bmar, Aeolus co or Bmar, Aeolus II to indicate it is the co polarized return.

AR: Thanks for the advice. Equation 1 has been revised as $\beta_{mar} = (1 + \delta_{mar,355nm}) \cdot \beta_{mar,Aeolus-co}$ .

7) Line 555. The "upward" trend in the middle of the plot seems very faint.(DIFF) Is it statistically significant?

AR: The revised Fig. 13 (Fig. 16 in the old version), shown as below, presents the $LR_{mar}$ variation with wind speed at $Layer_L$ and $Layer_H$, in (a) the NP area, (b) the SP area and (c) the SI area, respectively.

From this figure, the upward trends can be found at the middle wind speed range, 13 $m \cdot s^{-1}$-17 $m \cdot s^{-1}$ of the NP area, 9 $m \cdot s^{-1}$-16 $m \cdot s^{-1}$ of the SP area, 10 $m \cdot s^{-1}$-20 $m \cdot s^{-1}$ of the SI area, which are especially significant for $Layer_H$ . Therefore, it is considered statistically significant.

[Figure]

Figure 13: Averaged $LR_{mar}$ versus wind speed at $Layer_L$ and $Layer_H$ , in (a) the NP area, (b) the SP area and (c) the SI area, respectively.

---

## Author Response (AR2)

**Responses to RC2:**

1. Original Submission

1.1. Recommendation

Major revision

2. Comments to Author:

Overall opinion: Most comments I raised in the first iteration were addressed. However, the language (especially) in the abstract must be improved. The current narration suffers from unclarity, vagueness and is sometimes terminologically puzzling not because the reported phenomena are complex, but because the language is confusing. Besides that, I am concerned that the authors did not understand my remark on how important to understand the behavior of surface-contained bin while using Aeolus data and simply stated it is out of their scope by explaining what previous studies on that topic did from their point of view. I explain below why this might be a major pitfall especially given the fact that the authors registered enhanced backscatter in the lowest layer, e.g. marine boundary layer with the highest chance to hit the surface return. Except that, the results are convincing enough and confidently pave the way toward acceptance.

AR: Thanks for your comments. We have tried to improve the language and grammar throughout the manuscript according to your suggestion. In the aspect of the behavior of surface-contained bin while using Aeolus data, we supplemented the relevant explanation that "during the data processing, it was discovered that all data (Level 2A particle optical properties, Level 2C wind vectors) below 0.25 km, which could be contaminated by reflections from the land or ocean surface, were all screened out using Aeolus quality control flags, then the lowest data bins became at around 0.25 km. This may indicate that the actual altitude range of marine aerosol optical properties in $Layer_L$ is around 0.25 km to 1 km.

Although the data near the sea-air interface are missing, all available data avoids the contamination of the ground return signals and eliminates the risk of being affected by ocean surface dynamical conditions." in the revised manuscript. The point-to-point response are shown as below.

2.1. Major comments:

1. Title. Overall, the title is good, but it is beneficial for you to speak not about correlation but about physical link or association between marine aerosol optical properties and wind fields. This link is revealed then by correlation as a tool, nothing else. But it's up to you of course.

AR: Thanks for the advice. To better present the physical causal relationship between wind speed and marine aerosol optical properties, we intend to modify the title as "Effect of wind speed on marine aerosol optical properties over remote oceans with use of spaceborne lidar observations."

2. Abstract: From my point of view, it must be improved. It's very "wordy" ('Furthermore, thank to…" introductory structures are obviously unnecessary; the last sentence is just incomprehensible). It contains unclear terminology like "aerosol related atmospheric background states". Sentences like the one below also convey very simple ideas are structurally barely unreadable ("The marine aerosol extinction/backscatter coefficients and the background wind speeds show positive relationships and they were fitted by power law functions, of which the corresponding 2 R are all higher than 0.9"). Make abstract concise, clear, quantitative and free of vague terms that even atmospheric scientists will not comprehend. The current setbacks are not acceptable for the abstract of high-impact factor study.

AR: Thanks for the advice. We have reviewed the abstract. Removed, revised or rephrased the sentences that you pointed out. And we also made necessary changes to improve its clarity and coherence. The revised abstract is shown as below:

"In this paper, using Aeolus data, the relationships between the marine aerosol optical properties at 355 nm and the corresponding instantaneous co-located wind speeds of three remote ocean areas are investigated and analysed at two separate vertical atmospheric layers (0-1 km and 1-2 km, corresponding to the heights within and above marine atmospheric boundary layer (MABL)), revealing the effect of wind speed on marine aerosol over the remote ocean. Marine aerosol extinction/backscatter coefficients and background wind speeds show positive relationships. Their

correlations are modeled using power-law functions, with corresponding $R^2$ values all greater than

0.9. Both the MABL and the layer above it receive the marine aerosol produced and transported by the wind from the sea-air interface. The marine aerosol load in the lower layer (MABL) is stronger than in the higher layer. The intensity of marine aerosol extinction/backscatter coefficients enhancements caused by the background wind is greater in the MABL. The slope variation points occur during the marine aerosol extinction/backscatter coefficients increasing with wind speed. Above these points, the growth rate decreases. This may indicate that the wind-driven enhancement of marine aerosol involves two phases: a rapid growth phase with high wind dependence, followed by a slower growth phase after the slope variation points. The correlation between the marine aerosol optical depth at 355 nm and the corresponding wind speeds is established, and verified by comparing it with CALIPSO-derived results from previous research. The variation of the marine aerosol lidar ratio at 355 nm with wind speed is also examined, suggesting a possible increasing-decreasing-increasing trend of marine aerosol particle size as wind speed increases."

3. Language: Besides the remark on the abstract, the language suffers from unclarity and vagueness in some other parts of the manuscript. For instance, it is stated that "marine aerosol is mainly produced by wind" without mentioning sea/ocean surface. Grammatically, there is nothing wrong with this formulation, but technically, it's a misleading remark. There are more cases like that in the text I highlight in minor comments. Please thoroughly check the style and logic of your text even if grammar is ok. Ask English speaking colleague or language check service to help if needed; you will save your time by this.

AR: Thank you for your careful review. The sentence "Marine aerosol is mainly produced by wind" has been deleted in the revised manuscript. And we have modified all the cases you highlighted in minor comments. Besides, we also reviewed and checked the whole manuscript for the grammar and the logic.

4. Methodology:

• "it should be emphasized that the lowest altitude bins of Aeolus Level 2A and Level 2B products could be contaminated by reflections from the land or ocean surface, and are thus not representative for the atmospheric wind speed and the aerosol optical properties (Wu et al., 2022)." This made me thinking about methodological pitfalls of your study. Can you give me please the definition of atmospheric bin that does not contain surface from your methodological point of view? Do you have any risk to include this unaccounted bin into your MABL? Because the lowest lidar bin of Aeolus does not necessarily contain surface, this setting is changing, your lowest lidar bin can in some cases be way above 0 meters for instance.

AR: As you mentioned, the altitude of the Aeolus vertical bins could change along the track. Therefore, the altitudes of the lowest bins could be above or below the surface.

The altitude information of each bin is all provided in the Aeolus L2A/L2C products. In our methodological point of view, we used the altitude information of the bins to select data rather than

their order in the profiles. Specifically, for $Layer_L$ , we selected data bins with an altitude range of 0 km to 1 km, while for $Layer_H$ , we selected bins with an altitude range of 1 km to 2 km. Furthermore, after quality control with Aeolus quality control flags, there is no data available below 0.25 km. Therefore, it is considered that there is no risk in including surface bin into MABL.

In addition, the sentence "It is important to note that the lowest altitude bins of Aeolus observation products may contain the reflections from the land or ocean surface, thus they are contaminated and not representative of the atmospheric wind speed and the aerosol optical properties (Wu et al., 2022)." has been moved to Line 242 in the revised manuscript.

• My concern is confirmed by your answer on the comment by the way; you mention 0 km here ("Therefore, it is considered that the statistical results of the 0-1 km layers and the 1-2 km layers are capable to generally represent the atmospheric conditions within the MABL and above the MABL").

AR: Yes. As we described in the previous answer, we selected data for the two layers (0 km-1 km, 1 km-2 km) based on the altitude of each bin. This ensures that the statistical results of these data could represent the state within above the MABL.

• This further statement goes in conflict ( "in L Layer , the lowest Aeolus Level 2A products (particle optical properties) data bins with the altitude of lower than about 0.25 km, are absent to avoid the ground return signals' contamination.") with the statement above on the MABL range starting from 0 km. So you consider 0-1 km layers or 0.25 – 1.00 km layers, be precise with your methodological remarks.

AR: Thanks for your kind reminder. The actual meaning of this sentence was that in the data processing procedure it was found that after quality control all data below 0.25 km were screened out, then the lowest data bins were at around 0.25 km. To make it clear, this sentence has been modified as "during the data processing, it was discovered that all data (Level 2A particle optical properties, Level 2C wind vectors) below 0.25 km, which could be contaminated by reflections from the land or ocean surface, were all screened out using Aeolus quality control flags, then the lowest data bins became at around 0.25 km. This may indicate that the actual altitude range of marine aerosol optical properties in $Layer_L$

is around 0.25 km to 1 km. Although the data near the sea-air interface are missing, all available data avoids the contamination of the ground return signals and eliminates the risk of being affected by ocean surface dynamical conditions."

5. Conclusions: I'll go through conclusions here.

• Line 553 It is logical to say by using particle optical properties (Level 2A product -> in brackets), not the other way around because L2A is the Aeolus-specific term here. I already raised a concern that such terms are unknown for general readers.

AR: Thanks. The sentence has been revised as "By utilizing particle optical properties data (Level 2A products) and wind vector data (Level 2C products) provided by ALADIN, and L2 Vertical Feature Mask (VFM) products provided by CALIOP, the optical properties at 355 nm of pure marine aerosol are derived."

• Line 555 "Then" is not needed here, there is no temporal order or sequence here.

AR: Thanks, removed.

• Lines 553 – 559. Boldly emphasize what is your research aim, it's very vague now and blurred among all these activities that comprise your study.

AR: Thanks for the advice. To make it clearer, this paragraph has been revised as "By utilizing particle optical properties data (Level 2A products) and wind vector data (Level 2C products) provided by

ALADIN, and L2 vertical feature mask (VFM) products provided by CALIOP, the optical properties at 355 nm of pure marine aerosol are derived. The correlation between marine aerosol optical properties at 355 nm and the instantaneous co-located wind speed over remote ocean areas is investigated and discussed at two separate vertical atmospheric layers ( $Layer_L$ with the height of 0-1 km and $Layer_H$ with the height of 1-2 km, corresponding to the heights within and above marine atmospheric boundary layer (MABL)), revealing the effect of wind speed on marine aerosol within and above the MABL over the remote oceans."

• Line 569 Which statistical results, just name parameters or metrics directly

AR: Thanks. The sentence has been revised as "The correlation between the marine aerosol optical properties (extinction coefficient ( $\alpha_{mar}$ ) and backscatter coefficient ( $\beta_{mar}$ )) at 355 nm and the wind speed ( $ws$ ) are analysed at $Layer_L$ and $Layer_H$ , for three study areas respectively." in the revised manuscript.

• Line 572 The fact that MABL can receive marine aerosol is trivial finding. Perhaps, Aeolus can detect it is a thing to report?

AR: Thanks for the advice. This sentence has been revised as "It is found that the Aeolus observations can provide evidence of the fact that the MABL receives the marine aerosol produced and transported from the air-sea interface. Furthermore, the observations suggest that even the layer above the MABL may also receive the marine aerosol input." in the revised manuscript.

• Line 573 Once again, enhancement in the lowest MABL layer is suspected to be linked to inclusion of surface bin. Let me know why I am wrong here please.

AR: After using the quality control flag provided by Aeolus, it was found that there is no available data left below 0.25 km. Therefore, it is considered that no data bin could be linked to the inclusion of surface bin.

To illustrate this issue, the description has been added in Line 250 of the "Methodology" section, as "during the data processing, it was discovered that all data (Level 2A particle optical properties, Level 2C wind vectors) below 0.25 km, which could be contaminated by reflections from the land or ocean surface, were all screened out using Aeolus quality control flags, then the lowest data bins became at around 0.25 km. This may indicate that the actual altitude range of marine aerosol optical properties in $Layer_L$ is around 0.25 km to 1 km. Although the data near the sea-air interface are missing, all available data avoids the contamination of the ground return signals and eliminates the risk of being affected by ocean surface dynamical conditions."

The phenomenon that "The marine aerosol enhancements caused by the background wind are more intensive at the MABL." may result from that the MABL is closer to the sea-air interface so it will receive more effects. The explanation for this phenomenon has been added after this sentence in the revised manuscript, as "This may be due to the MABL's proximity to the sea-air interface, making it more susceptible to such effects."

• Line 575. Unsupported surmise about "might illustrate", give facts or rationale behind please.

AR: We think the phenomenon that "the slope variation points (15 $m \cdot s^{-1}$ for $\alpha_{mar}$ and 10 $m \cdot s^{-1}$ for $\beta_{mar}$ ) were found during $\alpha_{mar}$ and $\beta_{mar}$ increasing with wind speed, above which the growth

rates become lower" illustrates the statement that "the enhancement of marine aerosol driven by wind includes two phases, among them one is a rapid growth phase with high dependency of wind, and another is a slower growth phase with higher fluctuations after the slope variation points".

We calculated the averaged slopes and the corresponding standard deviations of $\alpha_{mar}$ and $\beta_{mar}$ of the two growth phases (below and above the slope variation points) and have supplemented these results in Table 2 of the revised manuscript, as the argument of this statement. Table 2 and the relevant descriptions in the revised manuscript are shown as below:

"We named these two wind speed (15 $m \cdot s^{-1}$ for $\alpha_{mar}$, 10 $m \cdot s^{-1}$ for $\beta_{mar}$) "slope variation point" in this paper. Table 2 presents the averaged slopes (Mean) and the corresponding standard deviations (SD) of $\alpha_{mar}$ and $\beta_{mar}$ below and above the slope variation point, for the two layers of the SP and SI areas. All of the averaged slopes below the slope variation points are larger than those above the slope variation points, except for the $\alpha_{mar}$ in the SI area. The reason for the inverse results of $\alpha_{mar}$ in the SI area may be due to its rapid increase above 24 $m \cdot s^{-1}$. All of the SDs of $\beta_{mar}$ above the slope variation points are greater than those below, indicating a more fluctuating growth phase above the slope variation points. These results could provide the evidence for the statement that the wind-driven enhancement of marine aerosol includes two phases: one is a rapid growth phase with high dependency of wind, and another is a slower growth phase with higher fluctuations.

**Table 2: Mean ± SD of the slopes below and above the slope variation point, grouped by areas and layers.**

| Optical property | Area | Layer | Mean ± SD of the slopes [ $Mm^{-1} \cdot (m \cdot s^{-1})^{-1}$ for $\alpha_{mar}$, $Mm^{-1} \cdot sr^{-1} \cdot (m \cdot s^{-1})^{-1}$ for $\beta_{mar}$ ] | |
| --- | --- | --- | --- | --- |
| | | | Below slope variation point | Above slope variation point |
| $\alpha_{mar}$ | SP | H | 2.48±1.81 | 1.79±5.71 |
| | | L | 3.11±4.62 | 1.26±16.11 |
| | SI | H | 1.96±3.10 | 2.81±12.59 |
| | | L | 2.16±4.28 | 3.28±8.79 |
| $\beta_{mar}$ | SP | H | 0.20±0.17 | 0.07±0.17 |
| | | L | 0.28±0.11 | 0.12±0.29 |
| | SI | H | 0.21±0.16 | 0.09±0.20 |
| | | L | 0.22±0.16 | 0.12±0.13 |

"

• Line 576… Why LRmar results are omitted from abstract? If they are not relevant, then there is no

sense to put them into conclusions as well. Please harmonize conclusions and abstract by content.

AR: Thanks. We have revised a more detailed description of marine aerosol lidar ratio in the abstract, as "The variation of the marine aerosol lidar ratio at 355 nm with wind speed is discussed, suggesting a possible increasing-decreasing-increasing trend of marine aerosol particle size as wind speed increases."

• Line 589 What do you consider as "total consistency" from statistical point of view. I'm hinting here – give numbers and rely on them while making such statements.

AR: Actually, what we meant here is that the regression curves are different because the regression equations of three study areas are not the same. To reduce the misleading, the phrase "not totally consistent" was revised as "inconsistent".

• Line 590 Conditions cannot participate anywhere, please rephrase and check the language throughout the article.

AR: The sentence has been revised as "while the meteorological and environmental parameters, apart from wind, differ across various regions." in the revised manuscript.

• Line 594 – 595 Implication sentence is not giving any new insights. If more data are available, more information can be derived. This has been known and is clear without this study and should be therefore removed.

AR: Thanks, removed.

• I feel that the sentence which can explain what your study added to this research field is missing. Let me elaborate on what your study added to this research field from my point of view. See below quick speculation based on your results that you may or may not use as your implications if you'd like to.

i. Perhaps, you deepened our understanding of relationship between marine aerosol optical properties and wind in remote ocean areas using unique Aeolus observations given its truly unique setup and ability to deliver winds?

ii. What I also think is nice that you demonstrated that the relationship between aerosol optical properties and wind speed are MORE COMPLEX than we might have expected if you need so many more parameters to scrutinize it in different regions?

iii. Moreover, this is an important lesson for missions similar to Aeolus like Aeolus-2? Where a synergy of aerosol and wind observations gives unique and proven ability to quantify aerosol-wind speed interactions in poorly observed regions of ocean? You showed basically very useful application of Aeolus synergistic analysis

AR: Thanks for your suggestions. We found all these three comments to be very helpful so, we incorporated them into the final paragraph of the revised manuscript. The revised final paragraph is shown as below:

"This study demonstrates Aeolus' ability to quantify interactions between aerosols and wind speeds in poorly observed ocean regions through a synergy of aerosol and wind observations. The $\alpha_{mar\_ws}$, $\beta_{mar\_ws}$ models within and above MABL at remote ocean areas were established with Aeolus provided data. These models deepens our understanding of the correlation between marine aerosol optical properties and wind in remote ocean areas across two vertical layers based on the unique setup and the ability to deliver winds of Aeolus observations. Nevertheless, the regression curves of $\alpha_{mar\_ws}$ and $\beta_{mar\_ws}$ above three study areas (the NP area, located in the Pacific Ocean, the low

latitudes of the Northern Hemisphere; the SP area, located in the Pacific Ocean, the middle latitudes of the Southern Hemisphere; the SI area, located in the Indian Ocean, the middle latitudes of the Southern Hemisphere) are inconsistent, while the meteorological and environmental parameters, apart from wind, differ across various regions. It implies that the relationships between marine aerosol optical properties and wind speed are more complex than a linear or exponential relation. In order to obtain more precise

$\alpha_{mar}$ and $\beta_{mar}$ models, besides wind speed, other meteorological and environmental factors, e.g., atmospheric stability, sea and air temperature, RH, etc. should participate in the establishment of the models, because the production, entrainment, transport and removal of the marine aerosol above the ocean are not only dominated by the wind, but also be impacted by these factors (Lewis and Schwartz, 2004)."

6. Previous comments from my side:

• Third, you discuss marine aerosol optical properties – near wind relationship from lidar perspective, but omitted a large corpus of works dedicated to this issue. I think you misunderstood my point on this initial comment here. You used a backscatter as a source of information about atmosphere using Aeolus over ocean, but the problem is that previous lidar studies have demonstrated that attenuated particle backscatter can be driven by the change of surface conditions of the ocean. And then, Aeolus has very coarse vertical resolution which means that the lowest bin contain surface. On top of that, you normally do not know where this surface bin is located because range gate of Aeolus setup was spatio-temporally changing. Most studies I mentioned are centered around one physical principle – atmospheric backscatter derived from lidar over ocean surface can be not affected, but is normally literally driven by the ocean surface conditions, not only atmospheric condition. You can learn a lot from this experience and be therefore more aware about these pitfalls. From statistical viewpoint, you may have been encountering adverse physical confounder in your analysis that you could not refute methodologically. You assumed in your answer that it's not the scope of your interest, but this assumption does not refute the fact that lidar returns from Aeolus could be affected by this ocean surface-related phenomena, thus inflicting unknown bias in your results about atmosphere. Your abstract immediately makes this assumption about possibility to study wind-aerosol relationship over ocean without considering surface contamination and range gate variations (lines 22-23 about "different vertical layers"). The wind-ocean surface return aspect of this issue does not seem to be critical in the current iteration though because in the mean-time, Aeolus-related work about surface returns showed that there is no wind-backscattering sensitivity at the lowest bin of Aeolus for oceans (highly non nadir incidence, weakening the returns, 355 nm – transparent ocean surface for beam heading to nadir). For experts with similar concerns to mine, this works makes it easier you can just instead add some sentence like that for clarity in your methodology. "On one hand, it is known that for ocean applications, lidar attenuated backscatter can be affected by the processes at the surface of ocean, namely, stronger winds, weaker backscattering [Josset et al. 2008; https://agupubs.onlinelibrary.wiley.com/doi/full/10.1029/2008GL033442]. On the other hand, the recent study on Aeolus surface returns [Labzovskii et al., 2023; https://www.nature.com/articles/s41598-023-44525-5] indicated that Aeolus returns are unlikely sensitive to ocean surface dynamics (related to wind), which makes the analysis of AOD/marine boundary layer conditions/*any atmospheric phenomenon* free from adverse effects, stemming from ocean surface." Rephrase it the way you want, though if you got my idea here.

AR: Thanks for the careful and patient explanation. We thought we got your idea about the relationship

between the attenuated backscatter coefficients of the bins containing ocean surface and the ocean surface dynamical conditions. We have incorporated your comments into the methodology section of the revised manuscript. The supplemented descriptions are shown as below:

"It is important to note that the lowest altitude bins of Aeolus observation products may contain the reflections from the surface or even be subsurface, thus they are contaminated and not representative of the atmospheric wind speed and the aerosol optical properties (Wu et al., 2022). Regarding the ocean applications of spaceborne lidars observations, it is known that the lidar attenuated backscatter coefficients of the bin containing the ocean surface can be affected by the processes at the surface of ocean, namely, stronger winds resulting in weaker backscattering (Josset et al., 2008). Labzovskii et al. (2023) indicated that Aeolus return signals are unlikely sensitive to ocean surface dynamical conditions (related to wind), which makes the analysis of marine aerosol optical properties in the MABL free from adverse effects stemming from ocean surface. Nevertheless, during the data processing, it was discovered that all data (Level 2A particle optical properties, Level 2C wind vectors) below 0.25 km, which could be contaminated by reflections from the land or ocean surface, were all screened out using Aeolus quality control flags, then the lowest data bins became at around 0.25 km. This may indicate that the actual altitude range of marine aerosol optical properties in $\text{Layer}_L$ is around 0.25 km to 1 km.

Although the data near the sea-air interface are missing, all available data avoids the contamination of the ground return signals and eliminates the risk of being affected by ocean surface dynamical conditions."

• Comment about Hoaglin et al. 1986. Your said that this approach of eliminating outliers… that "is a widely used approach in data analysis, especially in the statistical analysis". Most statistical approaches are tools to be used and they do not fit for every situation where we are having "big data" problem. You are working with physical observations. While exploratory data analysis can be a helpful approach in identifying outliers in certain types of data, it may not be suitable for physical observations where factors such as distribution, range, and scientific context need to be taken into consideration. Please give references of applications of this method to geophysical/atmospheric data or justify the choice.

• Same comment: You gave a very simple explanation like "Before the elimination, the outliers of extinction coefficients and

backscatter coefficients can catch up to 1000 1 Mm − and 30 1 1 Mm sr − −    while generally the particulate extinction coefficients and backscatter coefficients are within 300 1 Mm− and 101 1 Mm sr − −    ". But if this explanation is true the simplest confidence interval approach/z-score, any more simple statistical way to find outliers, would work for this purpose, no?

AR: Sorry for the misleading that the name of the outlier labeling and elimination method we used is "boxplot analysis", not Tukey's test. Below are some references that use the same method to label and eliminate outliers in the field of atmospheric research.

■ In the "Chapter 3 - Empirical Distributions and Exploratory Data Analysis" (Page 23-70) of the volume "Statistical Methods in the Atmospheric Sciences" (Volume 100) of the book series "International Geophysics", Wilks (2011) introduced the boxplot analysis for outliers labeling and elimination in atmospheric scientific analysis.

■ van Zoest et al. (2018) used the boxplot analysis to identify and label the outliers in a full year's urban air quality sensor observations divided into 16 spatio-temporal classes.

■ Guevara et al. (2021) used boxplot analysis to help establish the quality assurance flags of the NO2 observational dataset in the research of "Time-resolved emission reductions for atmospheric

chemistry modelling in Europe during the COVID-19 lockdowns".

- ■ Santhana Lakshmi and Vijaya (2023) also applied the boxplot analysis to identify the outlier present in the dateset, which are air pollution parameters including PM2.5, PM10, CO, SO2, ozone, NOX, and NH3.

So we consider that this method is applicable to atmospheric data, including particle optical properties provided by Aeolus.

**References:**

*Guevara, M., Jorba, O., Soret, A., Petetin, H., Bowdalo, D., Serradell, K., Tena, C., Denier van der Gon, H., Kuenen, J., Peuch, V.-H., and Pérez García-Pando, C.: Time-resolved emission reductions for atmospheric chemistry modelling in Europe during the COVID-19 lockdowns, Atmos. Chem. Phys., 21, 773–797, https://doi.org/10.5194/acp-21-773-2021, 2021.*

*Santhana Lakshmi, V., Vijaya, M.S. (2023). An Exploratory Data Analysis on Air Quality Data of Trivandrum. In: Joshi, A., Mahmud, M., Ragel, R.G. (eds) Information and Communication Technology for Competitive Strategies (ICTCS 2022). ICTCS 2022. Lecture Notes in Networks and Systems, vol 623. Springer, Singapore. https://doi.org/10.1007/978-981-19-9638-2_68.*

*van Zoest, V.M., Stein, A. & Hoek, G. Outlier Detection in Urban Air Quality Sensor Networks. Water Air Soil Pollut 229, 111 (2018). https://doi.org/10.1007/s11270-018-3756-7.*

*Wilks, D. S., Chapter 3 - Empirical Distributions and Exploratory Data Analysis, International Geophysics, Academic Press, 100, 23-70, https://doi.org/10.1016/B978-0-12-385022-5.00003-8, 2011.*

2.2. Minor comments (p-ll, where P – page, LL – line).

• Line 1 Marine aerosol is produced by wind, but what about role of ocean in this production? I once again refer to this language-related vagueness in your descriptions here.

AR: Thanks. This sentence has been removed in the revised manuscript.

• Line 20 "Aeolus, the worldwide first ever wind detection lidar satellite", it has sense to use more conventional terms for explaining what Aeolus is. See abstract here for good example (https://amt.copernicus.org/articles/14/6305/2021/)

AR: Thanks. This sentence has been removed in the revised manuscript.

• Line 25 Do you really provide a discussion or also analysis? Readers may think that you speculate on existing findings if you put it this way, but you did way more beyond discussion.

AR: Thanks. We have replaced the word "discussed" with "analysed" in the revised manuscript.

• Line 30 – 31 "The marine aerosol load at the lower layer (MABL) is stronger than at the higher layer." Here I refer to the comment on the methodological choices. If by some chance you included surface bin into your assumed MABL bins by not tracking the actual surface bin/bins of Aeolus, your backscattering in this bin would be higher than in the bin above, but due to other reason – surface return backscatter (nothing to do with aerosol backscatter).

AR: As we stated in the previous replies, we used the altitude information of the bins provided in the Aeolus products to select data. Specifically, for $\text{Layer}_L$, we selected data bins with an altitude range of 0 km to 1 km, while for $\text{Layer}_H$, we selected bins with an altitude range of 1 km to 2 km. Furthermore, after quality control with Aeolus quality control flags, there is no data available below 0.25 km. Therefore, it is considered that there is no risk in including surface bin into MABL.

• Line 32 "The gradient change points of marine aerosol extinction/backscatter coefficients appear during the growth of them with wind speed, above which the growth rate becomes lower." I try to understand it, but I don't. Gradient change points? I already raised the concern about this term (Line 440 of original submission). This is very counterintuitive term. Think about that. Gradient change can

happen temporally and spatially (both vertical and horizontal direction). And then "growth of them with wind" -> Very bad wording for English. Rephrase please, it's completely unclear.

AR: Thanks for the comments. We have replaced "gradient change point" with "slope variation point" throughout the manuscript.This sentence has been revised as "The slope variation points occur during the marine aerosol extinction/backscatter coefficients increasing with wind speed."

• Lines 33 – 34. It might illustrate that it is also a statistical coincidence without further statistical arguments. How to prove your claim here or at least support it?

AR: We calculated the averaged slopes and the corresponding standard deviations of $\alpha_{mar}$ and $\beta_{mar}$ of the two growth phases (below and above the slope variation points) and have supplemented these results in Table 2 of the revised manuscript, as the argument of this statement. Table 2 and the relevant descriptions in the revised manuscript are shown as below:

"We named these two wind speed (15 $\mathrm{m \cdot s^{-1}}$ for $\alpha_{mar}$, 10 $\mathrm{m \cdot s^{-1}}$ for $\beta_{mar}$) "slope variation point" in this paper. Table 2 presents the averaged slopes (Mean) and the corresponding standard deviations (SD) of $\alpha_{mar}$ and $\beta_{mar}$ below and above the slope variation point, for the two layers of the SP and SI areas. All of the averaged slopes below the slope variation points are larger than those above the slope variation points, except for the $\alpha_{mar}$ in the SI area. The reason for the inverse results of $\alpha_{mar}$ in the SI area may be due to its rapid increase above 24 $\mathrm{m \cdot s^{-1}}$. All of the SDs of $\beta_{mar}$ above the slope variation points are greater than those below, indicating a more fluctuating growth phase above the slope variation points. These results could provide the evidence for the statement that the wind-driven enhancement of marine aerosol includes two phases: one is a rapid growth phase with high dependency of wind, and another is a slower growth phase with higher fluctuations.

**Table 2: Mean ± SD of the slopes below and above the slope variation point, grouped by areas and layers.**

| | | | Mean ± SD of the slopes | |
|---|---|---|---|---|
| Optical property | Area | Layer | $[\mathrm{Mm^{-1} \cdot (m \cdot s^{-1})^{-1}}$ for $\alpha_{mar}$, $\mathrm{Mm^{-1} \cdot sr^{-1} \cdot (m \cdot s^{-1})^{-1}}$ for $\beta_{mar}]$ | |
| | | | Below slope variation point | Above slope variation point |
| $\alpha_{mar}$ | SP | H | 2.48±1.81 | 1.79±5.71 |
| | | L | 3.11±4.62 | 1.26±16.11 |
| | SI | H | 1.96±3.10 | 2.81±12.59 |
| | | L | 2.16±4.28 | 3.28±8.79 |
| $\beta_{mar}$ | SP | H | 0.20±0.17 | 0.07±0.17 |
| | | L | 0.28±0.11 | 0.12±0.29 |

| | | | |
|---|---|---|---|
| SI | H | $0.21 \pm 0.16$ | $0.09 \pm 0.20$ |
| | L | $0.22 \pm 0.16$ | $0.12 \pm 0.13$ |

"

• Line 35 "As derived data from Aeolus…", this sentence has no grammar and stylistic sense in the present form. Please rephrase the entire conclusive remark as well.

AR: Thanks. We rephrased the final sentence in the original manuscript as "The correlation between the marine aerosol optical depth at 355 nm and the corresponding wind speeds is established, and verified by comparing it with CALIPSO-derived results from previous research. The variation of the marine aerosol lidar ratio at 355 nm with wind speed is also examined, suggesting a possible increasing-decreasing-increasing trend of marine aerosol particle size as wind speed increases." in the revised manuscript.

• Lines 107 – 109. As mentioned above, Labzovskii et al., 2023; [https://www.nature.com/articles/s41598-023-44525-5] demonstrated that surface-containing Aeolus bin (where range bin has intersection with digital elevation model) manifest surface reflectivity over land with very high agreement with Lambertian equivalent reflectance from passive instrument. Note that you cannot speculate on the lowest bin because lowest bin can be subsurface in case of water if you do not track what is the lowest bin of Aeolus using either DEM data or Aeolus information on surface location (vertically).

AR: Thanks for the reminder. We have rephrased this sentence as "It is important to note that the lowest altitude bins of Aeolus observation products may contain the reflections from the surface or even be subsurface, thus they are contaminated and not representative of the atmospheric wind speed and the aerosol optical properties (Wu et al., 2022)." and moved it to line 242 of the methodology section. We used the altitude information of the bins provided in the Aeolus products to select data. After quality control using Aeolus quality control flags, no data is available below 0.25 km. Therefore, it is considered that all data bins are above the ocean surface, avoiding the impact of surface reflectivity.

• Line 113 "As mentioned above, Aeolus can provide global high spatial and temporal resolution aerosol optical properties profiles and wind speed profiles despite the lack of the lowest bins close to the ground". What is the lack of the lowest bins close to the ground? How do you judge abundance and lack of the bins quantitatively?

AR: We decided to remove "despite the lack of the lowest bins close to the ground" in the revised manuscript.

• Line 121 How did you quantify where is ocean surface? You make references to MABL, but it is unclear. The same applies to the orange block of Figure 2.

AR: We can quantify the 0 km altitude of Aeolus data bins using its altitude information. Therefore, we replaced "ocean surface" with "0 km" in the revised manuscript.

• Line 201 Not only contaminated, but also totally attenuated if your clouds are above studied aerosol layer, right?

AR: Yes. If the clouds above the studied aerosol layers are thick enough to attenuate all the light signals, then there will be no available data for the aerosol layers below. We revised this sentence as "The Aeolus products do not differentiate between aerosol and cloud layers, which means that the particle optical properties of a single data bin may contain a mixture of both types of information."

**Responses to RC4:**

The manuscript is greatly improved after the revision. The authors well-addressed the reviewer's comments and suggestions. One additional comment is:

The authors claim that 'the atmosphere of the two vertical layers bill both receive the marine aerosol input produced and transported by the wind and the turbulence'. However, the decrease of lidar ratio with wind speed indicates that 'marine aerosol get larger'. Given that aerosol optical properties are impacted by both the concentration and particle size, it is unclear whether the increase of optical properties with wind speed is dominated by more aerosols or by the growth of aerosols. So I believe the authors should tune down a bit as the manuscript strongly indicates that the increase of optical properties is caused by more aerosol inputs without fully considering the impact of aerosol particle size.

AR: Thanks for your reminder. We added the comment that "As aerosol optical properties are affected by both particle concentration and size, this reminds us that the increase in $\alpha_{mar}$ and $\beta_{mar}$ with wind speed may not only be due to the enhancement of particulate quantity produced from the sea-air interface, but may also be impacted by the size variation." in line 604 of the summary and conclusion section in the revised manuscript.

---

## Author Response (AR3)

Referee #2 comments:

I think the authors have fairly addressed all my concerns during my first round revision. Only minor remarks can be incorporated (optionally). The exception is formatting that suffers from some structural+language caveats and oversights that should be addressed.

AR: Thanks. We have reviewed, revised and responded according to your comments point by point. Besides, we also checked and revised the language and the format throughout the manuscript.

Abstract

- It's up to you, but it's better to avoid even such common technical terms like R^2 unexplained

AR: Thanks. We have replaced "corresponding R^2 values" with "the corresponding coefficients of determination".

- I would nail down the abstract with one strong implication sentence, explaining scientific significance of this paper. You may recall three implication recommendations I provided in the first iteration as examples, but you can come up with your powerful closing remark indeed.

AR: Thanks for the suggestion. The sentence that "This study deepens the understanding of the correlation between marine aerosol optical properties and wind speed based on the synergy of aerosol and wind observations from Aeolus, demonstrating their relationships are more complex than a linear or exponential relation." has been added in the final of the abstract.

Introduction

- You refer to 5th IPCC report, while the latest report is #6, why you refer to #5 then? Especially given the fact you mentioned IPCC 2021 report reference later on

AR: We found the statement "According to the Intergovernmental Panel on Climate Change (IPCC) Fifth Assessment Report, the total emission of marine aerosol (including marine primary organic aerosol) produced from ocean is 1400 to 6800 $Tg \cdot yr^{-1}$, which is considered the largest natural aerosol input to the atmosphere globally" in 5th IPCC report, but not in IPCC 2021. So we decided to refer to 5th IPCC report here.

Instrument Description and Methodology

- Line 143. Do we need a reference for Baseline 14 reprocessing here? Just thinking

AR: Thanks. We supplemented the website of the "Aeolus Online Dissemination System" as a reference here, which is shown as "(https://aeolus-ds.eo.esa.int/oads/access/, last access: 16 February 2023)".

- 2.2 Once again, I think giving names of instrument and satellite itself using slash might be misleading.

AR: Thanks. The title of the section 2.1 and 2.2 has been revised as "ALADIN" and "CALIOP".

- You refer to Figure 2 (line 171) earlier than to Figure 1. This is not correct from format point of view and it will be most likely highlighted as an error during proof-read

AR: Thanks. We moved the sentence referring to Fig. 2 to the final sentence of section 3, which is shown as "The procedures of the study methodology are summarized in a flowchart, shown as Fig. 2."

Results

-Figures 4-7. Too many figures one by one to comprehend through text-figure reading. It's very inconvenient. Also, technically it might be wrong because you discuss Figure 7 after you showed it. Normally, you describe the results in the figure BEFORE you show this figure in the manuscript. It is strongly recommended only to mentioned previously described figures after these figures are shown,

but not to discuss them in such details. Once again, think about a general reader.

AR: Thanks for the advice. We have rearranged the order of the text and figures to ensure that the detailed descriptions of the figures are all before the figures.

- Line 470. I am not sure that research can be used as countable noun ("researches", not sure it's correct here and elsewhere). Moreover, add these "all previous studies" as references here. If you say "almost all the previous researchers", then one may wonder, which studies were not focused on this aspect? You may add one short remark on that.

AR: We replaced "almost all the previous researches" with "almost all the previous studies". In the introduction section, we mentioned a study focusing on extinction of marine aerosol and wind speed (Shin et al., 2014), so we decided to retain the description that "almost all the previous studies".

*Reference:*

*Shin, D. H., Müller, D., Choi, T., Noh, Y. M., Yoon, Y. J., Lee, K. H., Shin, S. K., Chae, N., Kim, K., and Kim, Y. J.: Influence of wind speed on optical properties of aerosols in the marine boundary layer measured by ship-borne DePolarization Lidar in the coastal area of Korea, Atmospheric Environment, 83, 282-290, https://10.1016/j.atmosenv.2013.10.027, 2014.*

Summary and Cocnlusions

- Lines 571-572, mentioned not only ALADIN, but Aeolus as well as the latter term is more known for remote sensing community not directly working with the ALADIN data. Let alone, it's a more known term for general readers for the mission.

AR: Thanks for the suggestion. The sentence has been revised as "By utilizing particle optical properties data (Level 2A products) and wind vector data (Level 2C products) provided by ALADIN onboard Aeolus satellite, and L2 vertical feature mask (VFM) products provided by CALIOP onboard CALIPSO satellite, the optical properties at 355 nm of pure marine aerosol are derived." in the revised manuscript.

Format

- Perform a thorough language and format check, you have some caveats like trailing spaces (line 70 for example). Wrong verb ("is" instead of "are" in Line 96) use. Wrong tense use ("flies" in line 128, while Aeolus is not flying anywhere anymore, so it's technically wrong). It's not a good sign for final version of the paper. Your paper should be well readable by general readers.

AR: Thanks. We checked the language and the format throughout the manuscript and corrected them as far as possible.

---

## Author Response (AR4)

Dear Editor, Prof. Xiaohong Liu,

Thank you for your careful review and helpful suggestions.

1. We have reorganized and condensed the Abstract, and revised the "Summary and Conclusions" section, following ACP guidelines. The revised Abstract and the "Summary and Conclusion" section are shown as below:

[revised manuscript text omitted]

2. As for the citation of IPCC report, we actually referred to the **Summary for Policymakers** provided by IPCC. The citation information for the Summary for Policymakers is "IPCC: Summary for Policymakers, in: Climate Change 2021: The Physical Science Basis. Contribution of Working Group I to the Sixth Assessment Report of the Intergovernmental Panel on Climate Change, edited by: Masson-Delmotte, V., Zhai, P., Pirani, A., Connors, S. L., Péan, C., Berger, S., Caud, N., Chen, Y., Goldfarb, L., Gomis, M. I., Huang, M., Leitzell, K., Lonnoy, E., Matthews, J. B. R., Maycock, T. K., Waterfield, T., Yelekçi, O., Yu. R., and Zhou, B., Cambridge University Press, Cambridge, United Kingdom and New York, NY, USA, 3-32, https://doi.org/10.1017/9781009157896.001, 2021." Feurdean et al. (2022), Callewaert et al. (2022), and Terhaar et al. (2022) also cited the same reference. Their citation formats of the Summary for Policymakers in the text are "(IPCC, 2021)". So we decided to retain the citation formats "(IPCC, 2021)" in the text.